# Precise modulation of transcription factor levels identifies features underlying dosage sensitivity

**Sahin Naqvi** [1,2,3], **Seungsoo Kim**[1,3,4], **Hanne Hoskens**[5,6,7], **Harold S. Matthews**[5,6], **Richard A. Spritz** [8], **Ophir D. Klein** [9,10], **Benedikt Hallgrímsson** [7], **Tomek Swigut**[1], **Peter Claes** [5,6,11,12], **Jonathan K. Pritchard** [2,13] ✉ & **Joanna Wysocka** [1,3,4,13] ✉

Transcriptional regulation exhibits extensive robustness, but human genetics indicates sensitivity to transcription factor (TF) dosage. Reconciling such observations requires quantitative studies of TF dosage effects at trait-relevant ranges, largely lacking so far. TFs play central roles in both normal-range and disease-associated variation in craniofacial morphology; we therefore developed an approach to precisely modulate TF levels in human facial progenitor cells and applied it to SOX9, a TF associated with craniofacial variation and disease (Pierre Robin sequence (PRS)). Most SOX9-dependent regulatory elements (REs) are buffered against small decreases in SOX9 dosage, but REs directly and primarily regulated by SOX9 show heightened sensitivity to SOX9 dosage; these RE responses partially predict gene expression responses. Sensitive REs and genes preferentially affect functional chondrogenesis and PRS-like craniofacial shape variation. We propose that such REs and genes underlie the sensitivity of specific phenotypes to TF dosage, while buffering of other genes leads to robust, nonlinear dosage-to-phenotype relationships.

Transcriptional regulation is fundamental to gene expression control, and is mediated by sequence-specific TFs, a class of proteins that modulate target gene expression by binding to specific DNA motifs within noncoding REs; TFs are thus the main drivers of cellular and developmental identity[1]. The stability of organismal development despite environmental and genetic variation[2] suggests that cellular and developmental programs are robust to modest fluctuations in TF levels. *Cis*-regulatory landscapes are often similarly robust, with naturally occurring genetic variation or loss of individual REs often leading to minimal effects on gene expression and/or morphology[3–6].

[1]Department of Chemical and Systems Biology, Stanford University School of Medicine, Stanford, CA, USA. [2]Departments of Genetics and Biology, Stanford University, Stanford, CA, USA. [3]Department of Developmental Biology, Stanford University School of Medicine, Stanford, CA, USA. [4]Howard Hughes Medical Institute, Stanford University School of Medicine, Stanford, CA, USA. [5]Department of Human Genetics, KU Leuven, Leuven, Belgium. [6]Medical Imaging Research Center, University Hospitals Leuven, Leuven, Belgium. [7]Department of Cell Biology & Anatomy, Alberta Children's Hospital Research Institute and McCaig Bone and Joint Institute, Cumming School of Medicine, University of Calgary, Calgary, Alberta, Canada. [8]Human Medical Genetics and Genomics Program and Department of Pediatrics, University of Colorado School of Medicine, Aurora, CO, USA. [9]Departments of Orofacial Sciences and Pediatrics, Program in Craniofacial Biology, and Institute for Human Genetics, University of California, San Francisco, San Francisco, CA, USA. [10]Department of Pediatrics, Cedars-Sinai Medical Center, Los Angeles, CA, USA. [11]Department of Electrical Engineering, ESAT/PSI, KU Leuven, Leuven, Belgium. [12]Murdoch Children's Research Institute, Melbourne, Victoria, Australia. [13]These authors jointly supervised this work: Jonathan K. Pritchard, Joanna Wysocka. ✉e-mail: pritch@stanford.edu; wysocka@stanford.edu

Despite such robustness, human genetic studies have identified widespread phenotypic sensitivity to TF dosage. For instance, TFs are strongly enriched for haploinsufficient disease associations, resulting from the loss of one functional allele, and are depleted of loss-of-function variants in the general population[7,8]. Genome-wide association studies have revealed thousands of trait-associated variants, many of which likely act by modulating RE activity and gene expression levels[9,10]; trait-associated variants are also highly enriched around TF genes[11,12]. Both experimental and population-level data suggest that such common variants show per-allele effects on gene expression of up to 10–15% (refs. [13],[14]). Thus, evidence indicates that RE-driven, relatively minor variation in TF levels leads to complex trait variation, while larger dosage reductions through mechanisms such as haploinsufficiency lead to severe disorders.

Understanding how cellular and developmental programs are simultaneously robust and sensitive to TF levels is a fundamental problem, requiring quantitative studies of endogenous TF dosage effects at physiologically relevant levels. However, most studies of TF function have used knockouts, overexpression beyond trait-relevant dosage ranges, and/or genome-wide assays of unperturbed binding. Such studies have found that TFs typically regulate hundreds to thousands of REs and genes[15–18], and when knocked out during development, produce pleiotropic, often embryonic lethal, phenotypes. Nonlinearity in the effects of TF dosage have been proposed to underlie TF haploinsufficiency[19,20], a concept based on Fisher's 1931 dominance model[21], but such ideas have not been tested experimentally.

Transcriptional regulation is central to the development of the human face, which is key to individual identity and is disrupted in numerous craniofacial disorders that together account for approximately one-third of birth defects[22]. Much of both normal-range and disease-associated variation in facial shape derives from cranial neural crest cells (CNCCs), a transient, embryonic cell population that arises from the neural folds and migrates to the developing facial prominences, giving rise to most of the craniofacial skeleton and connective tissue[23]. Our recent review of human craniofacial genetics found that TF-encoding loci are frequently involved in both common (influencing normal-range shape) and rare (causative for Mendelian, haploinsufficient disorders) variation[24]. Thus, studying the quantitative effects of TF dosage alterations in craniofacial development could provide general insights into mechanisms underlying dosage sensitivity and/or robustness.

Multiple lines of evidence highlight the developmentally important TF SOX9 as an attractive model for studying TF dosage. Heterozygous loss-of-function mutations in *SOX9* cause campomelic dysplasia, a disorder manifesting in long bone and sex determination defects, and a set of craniofacial features termed Pierre Robin sequence (PRS), characterized by underdevelopment of the lower jaw (micrognathia)[25,26]. These observations suggest that among the diverse cell types regulated by SOX9 (reviewed in ref. [27]), CNCCs, chondrocytes and Sertoli cells exhibit heightened sensitivity to about 50% SOX9 dosage reduction. PRS without long bone defects can be caused by heterozygous deletion of CNCC-specific enhancers of *SOX9* (refs. [28],[29]), whereas common genetic variants in noncoding regions near *SOX9* are associated with normal-range facial variation in individuals of primarily European and East Asian ancestry[30–32]. Furthermore, CNCC-specific perturbations in mice revealed that craniofacial development is sensitive to *Sox9* dosage changes over a broad range[29], with even 10–13% reduction in *Sox9* mRNA levels producing a subtle but reproducible change in lower jaw morphology[29].

Here we sought to understand the response to quantitative changes in SOX9 dosage at multiple levels: chromatin, gene expression, cellular phenotypes and facial morphology. We applied the degradation tag (dTAG) system to achieve tunable modulation of SOX9 dosage in an in vitro model of human CNCC development. We found RE chromatin accessibility to be broadly buffered against small to moderate changes in SOX9 dosage, with a subset of REs associated with specific regulatory features showing heightened sensitivity. Gene expression shows a similar, primarily buffered, response to SOX9 dosage, with a subset of sensitive genes; these responses can be partially predicted from chromatin accessibility. Pro-chondrogenic genes, in vitro chondrogenesis itself, and genes and REs associated with PRS-like phenotypes exhibit heightened sensitivity to SOX9 dosage. We propose a model in which dosage-sensitive REs and genes transmit quantitative TF dosage changes to specific cellular and morphological phenotypes, while other phenotypically important REs and genes are regulated by SOX9 but highly buffered and are thus robust to dosage.

## Results

### Precise modulation of SOX9 dosage in hESC-derived CNCCs
On the basis of reports that the dTAG system could be used for rapid or tunable target degradation[33–35], we sought to apply dTAG to modulate SOX9 dosage in human embryonic stem cell (hESC)-derived CNCCs. Our approach involves genome editing in hESCs to tag *SOX9* with FKBP12-F36V, which mediates degradation following addition of a heterobifunctional molecule (dTAG$^V$-1), the fluorescent protein mNeon-Green as a quantitative proxy for SOX9 levels, and the V5 epitope for biochemical assays. Using a selection-free genome editing method[36], we obtained two hESC clones with biallelic knock-in of the FKBP12-F36V–mNeonGreen–V5 tag at the *SOX9* carboxy terminus (Extended Data Fig. 1a).

To avoid indirect effects of depleting SOX9 during hESC-to-CNCC differentiation, we first differentiated *SOX9*-tagged hESCs using an established protocol that yields molecularly nearly homogenous CNCCs[37,38], and subsequently titrated SOX9 levels by adding different dTAG$^V$-1 concentrations (Fig. 1a). Differentiation of *SOX9*-tagged hESCs revealed nuclear fluorescence in a subset of cells within neuroepithelial spheres and in early-stage migratory CNCCs (Fig. 1b), consistent with known roles of SOX9 in CNCC specification and migration[39,40]. Later-stage *SOX9*-tagged CNCCs showed similar SOX9 levels as untagged (wild type (WT)) CNCCs (Fig. 1c), and absolute SOX9 levels between the two *SOX9*-tagged clones were very similar (Extended Data Fig. 1b). Treating *SOX9*-tagged CNCCs with a tenfold dilution series of dTAG$^V$-1 for 24 h yielded a gradual change in SOX9 levels (Fig. 1c). Optimization of dTAG$^V$-1 concentrations and 48-h treatment yielded six distinct and reproducible SOX9 concentrations (Fig. 1d, right). Single-cell fluorescence quantification revealed a unimodal distribution that shifted to lower signals with higher dTAG$^V$-1 concentrations, indicating uniform effects of dTAG$^V$-1 despite some heterogeneity in SOX9 levels (Fig. 1d, left). Together, these results indicate that dTAG can be used to precisely modulate SOX9 dosage in hESC-derived CNCCs.

### Effects of SOX9 dosage changes on RE chromatin accessibility
To assess the effect of SOX9 dosage changes on chromatin accessibility, we carried out the assay for transposase-accessible chromatin with sequencing (ATAC-seq) on *SOX9*-tagged CNCCs with six different SOX9 dosages achieved by varied dTAG$^V$-1 concentrations (Fig. 1d), as well as on WT CNCCs treated with either dimethylsulfoxide or the highest dTAG$^V$-1 concentration (500 nM). Principal component (PC) analysis on ATAC counts per million (CPM) values of the 151,457 reproducible peak regions (which are candidate REs and are herein referred to as REs) revealed a batch-independent dTAG$^V$-1 (and thus SOX9 dosage titration) effect in PC space (Extended Data Fig. 2a).

WT CNCCs treated with 500 nM dTAG$^V$-1 clustered with untreated CNCCs in PC space and had no significantly (5% false discovery rate (FDR)) changed REs, as compared to 6,169 changed REs from two *SOX9*-tagged replicates treated with 500 nM dTAG$^V$-1 (Extended Data Fig. 2b), indicating minimal off-target effects. Plotting each *SOX9*-tagged sample's loading in this PC direction versus SOX9 dosage revealed a nonlinear relationship, indicated by a lower Aikake information criterion (AIC) for a nonlinear Hill equation than a linear

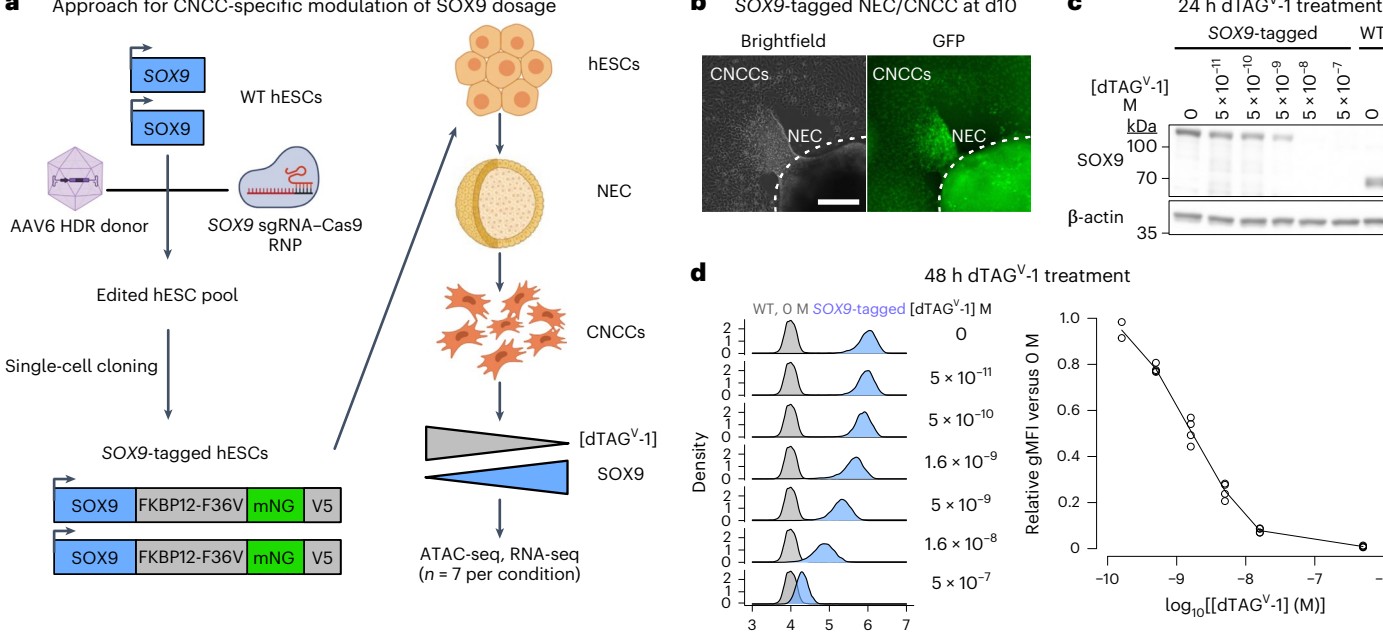

**Fig. 1 | Precise modulation of SOX9 dosage in hESC-derived CNCCs.**
**a**, Schematic of hESC editing and CNCC-specific SOX9 titration approach using dTAG. AAV, adeno-associated virus; HDR, homology-directed repair; RNP, ribonucleoprotein; NECs, neuroectodermal spheres; mNG, mNeonGreen. **b**, Live-cell imaging of mNeonGreen fluorescence in attached NEC and migrating CNCCs derived from *SOX9*-tagged hESCs at the time of CNCC delamination from neuroepithelial spheres (day 10). Images representative of three independent differentiations. Scale bar, 200 μm. **c**, Western blot of SOX9 in *SOX9*-tagged

or WT passaged mesenchymal CNCCs, treated with indicated concentrations of dTAG$^V$-1 for 24 h. Representative of two independent experiments. **d**, Flow cytometry analysis of mNeonGreen fluorescence intensity at 48 h in *SOX9*-tagged CNCCs as a function of increasing dTAG$^V$-1 concentrations across single cells (left, at least 7,000 cells per histogram, representative of two independent experiments for $5 \times 10^{-11}$ M and four independent experiments for all other concentrations) or averaged per biological replicate (differentiation or clone, right). gMFI, geometric mean.

function (Fig. 2a). These results also suggest a largely monotonic effect of SOX9 dosage on individual RE accessibility, which we confirmed by pairwise comparisons between all reduced SOX9 dosages and full dosage (Extended Data Fig. 2c,d).

Individual REs showed distinct responses to SOX9 dosage, with some showing constant decreases in accessibility correlated with SOX9 dosage, and others showing buffered responses (that is, minimal accessibility changes until SOX9 dosage is greatly reduced; Fig. 2b,e), with a similar variety for upregulated REs (Extended Data Fig. 2e). Previous studies have observed similar variation in dose–response curves, either among select targets of the immune TF NF-κB[41] or in cytokine-induced signaling[42]. To model these responses, we first defined all SOX9-dependent REs as those responding significantly (5% FDR) to full depletion of SOX9 over 48 h, using all seven replicates in either condition. This yielded 35,713 REs, of which 20,346 decreased and 15,367 increased in accessibility following SOX9 depletion. Most SOX9-dependent REs were better fitted by the Hill equation than a linear model (median linear minus Hill ΔAIC of 5.2, 73.9% with ΔAIC > 2; Fig. 2f). To allow direct comparisons among REs, we fitted the Hill equation to all SOX9-dependent REs for all subsequent analyses.

The Hill equation yields two key parameters: the empirical dose 50 ($ED_{50}$) representing the SOX9 dosage at which the RE reaches half of its maximal levels, and the Hill exponent, which indicates how switch-like the RE response is (Fig. 2d). In this study, we define sensitivity (or its inverse, buffering) based on the RE response to decreasing SOX9 dosage from 100%. Higher $ED_{50}$ values (at constant Hill exponent) indicate increased sensitivity, while higher Hill exponents (at constant $ED_{50}$) indicate decreased sensitivity. Both values varied between REs (Fig. 2g), but the $ED_{50}$ was substantially more correlated with an alternative measure of sensitivity than the Hill exponent (Extended Data

Fig. 3; Spearman $\rho$ of −0.96 and −0.45 for $ED_{50}$ and Hill, respectively), indicating that it is the main determinant of sensitivity/buffering. Of all SOX9-dependent REs, 26,026 (73%) have $ED_{50} < 30$ (buffered), 5,276 (14%) of REs have $ED_{50}$ between 30 and 40 (moderately sensitive), and 4,411 (13%) of REs have $ED_{50} > 40$ (highly sensitive; Supplementary Table 1). The proportion of downregulated or upregulated REs in each of these groups is consistent (about 68% downregulated versus about 32% upregulated). Together, these results indicate a range in RE responses to SOX9 dosage, with most SOX9-dependent REs buffered against changes in SOX9 dosage but some showing more sensitive responses.

**Features affecting RE sensitivity to SOX9 dosage**
We next sought to identify regulatory features associated with RE sensitivity to SOX9 dosage. For the remainder of this paper, we use a bootstrapping approach when comparing $ED_{50}$ values between groups of REs/genes to incorporate fitting uncertainty ($n = 200$ bootstraps; Methods). We reasoned that the SOX9-dependent REs comprised a mix of direct effects of SOX9 regulation and indirect effects acting through other TFs. Direct SOX9 effects should arise rapidly after full SOX9 depletion whereas indirect effects should be delayed. We therefore carried out ATAC-seq 3 h after 500 nM dTAG$^V$-1 treatment of *SOX9*-tagged CNCCs (yielding full SOX9 depletion within 1 h; Extended Data Fig. 4a). Of the 35,713 48-h SOX9-dependent REs, 9,279 showed significant (5% FDR) accessibility changes at 3 h, of which almost all (96.3%) were decreases (Fig. 3a). Relative to delayed and/or upregulated REs, rapidly downregulated REs were substantially more likely to harbor the SOX9 palindrome sequence motif (Fig. 3b), as well as SOX9 binding as assessed by V5 chromatin immunoprecipitation and sequencing (ChIP–seq; Extended Data Fig. 4b). These results are consistent with

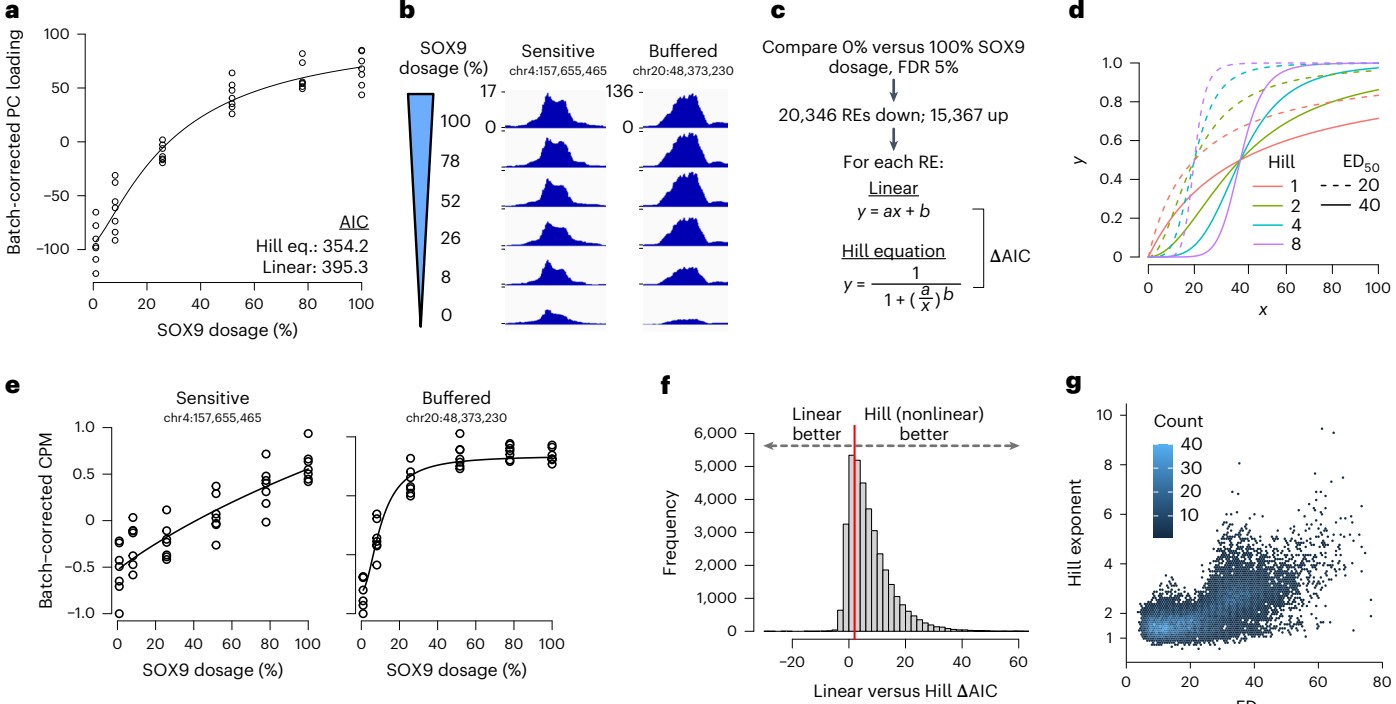

**Fig. 2 | Most SOX9-dependent REs are buffered in their response to SOX9 dosage changes, with a sensitive subset. a**, Loadings from PC analysis of ATAC-seq CPM of all 151,457 REs across all CNCC samples (see Extended Data Fig. 2a), corrected for differentiation batch and plotted as a function of estimated relative SOX9 dosage (shown as percentage relative to no dTAG$^V$-1). Black line represents Hill equation fit. Points are biological (differentiation or clone) replicates. **b**, Example ATAC-seq browser tracks of individual RE accessibility at different SOX9 dosages (y axis, normalized coverage in 10-bp bins, identical range in all browser tracks), averaged across all replicates at each dosage. **c**, Schematic of approach to model nonlinearity of SOX9-dependent REs. $a$ and $b$ in linear model refer to slope and intercept, respectively. $a$ and $b$ in Hill equation refer to $ED_{50}$ and Hill exponent, respectively. **d**, Illustration of different $ED_{50}$ and Hill exponent values on theoretical Hill equation curves. **e**, Individual REs from **b** with replicates, fitted by Hill equation (black line). **f**, Histogram of ΔAIC of all 35,713 SOX9-dependent REs. Red line indicates ΔAIC = 2. **g**, Scatterplot of $ED_{50}$ and Hill exponent of 23,414 SOX9-dependent REs with good fit ($P < 0.05$ for both parameters).

SOX9 acting as a direct activator of REs at rapidly downregulated sites ($n = 9,279$) and indirectly regulating delayed and/or upregulated sites ($n = 26,434$). Compared to delayed and/or upregulated REs, rapidly downregulated sites were substantially more sensitive to SOX9 dosage (that is, had higher $ED_{50}$ values; Fig. 3c,d).

While direct versus indirect regulation is one contributor to RE sensitivity to SOX9 dosage, there is additional variation in sensitivity among the 9,279 direct SOX9 target sites: 4,266 (46%) have $ED_{50} < 30$ (buffered), 2,760 (30%) have $ED_{50}$ between 30 and 40 (moderately sensitive), and 2,253 (24%) have $ED_{50} > 40$ (highly sensitive). We therefore sought to identify additional features associated with variation in sensitivity among all direct SOX9 target sites. REs containing the full SOX9 palindrome with spacing of 3–5 base pairs (bp) were more sensitive than sites containing either one or several partial palindromes, with REs containing no detected motif being the least sensitive (Fig. 3e). The 3–5-bp SOX9 palindrome was also associated with a modest increase in the Hill exponent, consistent with the palindrome's reported requirement for cooperative SOX9 binding[43] (Extended Data Fig. 4c,d). Thus, motif type and resultant SOX9 binding mode modulate RE sensitivity to SOX9 dosage among its direct targets. Among direct SOX9 targets, those with larger effects of SOX9 depletion were most sensitive to SOX9 dosage (Extended Data Fig. 5a).

We assessed whether additional factors beyond SOX9 motif type could modulate RE sensitivity to SOX9 dosage among its direct targets. We focused on binding by other TFs, specifically TWIST1, NR2F1 and TFAP2A, as they have well-known roles in CNCCs and their binding in hESC-derived CNCCs has previously been characterized[29,38]. Binding of other TFs at SOX9 direct target sites substantially decreased RE sensitivity to SOX9 dosage; the strongest effects were seen for TWIST1 and

TFAP2A, with minor effects of NR2F1 at TWIST1- and TFAP2A-bound REs (Fig. 3f). We replicated this result using TF sequence motifs (Extended Data Fig. 5b). Baseline levels of both the active histone mark H3K27ac and chromatin accessibility were negatively correlated with sensitivity (Extended Data Fig. 5c,d). REs containing the SOX9 palindrome motif also unbound by other TFs were most sensitive to SOX9 dosage (Fig. 3g and Extended Data Fig. 5e). Together, these results indicate that at least three features independently contribute to variation in RE sensitivity to SOX9 dosage: direct versus indirect regulation by SOX9; among directly regulated SOX9 targets, the type of SOX9 motif and resulting binding mode; and binding of other key CNCC TFs. Thus, REs for which SOX9 is likely the primary TF directly driving accessibility are most sensitive to SOX9 dosage.

To understand mechanisms underlying buffering against changes in SOX9 dosage among its direct targets, we carried out ChIP–seq of SOX9 (using the V5 tag) and TWIST1 in *SOX9*-tagged CNCCs with four different SOX9 concentrations achieved using dTAG$^V$-1. We grouped direct SOX9 targets into three bins based on ATAC-seq sensitivity to SOX9 dosage and plotted their ChIP–seq fold changes at each SOX9 concentration versus 100%. REs most sensitive to SOX9 dosage in accessibility are enriched for the SOX9 palindrome motif and have a lower fraction and level of TWIST1 binding (Extended Data Fig. 6a,c). These REs were also most sensitive in SOX9 binding, whereas the accessibility-buffered REs were also buffered in SOX9 binding. At about 50% SOX9 dosage, buffered sites retained nearly unperturbed levels of SOX9 binding (Extended Data Fig. 6b). TWIST1 binding showed similar responses to SOX9 dosage (Extended Data Fig. 6d), with no increase at partial SOX9 dosage, as might have been expected by compensatory buffering. Notably, stronger SOX9 perturbation (≤20%) resulted

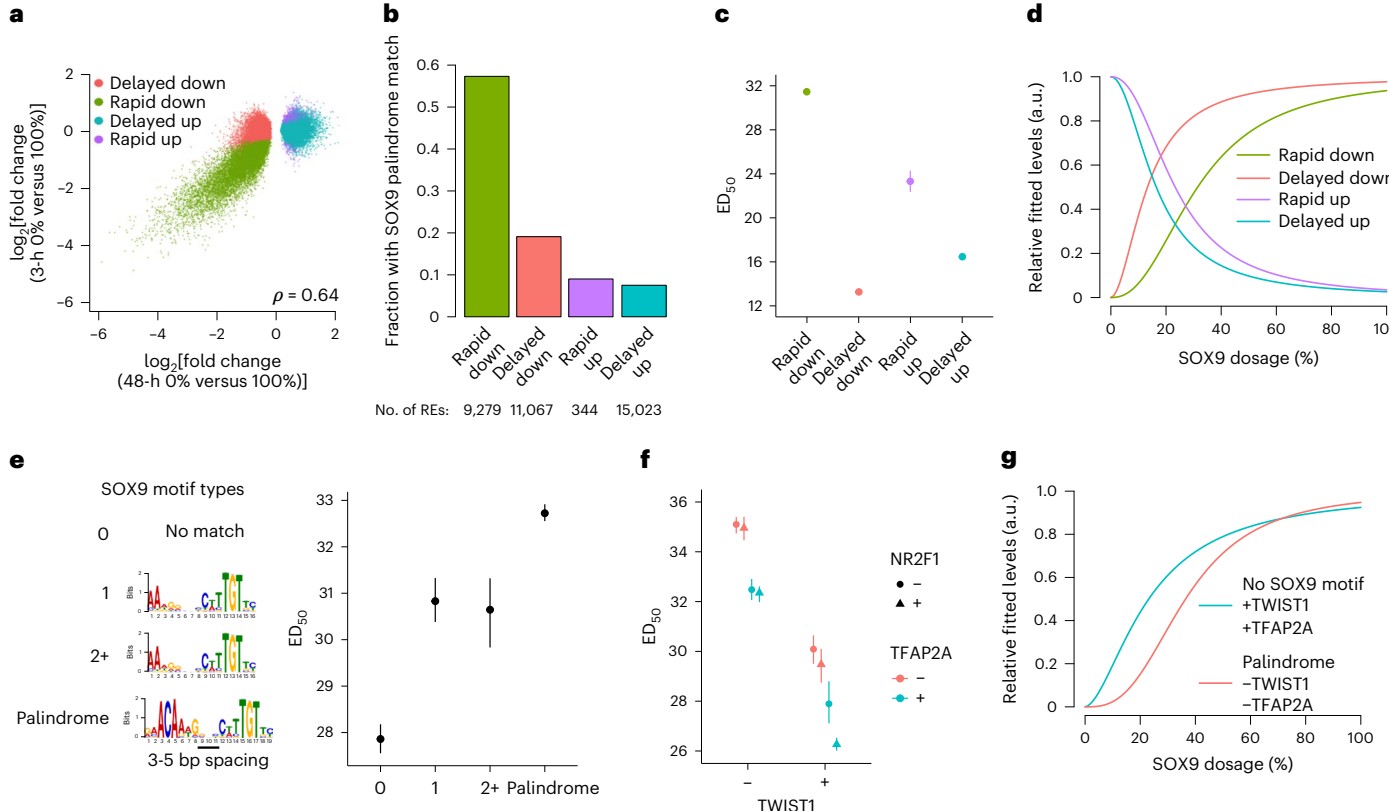

**Fig. 3 | Features affecting sensitivity of the RE response to SOX9 dosage.**
**a**, Scatterplots of full SOX9 depletion effect on chromatin accessibility at 48 h (x axis) versus 3 h (y axis) for all 48-h SOX9-dependent REs. **b**, Barplot indicating fraction of REs containing a SOX9 palindrome (3–5-bp spacing), stratified by time delay of response. **c**, $ED_{50}$ of REs stratified by time delay of response. **d**, Median dosage curves based on the median fitted $ED_{50}$ and Hill exponents for each group. a.u., arbitrary units. N in **c**,**d** indicated in **b**. **e**,**f**, $ED_{50}$ of rapid-down REs (likely

direct SOX9 targets) stratified by SOX9 motif type, with motif position weight matrices on the left (**e**), or overlap with ChIP–seq peaks for TWIST1 (x axis) TFAP2A (color) and NR2F1 (shape) (**f**). n of groups from left to right in **e**: 2,263, 1,315, 360, 5,221. n of groups from left to right in **f**: 1,874, 736, 985, 1,805, 774, 602, 296, 2,087. **g**, Median dosage curves as in **d** for the indicated combinations of SOX9 motif and binding of other TFs. Points and error bars in (**c**,**e**,**f**) represent median and 95% confidence intervals as computed by 200 bootstraps (see Methods).

in diminished TWIST1 binding at both buffered and sensitive sites (Extended Data Fig. 6d), consistent with a model in which SOX9 and TWIST1 binding is co-dependent at a subset of direct SOX9 target sites, but their synergistic function buffers co-regulated REs against small TF dosage changes (Extended Data Fig. 6e).

### Effects of SOX9 dosage on gene expression
We next assessed the gene expression response to SOX9 dosage by RNA sequencing (RNA-seq) analysis of the same SOX9 dosage series. This response was largely monotonic (Extended Data Fig. 7a) and varied in its shape between individual genes (Fig. 4a and Extended Data Fig. 7b). SOX9 dosage changes had overall nonlinear effects on gene expression in PC space (Extended Data Fig. 7c), and most (70.3%) of the 1,232 SOX9-dependent genes (of which 688 decreased and 544 increased following full depletion, 5% FDR) were better fitted by a Hill than a linear equation (Fig. 4b and Supplementary Table 2), with variability in the $ED_{50}$ and Hill exponents (Fig. 4c). Of all SOX9-dependent genes, 76% have $ED_{50} < 30$ (buffered), 12% have $ED_{50}$ between 30 and 40 (moderately sensitive), and 12% have $ED_{50} > 40$ (highly sensitive; Supplementary Table 2). Thus, like REs, most genes are buffered against SOX9 dosage, and a minor subset is sensitive.

We investigated whether RE responses can predict the responses of their cognate target genes. We focused on the subset of SOX9-dependent genes showing changes in nascent transcription in response to 3-h or 24-h SOX9 depletion, assessed by thiol (SH)-linked alkylation for the metabolic sequencing of RNA (SLAM-seq)[44]. Of the

1,232 48-h SOX9-dependent genes, 122 (62 down, 60 up) responded significantly (10% FDR) at 3 h, and 395 (206 down, 189 up) responded at 24 h (Supplementary Table 2). Effect sizes at 24 h were correlated with, albeit larger than, those at 3 h (Extended Data Fig. 8a). Known direct SOX9 targets such as *COL2A1* (ref. [45]) responded at 24 h but not 3 h (Extended Data Fig. 8b), suggesting a time lag between chromatin and transcriptional effects. Accordingly, we sought to predict the gene expression responses for genes responding transcriptionally to 24 h of SOX9 depletion.

The activity-by-contact (ABC) model predicts RE–gene connections by computing the contribution of each RE to transcription (ABC score) as its 'activity' (combination of accessibility and H3K27ac) divided by its contact (estimated by chromatin conformation capture or a genomic distance–power law function), normalized to the contributions of other REs[46]. We used ABC to predict the effect of multiple REs changing in 'activity' at each SOX9 dosage. Although 'activity' includes H3K27ac levels, effects of full SOX9 depletion on accessibility and H3K27ac were highly correlated (Extended Data Fig. 8c,d). Thus, the fold change in the expression of a gene is predicted as the average fold change in accessibility at all nearby REs, weighted by the contribution of each RE (ABC score; Fig. 4d).

We first assessed prediction of directionality of expression changes, comparing observed and predicted responses for transcriptionally upregulated or downregulated genes as well as SOX9-independent genes. Predicted responses significantly stratified these genes in the same manner as observed responses, most accurately

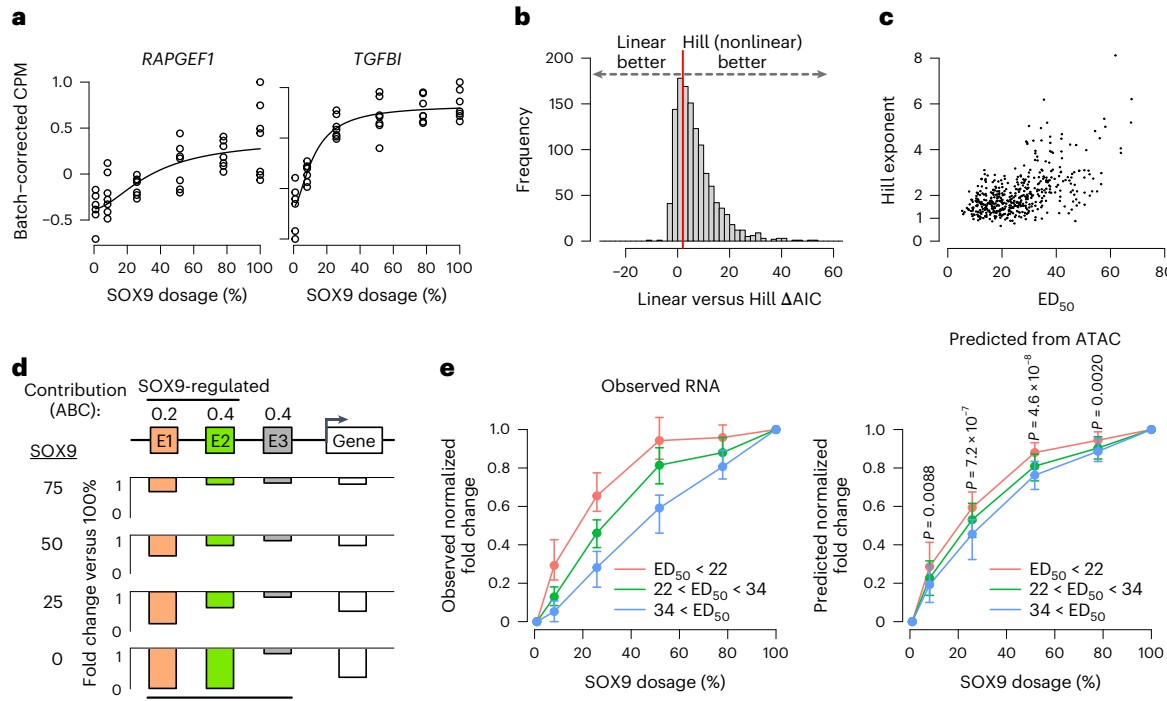

**Fig. 4 | RE dose–response curves partially predict the shape of gene dose–response curves. a**, Examples of genes with sensitive (left) or buffered (right) responses to SOX9 dosage changes, as assessed by RNA-seq. Black lines represent Hill equation fits. **b**, Histogram of ΔAIC of all 1,232 SOX9-dependent genes, calculated as in Fig. 2c. Red line indicates ΔAIC = 2. **c**, $ED_{50}$ and Hill exponent of 832 SOX9-dependent genes with good fit ($P < 0.05$ for both parameters). **d**, Schematic of approach to predict RNA level changes based on ABC contribution scores and fold changes of REs at each SOX9 dosage. **e**, Distributions of observed (left) or predicted (right) fold changes versus full SOX9 dosage at each concentration, normalized to full-depletion fold change and stratified by $ED_{50}$ (colors). Only genes transcriptionally downregulated by 24-h full SOX9 depletion (assessed by metabolic labeling, SLAM-seq) are analyzed. $n$ of groups by color: red, 66; green, 77; blue, 57. Points and error bars represent median and 25th and 75th percentiles of distribution. Indicated $P$ values are from two-sided Kruskal–Wallis test comparing distributions of the three colored groups.

for downregulated genes (Extended Data Fig. 8e). We thus focused on predicting differences in SOX9 dosage sensitivity among downregulated genes. Predicted responses separated SOX9-downregulated genes binned by their observed sensitivity to SOX9 (Fig. 4e), although to a lesser extent than the observed responses. Inspection of top genes indicated accurate prediction of both buffered (*TENT5B*) or sensitive (*SOX5*) responses (Extended Data Fig. 8f). Genes with less accurate predictions have fewer SOX9-dependent REs in their vicinity (Extended Data Fig. 8g), suggesting that the model does not fully capture complex *cis*-regulatory landscapes with multiple inputs. Nevertheless, genes with more sensitive nearby REs were more sensitive to SOX9 dosage (Extended Data Fig. 8h). Together, these results indicate that broadly, the RE response to SOX9 dosage (sensitive or buffered) translates into the expression response of cognate genes based on the contribution of the RE to the transcription of that gene.

**The pro-chondrogenic program is sensitized to SOX9 dosage**
We next sought to assess the impact of SOX9-sensitive genes on cellular phenotypes, focusing on chondrogenic differentiation potential as SOX9 functions in both entry into and continuation of chondrogenesis[47]. Genes with both cartilage development functions and increased expression during in vitro chondrogenesis ('pro-chondrogenic genes') showed substantially higher $ED_{50}$ values than other gene groups (Fig. 5a). Examples include the collagen-encoding genes *COL11A1* (highly sensitive) and *COL2A1* (moderately sensitive; Fig. 5b), as well as genes encoding other transcriptional regulators such as *SOX5* (Extended Data Fig. 8f). SOX9-upregulated genes did not yield a similar pattern of sensitivity (Extended Data Fig. 9a), suggesting that pro-chondrogenic functions of SOX9 may be especially dosage sensitive. Gene Ontology analysis of all moderately and

highly sensitive ($ED_{50} > 30$) genes revealed enrichment for cartilage condensation function as well as additional pathways with important roles in CNCCs, such as transforming growth factor beta and bone morphogenetic protein, but also neuronal/glial-related pathways unlikely to have important functions in mesenchymal CNCCs (Supplementary Table 3).

To test whether increased sensitivity of pro-chondrogenic genes results in increased sensitivity of chondrogenesis, we titrated SOX9 dosage to five distinct levels both before and during 21-day differentiation of CNCCs to chondrocytes[29] (Fig. 5c and Extended Data Fig. 9b). To quantify functional chondrogenesis, we measured total levels of sulfated glycosaminoglycans (sGAGs), linear polysaccharides that mark extracellular matrix of mature cartilage, using a colorimetric assay. This revealed a nonlinear relationship between SOX9 dosage and functional chondrogenesis (Fig. 5d), with no effect of the highest dTAG[V]-1 concentration (500 nM) on WT CNCC chondrogenesis, indicating minimal off-target effects (Extended Data Fig. 9c). The SOX9 dosage–sGAG curve more closely matched the curve for pro-chondrogenic genes than for other genes or REs (Fig. 5e and Extended Data Fig. 9d). Thus, in vitro chondrogenesis is sensitized to SOX9 dosage, more so than most genes or REs, at least partly owing to the heightened sensitivity of important pro-chondrogenic genes.

**Genes and REs associated with PRS-like phenotypes are sensitized to SOX9 dosage**
We assessed the impact of SOX9-sensitive genes and REs on human morphological and craniofacial disease phenotypes. SOX9-dependent genes associated with dominant (likely dosage-sensitive) craniofacial disorders phenotypically unrelated to PRS had lower $ED_{50}$ values than genes not associated with craniofacial disorders, while genes associated

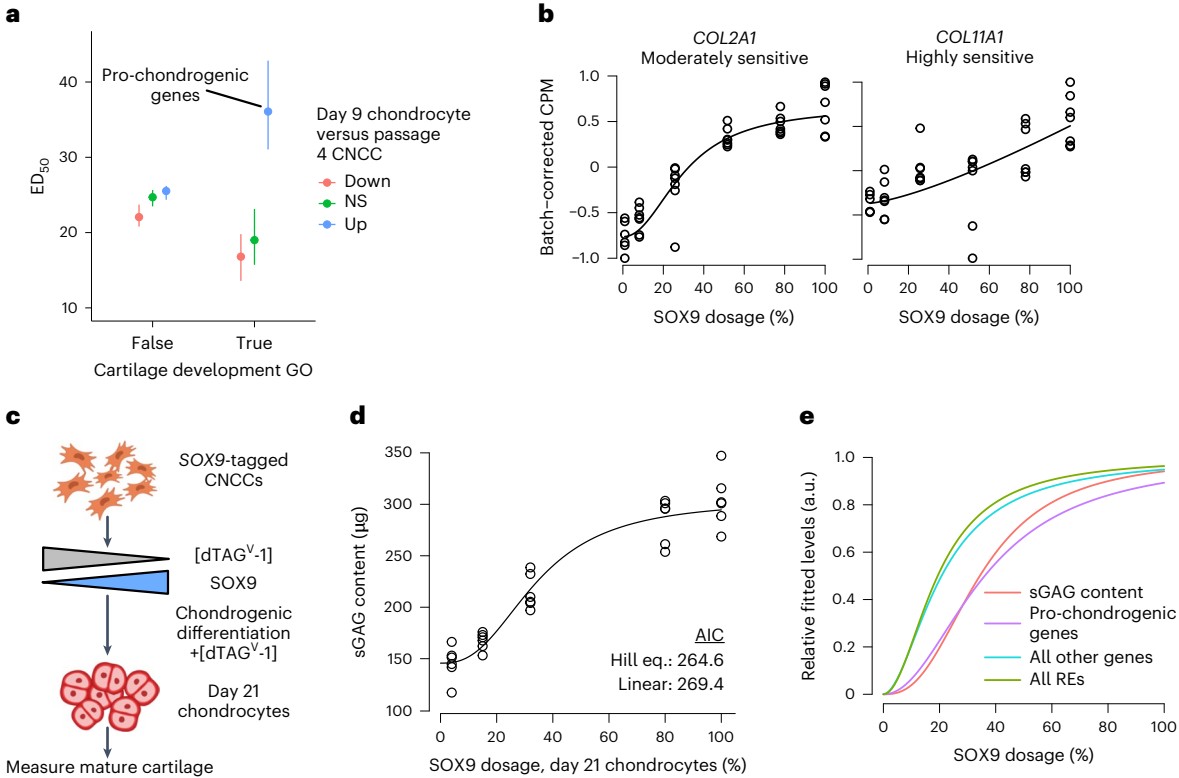

**Fig. 5 | The pro-chondrogenic program is sensitive to SOX9 dosage. a**, $ED_{50}$ of SOX9-downregulated genes stratified by presence in the 'Cartilage development' Gene Ontology (GO) category (x axis), and expression change in chondrocytes compared to CNCCs (color, data from ref. [29]). *n* of groups from left to right: 157, 269, 241, 6, 4, 11. Points and error bars represent median and 95% confidence intervals as computed by 200 bootstraps (see Methods). **b**, Examples of known pro-chondrogenic genes that are moderately or highly sensitive to SOX9 dosage. **c**, Schematic of approach to titrate SOX9 dosage during 21-day chondrogenic differentiation. **d**, Sulfated glycosaminoglycan (sGAG, representative of mature cartilage) content at day 21 of chondrogenesis as a function of SOX9 dosage as estimated in Extended Data Fig. 9b. Black curve represents Hill equation fit. **e**, Median dosage curves based on fitted $ED_{50}$ and Hill exponents for all REs and genes, pro-chondrogenic genes (purple, labeled group in **a**) and sGAG content (from **d**).

with recessive (likely not dosage-sensitive) disorders had higher $ED_{50}$ values (Fig. 6a,b); this suggests buffering of important, dosage-sensitive genes that strongly impact craniofacial development. However, genes associated with PRS-like craniofacial defects[48] were most sensitive to SOX9 dosage (Fig. 6a,b). These include the pro-chondrogenic genes *COL2A1* and *COL11A1*, haploinsufficiency of which is associated with Stickler syndrome[49,50], which like PRS, includes lower jaw hypoplasia. Similar results were not observed with SOX9-upregulated genes (Extended Data Fig. 9e). Thus, while dosage-sensitive, SOX9-dependent craniofacial genes are generally buffered against SOX9 dosage, those with PRS-like phenotypes and pro-chondrogenic roles are highly sensitive and may mediate the phenotypic specificity of SOX9 dosage perturbation during craniofacial development.

We next assessed whether similar principles of selective sensitivity to SOX9 dosage apply to normal-range variation in facial shape. We applied multivariate phenotyping approaches[30] to three-dimensional facial scans from 8,246 healthy individuals and 13 patients with PRS, identifying single nucleotide polymorphisms (SNPs) associated with normal-range variation in healthy individuals along the axis from typical to PRS (PRS endophenotype; Fig. 6c and Extended Data Fig. 10). This genome-wide association study (GWAS) revealed 2 independent signals near *SOX9* (Extended Data Fig. 10); 20 additional loci across the genome reached genome-wide significance ($P < 5 \times 10^{-8}$; Supplementary Table 4, Fig. 6d) and highlighted genes such as *SFRP2*, of which loss of function causes defects in chondrogenesis, and *DLX5*/*DLX6*, required for lower jaw identity. Thus, variation along the healthy-to-PRS axis is modulated by variants near *SOX9*, as expected given associations between *SOX9* mutations and PRS itself, but is also polygenic.

The 20 genome-wide significant loci were a subset of previously identified loci affecting normal-range facial variation; we thus segregated previously reported facial GWAS lead SNPs[24] on the basis of association with the PRS endophenotype (Bonferroni-corrected $P < 0.05$). SOX9-dependent REs in linkage disequilibrium ($r^2 > 0.5$) with signals for PRS-unrelated facial phenotypes had slightly lower $ED_{50}$ values than other SOX9-dependent REs. By contrast, REs in linkage disequilibrium with PRS endophenotype signals had higher $ED_{50}$ values (Fig. 6e). Combined with the analyses of gene–disorder associations, these results indicate that REs and genes with corresponding phenotypes distinct from those caused by SOX9 dosage changes are generally buffered against changes in TF dosage, even if they ultimately are SOX9 dependent, while REs and genes associated with phenotypes similar to those caused by SOX9 dosage changes are most sensitive.

## Discussion

Here we have quantified the relationship between TF dosage and phenotype at molecular, cellular and morphological levels, using SOX9 as a model. To synthesize our observations, we propose a model (Fig. 7a) in which REs regulated by SOX9 range from sensitive to buffered as a result of their *cis*-encoded features that determine the mode and level of binding by SOX9 and other key CNCC TFs. Genes with nearby sensitive REs will themselves show more sensitive responses to SOX9 dosage, while those with nearby buffered REs are more robust. Genes with generally important roles in CNCC biology but causing phenotypes distinct from those associated with SOX9 are buffered against SOX9 dosage change, but a subset of sensitive genes impacts specific cellular processes and morphological features similar to those associated with

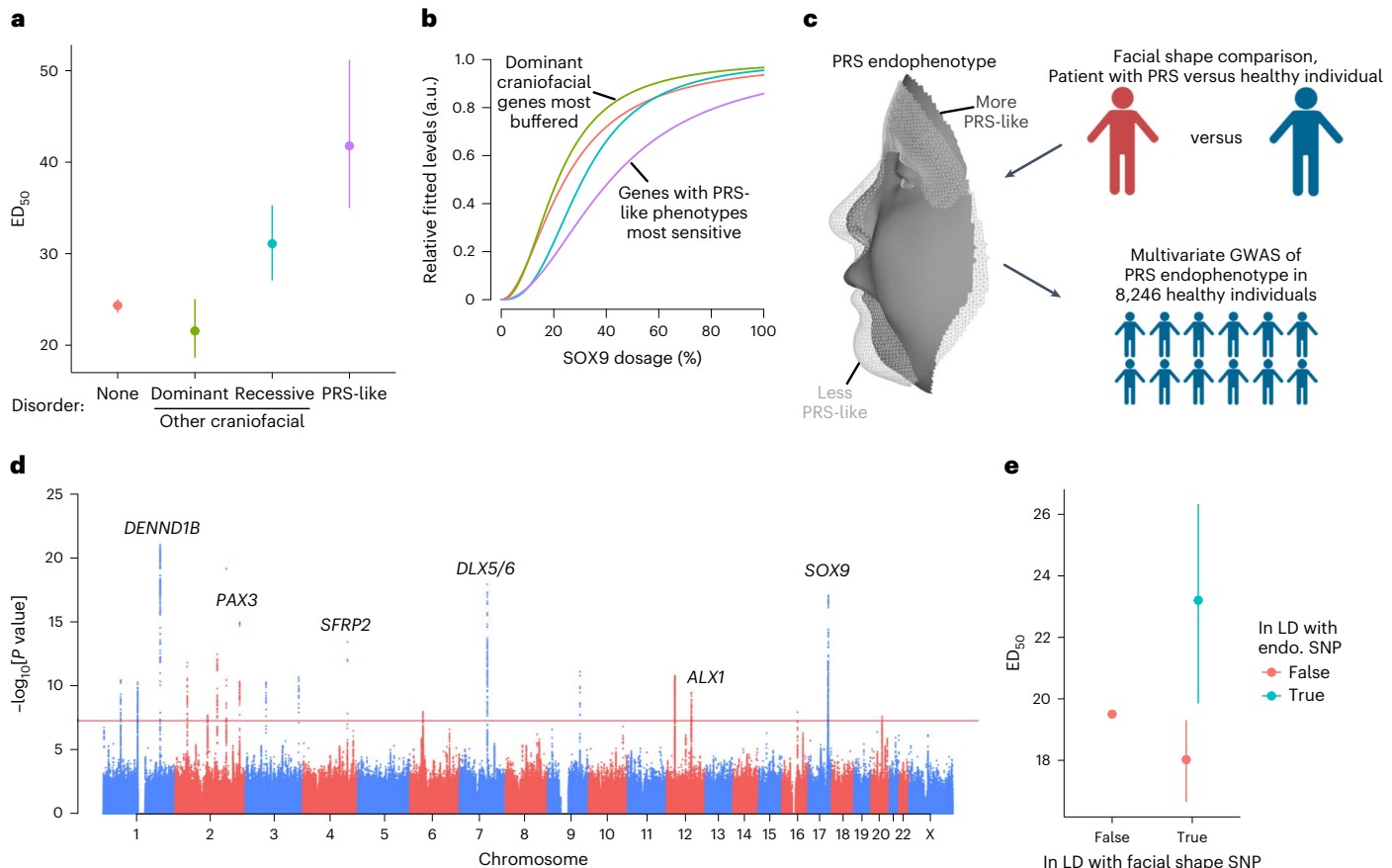

**Fig. 6 | Genes and REs associated with PRS-like phenotypes are uniquely sensitized to SOX9 dosage. a**, ED$_{50}$ of SOX9-downregulated genes stratified by craniofacial disorder association. PRS-like associations as defined by Perrine et al. 2020 (ref. [48]). $n$ of groups from left to right: 665, 13, 6, 8. **b**, Median dosage curves (based on both ED$_{50}$ and Hill exponent) of the same groups from **a**. **c**, Schematic of approach to conduct a multivariate GWAS on a PRS-defined endophenotype in healthy human individuals. **d**, Manhattan plot of PRS endophenotype GWAS. Red line indicates genome-wide significance. Candidate genes near top GWAS signals are labeled. **e**, ED$_{50}$ of REs stratified by linkage disequilibrium (LD, $r^2 > 0.5$) with any facial shape GWAS lead SNP ($x$ axis, as defined in Naqvi, Hoskens et al. 2022 (ref. [24]) and further with any lead SNP associated with the PRS endophenotype GWAS from **d** (color). $n$ of groups from left to right: 35,450, 209, 54. Points and error bars in **a**,**e** represent median and 95% confidence intervals as computed by 200 bootstraps (see Methods).

SOX9. Specifically, we find that several key pro-chondrogenic genes and in vitro chondrogenesis are sensitized to SOX9 dosage. Thus, the observed sensitivity of both chondrogenic effector genes and chondrogenesis itself could account for the specificity of SOX9-associated, PRS-like mandibular phenotypes, perhaps via effects on Meckel's cartilage, a cartilage 'template' involved in mandible formation[51].

Our model can explain distinct phenotypes observed across the range of SOX9 dosage (Fig. 7b). The ≈75–100% dosage regime yields subtle effects on SOX9-sensitive genes affecting chondrogenesis and mandibular development, driving normal-range variation along the healthy–PRS axis. At dosages closer to 50%, further decreased activity of dosage-sensitive effectors (and potentially effects from additional genes) exacerbates the phenotypic effects, resulting in a specific disease (PRS). Finally, lower SOX9 dosages (about 25% or less) lead to broad dysregulation of other craniofacial developmental pathways, which, combined with greater perturbations to dosage-sensitive effectors, result in wide phenotypic impacts and embryonic lethality[52].

Core concepts of our model may generalize. Haploinsufficiency of other craniofacial TFs often causes syndromes comprising characteristic facial features (for example, PAX3 in Waardenburg, TWIST1 in Saethre–Chotzen, TFAP2A in branchiooculofacial syndromes), but similar to SOX9, these TFs bind to and presumably regulate thousands of REs (and perhaps hundreds of genes). Effector REs/genes uniquely sensitive to dosage of each TF would result in phenotypic specificity at about 50% TF dosage while allowing for broad regulatory

programs. A study of *TBX5*, encoding a cardiac TF, found that a subset of genes dysregulated by homozygous *TBX5* deletion showed consistent but milder changes following heterozygous deletion; some of these genes may represent dosage-sensitive effectors[53]. A study using doxycycline-induced expression found level-dependent effects of SOX2 during caudal epiblast development[54]. Finally, genetic manipulation of expression of the *Drosophila* gene encoding the TF bicoid found classes of concentration-sensitive and concentration-insensitive targets[55].

Our model allows for both robustness and phenotypic sensitivity to TF dosage. Robustness can be explained by nonlinear relationships between gene dosage and phenotype suggested by human[56,57] and mouse[58] genetics. Our model suggests that these relationships may be a composite of distinct molecular responses: most SOX9 targets are buffered against moderate changes in SOX9 dosage, while trait variation and disease is primarily driven by the SOX9-sensitive effectors. Buffered targets can explain robustness to TF dosage perturbation, while sensitive effectors likely mediate phenotypic specificity associated with TF dosage changes.

How TFs modulate highly polygenic variation in complex trait and disease risk is not known[59]. One possibility is that downstream effects of a trait-associated TF are distributed among its many targets. While SOX9, like many TFs, regulates thousands of REs/genes, our study indicates that most of these targets are buffered and have individually tiny effects at the <50% variation in TF dosage observed in GWAS, such that effects with individually appreciable contributions to variation result

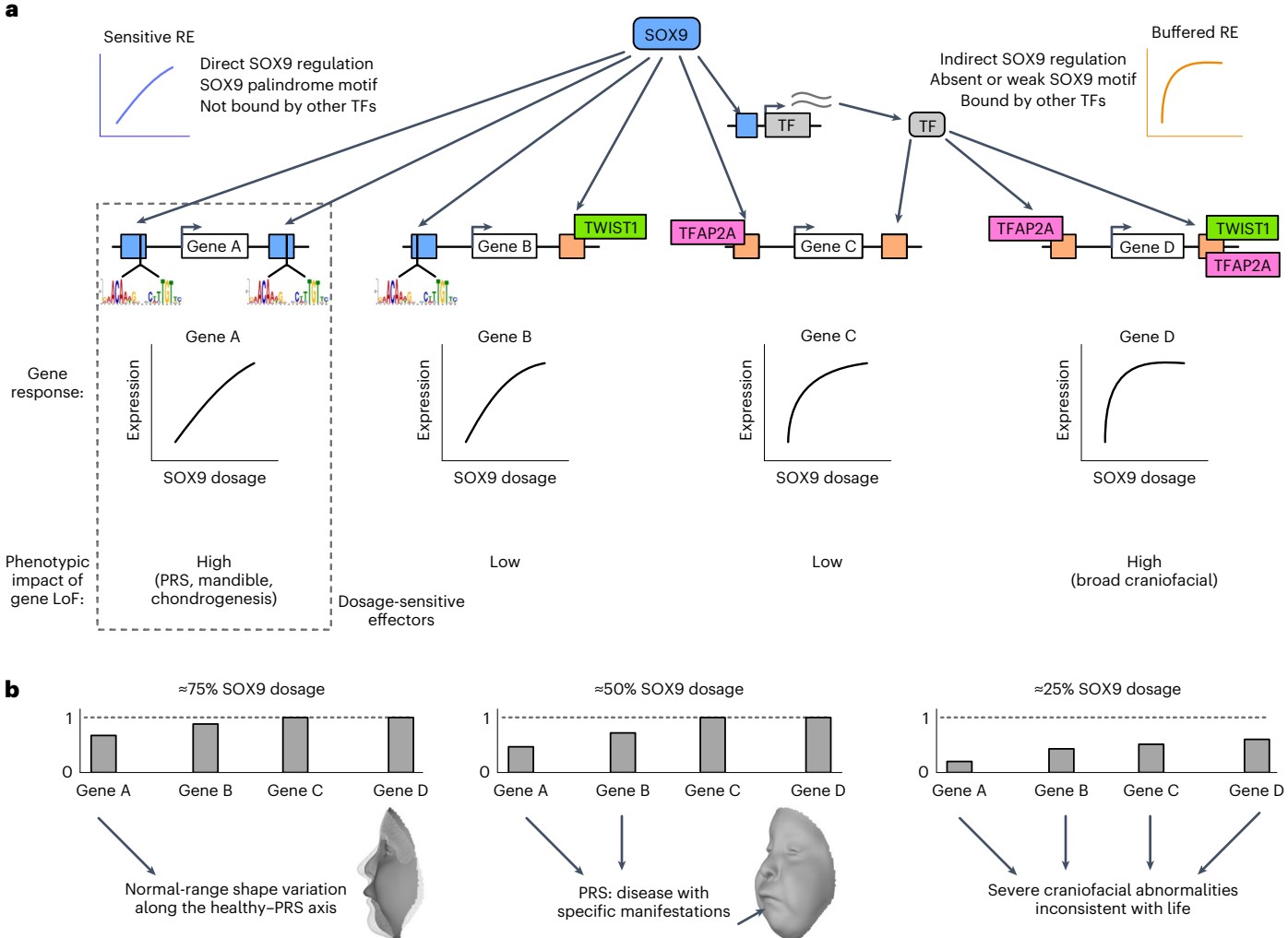

**Fig. 7 | Dosage-sensitive effectors transmit the effect of quantitative changes in SOX9 dosage to provide phenotypic specificity. a**, Schematic indicating which features make REs sensitive (blue) or buffered (orange) to SOX9 dosage. Dosage-sensitive effectors are REs and genes mediating changes in cellular behaviors as a result of quantitative changes to their activity or expression, and thus with high phenotypic impact (dashed rectangle), as compared to sensitive genes with low phenotypic impact (gene B) or phenotypically impactful genes that are buffered (gene D). LoF, loss of function. **b**, Schematic of gene expression changes in response to the indicated SOX9 dosages based on sensitivities from **a**. Arrows indicate the contribution of the gene to phenotype.

from a subset of SOX9-sensitive targets that impact chondrogenesis and PRS-like phenotypes. Such effector genes are conceptually similar to core genes that act directly on a trait, recently proposed in the omnigenic model[60,61].

## Online content

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

## Methods

### Inclusion and ethics

Collection of data from patients with PRS was carried out with overall approval and oversight of the Colorado Multiple Institutional Review Board (IRB no. 09–0731), was additionally approved by the IRBs of the University of Calgary, Florida State University, the University of California San Francisco and the Catholic University of Health and Allied Sciences (Mwanza, Tanzania), and was carried out with the approval of the National Institute for Medical Research (Tanzania). Written informed consent was obtained from all study participants or their parents, as appropriate. No participants received compensation.

The PRS endophenotype GWAS in this study was conducted on individuals of European ancestry. The conclusions of this GWAS therefore may not be applicable to individuals of other, diverse ancestries. For the PRS endophenotype GWAS conducted in healthy individuals, ethical approval was obtained at each recruitment site and all participants gave their written informed consent before participation. For individuals under 18 years of age, written consent was obtained from a parent or legal guardian. For the US sample, the following local ethics approvals were obtained: Pittsburgh, PA (PITT IRB no. PRO09060553 and no. RB0405013); Seattle, WA (Seattle Children's IRB no. 12107); Houston, TX (UT Health Committee for the Protection of Human Subjects no. HSC-DB-09-0508); Iowa City, IA (University of Iowa Human Subjects Office IRB no. 200912764 and no. 200710721); Urbana-Champaign, IL (PSU IRB no. 13103); New York, NY (PSU IRB no. 45727); Cincinnati, OH (UC IRB no. 2015–3073); Twinsburg, OH (PSU IRB no. 2503); State College, PA (PSU IRB no. 44929 and no. 4320); Austin, TX (PSU IRB no. 44929); San Antonio, TX (PSU IRB no. 1278); Indianapolis, IN and Twinsburg, OH (IUPUI IRB no. 1409306349). For the UK sample, ethical approval for the study (Project B2261: 'Exploring distinctive facial features and their association with known candidate variants') was obtained from the ALSPAC Ethics and Law Committee and the Local Research Ethics Committees. Informed consent for the use of data collected through questionnaires and clinics was obtained from participants following the recommendations of the ALSPAC Ethics and Law Committee at the time. Consent for biological samples has been collected in accordance with the Human Tissue Act (2004).

The use of hESCs in this study was approved by the Stanford Stem Cell Research Oversight and IRB committees under protocol no. SCRO-510. The H9 hESC line used was obtained commercially (WiCell) and was therefore derived under informed consent.

### Statistics and reproducibility

No statistical method was used to predetermine sample size. The experiments were not randomized. The investigators were not blinded to allocation during experiments and outcome assessment. Facial scans were excluded if participants were laughing, crying or otherwise emoting or if the non-rigid registration failed. Facial scans from participants with missing covariate information (for example, age or sex) were additionally removed. Two RNA-seq samples (WT_R8_5e-7M and S9CC47_R6_5e-7M) were identified as extreme outliers in initial PC analysis. This was confirmed to be due to a library quality issue, and so these samples were excluded before any further downstream analyses. For modeling of individual REs/genes as a function of SOX9 dosage, outlier samples, defined as $z$-score greater than 3, were removed from the analysis of that RE/gene.

### Cell culture

Female H9 (WA09; RRID: CVCL_9773) hESCs were obtained from WiCell and cultured in either mTeSR1 (Stem Cell Technologies 85850) for at least one passage before differentiation into CNCCs or mTeSR Plus (Stem Cell Technologies 100–0276) for gene editing, single-cell cloning, expansion and maintenance. hESCs were grown on Matrigel growth factor reduced basement membrane matrix (Corning 354230) at 37 °C. hESCs were fed every day for mTeSR1 or every 2 days for mTeSR Plus and passaged every 5–6 days using ReLeSR (Stem Cell Technologies 05872).

HEK293FT cells were obtained from Invitrogen (R70007) and cultured in complete medium (DMEM-HG (GE Healthcare Life Science SH30243.01), 10% FBS, 1× Non-essential amino acids (Gibco 1114-0050), 1× GlutaMAX (Gibco 4109-0036), 1× antibiotic–antimycotic (Gibco 1524-0062)). Cells were fed every other day and passaged every 2–3 days using trypsin–EDTA (Gibco 25200072).

### AAV production and titration for CRISPR–Cas9 genome editing

Left and right homology arms (about 1 kilobase (kb)) surrounding the *SOX9* stop codon, flanking the linker–FKBP12-FV36–linker–mNeonGreen–linker–V5-stop tag, were cloned into an AAV backbone (pAAV-GFP, Addgene plasmid no. 32395). This vector plasmid, along with the AAV6 packaging plasmid pDGM6 (Addgene, plasmid no. 110660), was transfected into 70–80% confluent, early-passage HEK293FT cells seeded 24 h before transfection at about 8–9 million cells per 15-cm plate, and changed with fresh medium 2–6 h before transfection. For each 15-cm plate (2 per individual AAV6 preparation), the transfection mix was: 22 µg pDGM6, 6 µg vector plasmid, 120 g polyethyenimine (Sigma-Aldrich 408719), and Opti-MEM (Gibco 3198-5070) to 1 ml. At 24-h post-transfection, cells were changed into slow-growth medium (same as complete medium but with 1% FBS). Cells were collected 48 h after changing to slow-growth medium with the AAVpro Purification Kit Midi (Takara, 6675) as per the manufacturer's instructions.

Titration of purified AAV6 was carried out by quantitative PCR. Briefly, a previously flash-frozen and thawed 10-µl aliquot of virus was treated with TURBO DNase (Invitrogen, AM2238) as per the manufacturer's instructions to digest unpackaged DNA. DNase was inactivated by 0.001 M EDTA (final concentration) and incubation at 75 °C for 10 min. Virus DNA was released by proteinase K treatment (1 M NaCl, 1% w/v *N*-lauroylsarcosine, 100 µg ml$^{-1}$ proteinase K (Invitrogen, 25530049)) at 50 °C for 2 h to overnight. Samples were then boiled for 10 min, and diluted twice in H$_2$O such that the final dilution was 1:200,000. DNA standards comprising $10^{10}$–$10^3$ molecules were prepared using AAV6 backbone plasmids containing inverted terminal repeats. Quantitative PCR was carried out on standards and test samples using the Lightcycler 480 Probes Master kit (Roche, 04707494001) with inverted terminal repeat probe and primer sequences indicated in Supplementary Table 5.

### Generation of CRISPR–Cas9 and AAV genome-edited cell lines

hESCs were pre-treated with 10 µM RHO/ROCK pathway inhibitor Y-27632 (Stem Cell Technologies, 72304) for 2–24 h, collected and brought to single cells with Accutase and vigorous pipetting, and about 800,000 were nucleofected with a Cas9–sgRNA RNP complex using the Lonza 4D Nucleofection system. RNP consisted of 17 µg Sp-Cas9 HiFi (IDT) and 300 pmol sgRNA duplex (sequence in Supplementary Table 5). Cells were plated on Matrigel-coated plates with mTeSR Plus with 10 µM Y-27632 and AAV at desired multiplicity of infection (typically about 25,000). Cells were changed into mTeSR Plus with 10 µM Y-27632 but no AAV 4–24 h after initial plating, and an additional equal volume of mTeSR Plus with no Y-27632 was added 2 days later. Subsequent feedings were carried out with no Y-27632 until cells approached confluency, at which point cells were again collected and dissociated to single cells with Accutase (after 10 µM Y-27632 pre-treatment) and 500–1000 cells were plated per well of a 6-well plate. Cells were then expanded until colonies were of sufficient size to pick, before which cells were again pre-treated with 10 µM Y-27632 for 2–24 h. Colonies were picked into 24- or 48-well plates without Y-27632 and allowed to expand for about 5 days and passaged 1:2 using ReLeSR, with one half plated on another 24- or 48-well plate and the other half used for lysis with QuickExtract (Lucigen, QE9050). Genotyping PCR was carried out with one primer outside the homology arms and one primer inside the opposite homology arm (see Supplementary Table 5 for sequence). Clones containing the desired knock-in were expanded and used for genomic DNA extraction with the Quick-DNA miniprep kit (Zymo D3024), followed by the same genotyping PCR and Sanger sequencing to confirm knock-in.

## Differentiation of hESCs to CNCCs and chondrocytes

hESCs were grown for 5–6 days until large colonies formed, and then they were disaggregated using collagenase IV and gentle pipetting. Clumps of about 200 hESCs were washed in PBS and transferred to a 10 cm Petri dish in neural crest differentiation medium (1:1 ratio of DMEM-F12 and Neurobasal, 0.5× Gem21 NeuroPlex supplement with vitamin A (Gemini, 400-160), 0.5× N2 NeuroPlex supplement (Gemini, 400-163), 1× antibiotic–antimycotic, 0.5× Glutamax, 20 ng ml$^{-1}$ bFGF (PeproTech, 100-18B), 20 ng ml$^{-1}$ EGF (PeproTech, AF-100-15) and 5 μg ml$^{-1}$ bovine insulin (Gemini Bio-Products, 700-112P)). After 7–8 days, neural crest emerged from neural spheres attached to the Petri dish, and after 11 days, neural crest cells were passaged onto fibronectin-coated 6-well plates (about 1 million cells per well) using Accutase (Sigma-Aldrich A6964) and fed with neural crest maintenance medium (1:1 ratio of DMEM-F12 and neurobasal, 0.5× Gem21 NeuroPlex supplement with vitamin A (Gemini, 400-160), 0.5× N2 NeuroPlex supplement (Gemini, 400-163), 1× antibiotic–antimycotic, 0.5× Glutamax, 20 ng ml$^{-1}$ bFGF, 20 ng ml$^{-1}$ bFGF EGF and 1 mg ml$^{-1}$ BSA (Gemini)). After 2–3 days, neural crest cells were plated at about 1 million cells per well of a 6-well plate, and the following day cells were fed with neural crest long-term medium (neural crest maintenance medium + 50 pg ml$^{-1}$ BMP2 (PeproTech, 120-02) + 3 μM CHIR-99021 (Selleck Chemicals, S2924; BCh medium)). After transition to BCh medium, CNCCs at subsequent passages were plated at about 800,000 cells per well of a 6-well plate. CNCCs were then passaged twice to passage 4, at which depletion experiments were carried out, or cells were further differentiated to chondrocytes. For depletion experiments, dTAG$^{V}$-1 (Tocris, 6914) at a range of concentrations was added to BCh medium, with an equivalent amount of dimethylsulfoxide (DMSO) as vehicle control.

To differentiate CNCCs to chondrocytes, passage 3 CNCCs were passaged to passage 4, seeded at about 250,000 CNCCs per well of a 12-well plate, and grown for 3 days in BCh medium. Then, CNCCs were transitioned to chondrocyte medium without TGFb3 (ChM: DMEM-HG, 5% FBS, 1× ITS premix, 1 mM sodium pyruvate, 50 μg ml$^{-1}$ ascorbic acid, 0.1 μM dexamethasone and 1× antibiotic–antimycotic), with or without dTAG$^{V}$-1. The following day, cells were fed with chondrocyte medium with TGFb3 (ChMT: ChM + 10 ng ml$^{-1}$ TGFb3), with or without dTAG$^{V}$-1. Cells were fed every subsequent 3 days with ChMT. Cells were collected at day 10 and/or day 21 of the differentiation.

## sGAG quantification

Total sGAG levels per well of chondrocytes independently differentiated from CNCCs for 21 days, representing mature cartilage formation, were quantified using the Blyscan glycosaminoglycan assay (Biocolor). Briefly, collagen in the extracellular matrix was digested by washing cells with PBS and then adding 1 ml of Papain digestion buffer per well of a 12-well plate. Cells were incubated at 65 °C for 3 h with gentle agitation every 30 min, then 0.5 ml additional digestion buffer was added and lysate was moved to Eppendorf tubes and incubated at 65 °C overnight. Quantification of sGAG content from about 10 μl of the lysates was carried out as per the manufacturer's instructions, and the volume of each lysate was measured separately and used to infer the total sGAG content of the entire well.

## Flow cytometry

CNCCs were collected for flow cytometry using Accutase and quenching with FACS buffer (5% FBS in PBS). Chondrocytes were incubated in digestion medium (DMEM-KO, 1 mg ml$^{-1}$ Pronase (Roche, 11459643001), 1 mg ml$^{-1}$ collagenase B (Roche, 11088815001), 4 U ml$^{-1}$ hyaluronidase (Sigma, H3506-500 mg)) for about 1 h with gentle agitation every 15 min. Digested cells were then washed twice in PBS. Flow cytometry was used to measure mNeonGreen fluorescence after excluding doublets and debris based on forward and side scatter (Beckman Coulter Cytoflex). Fluorescence values were summarized per biological replicate using geometric means. The relative SOX9 level as percentage of the *SOX9*-tagged, unperturbed (treated with DMSO) sample was calculated by first subtracting the

geometric mean fluorescence of the untagged (WT) sample from both the unperturbed and dTAG$^{V}$-1-treated sample, and then dividing the dTAG$^{V}$-1-treated sample fluorescence by the unperturbed sample fluorescence.

## Protein collection and western blotting

Cells were washed with PBS and scraped into RIPA buffer (50 mM Tris, 150 mM NaCl, 1% NP-40, 0.1% Na deoxycholate, 0.1% SDS in H$_2$O with 1× protease inhibitor cocktail (Sigma-Aldrich 4693132001)), incubated on ice for 10 min, and sonicated to disrupt pelleted DNA using Bioruptor Plus (Diagenode). Sonicated lysates were incubated on ice for 10 min, and centrifuged at 16,000g for 10 min at 4 °C to pellet debris. Supernatants were normalized to the same protein content using the Pierce BCA Protein Assay kit (ThermoFisher, 23225), mixed with 4× SDS sample loading buffer (Invitrogen NP0007) and 0.1 M dithiothreitol (DTT), and boiled for 7 min. Samples were separated on Tris-glycine polyacrylamide gel electrophoresis (PAGE) gels in 1× Tris-glycine buffer with 0.1% SDS, transferred in 1× Tris-glycine buffer with 20% methanol, blocked in 5% milk + 1% BSA in PBST, immunoblotted with either SOX9 antibody (1:1,000, Sigma-Aldrich AB5535) or β-actin antibody (1:20,000, Abcam ab49900) overnight at 4 °C, probed with the appropriate secondary, developed using Pierce ECL Western Blotting Substrate (ThermoFisher, 32106), and imaged using an Amersham ImageQuant 800 system (Cytiva).

## RNA isolation and preparation of RNA-seq libraries

Total RNA was extracted from CNCCs using Trizol reagent (Invitrogen) followed by Quick-RNA Miniprep kit (Zymo) with on-column DNase I digestion. Unstranded mRNA libraries were prepared with the NEBNext Ultra II RNA Library Prep Kit for Illumina (NEB no. E7770S/L).

## Metabolic RNA labeling and preparation of SLAM-seq libraries

4-Thiouridine was incorporated into nascent transcripts by incubating CNCCs with BCh medium containing 100 μM 4-thiouridine, as well as DMSO or 500 nM dTAG$^{V}$-1 depending on experimental condition, for 2 h. Plates were covered in foil and handling was carried out in a hood with no light where possible. For 3- and 24-h depletion experiments, labeling was started at 1 and 22 h after dTAG$^{V}$-1 addition, respectively.

Total RNA was extracted using Trizol reagent, phenol–chloroform extraction was carried out, and the aqueous phase was used as input to the Quick-RNA Miniprep kit. During RNA extraction with Quick-RNA Miniprep kit, 0.1 mM DTT was added to the RNA wash and RNA pre-wash buffers, but the on-column DNase I step was skipped. RNA was eluted in H$_2$O with 1 mM DTT, quantified with Qubit RNA Broad Range assay (ThermoFisher, Q10211), and ≥2 μg total RNA was used as input to the alkylation reaction. Alkylation was carried out in dark tubes after which light exposure was allowed, and after quenching RNA was purified and subjected to on-column DNase I digestion using the RNA Clean & Concentrator-5 kit (Zymo, R1013).

A 500 ng quantity of alkylated RNA was used as input to QuantSeq 3′ mRNA-Seq Library Prep Kit FWD with unique dual index add-on (Lexogen, 113.96), with 15 cycles of PCR amplification. Library size distributions were confirmed by separation on a PAGE gel and staining with SYBRGold and pooled on the basis of quantifications from Qubit dsDNA High Sensitivity Kit (ThermoFisher Q32854). Pooled libraries were sequenced using Novaseq 6000 platform (2× 150 bp).

## ATAC-seq collection and library preparation

CNCCs were incubated with BCh medium containing 200 U ml DNase I (Worthington, LS002007) for 30 min and collected using Accutase. Viable cells were counted using a Countess Automated Cell Counter (Invitrogen), and 50,000 viable cells were pelleted at 500 RCF for 5 min at 4 °C and resuspended in ATAC-resuspension buffer (10 mM Tris-HCl pH 7.4, 10 mM NaCl, 3 mM MgCl$_2$ in sterile water) containing 0.1% NP-40, 0.1% Tween20 and 0.01% digitonin and incubated on ice for 3 min. Following wash-out with cold ATAC-resuspension buffer containing 0.1% Tween20, cells were pelleted and resuspended in 50 μl transposition mix

(25 μl 2× TD buffer, 2.5 μl transposase (100 nM final), 16.5 μl PBS, 0.5 μl 1% digitonin, 0.5 μl 10% Tween20, 5 μl H₂O) and incubated for 30 min at 37 °C with shaking. The reaction was purified using the Zymo DNA Clean & Concentrator kit and PCR-amplified with NEBNext High-Fidelity 2× PCR Master Mix (NEB, M0541L) and primers as defined in ref. 62. Libraries were purified by two rounds of double-sided size selection with AMPure XP beads (Beckman Coulter, A63881), with the initial round of 0.5× sample volume of beads followed by a second round with 1.3× initial volume of beads. Library size distributions were confirmed by separation on a PAGE gel and staining with SYBRGold and pooled on the basis of quantifications from Qubit dsDNA High Sensitivity Kit. Pooled libraries were sequenced using the Novaseq 6000 platform (2 × 150 bp).

### ChIP and library preparation

One fully confluent 10-cm plate of cells was crosslinked per ChIP experiment in 10 ml PBS with 1% methanol-free formaldehyde for 10 min and quenched with a final concentration of 0.125 M glycine for 10 min with nutation. Crosslinked cells were scraped into tubes with 0.001% Triton X in PBS, washed with PBS without Triton, pelleted by centrifugation, flash-frozen in liquid nitrogen and stored at −80 °C. Samples were defrosted on ice and resuspended in 5 ml LB1 (50 mM HEPES-KOH pH 7.5, 140 mM NaCl, 1 mM EDTA, 10% glycerol, 0.5% NP-40, 0.25% Triton X-100, with 1× cOmplete Protease Inhibitor Cocktail and phenylmethylsulfonyl fluoride) and rotated vertically for 10 min at 4 °C. Samples were centrifuged for 5 min at 1,350 g at 4 °C, and resuspended in 5 ml LB2 (10 mM Tris, 200 mM NaCl, 1 mM EDTA, 0.5 mM EGTA, with 1× cOmplete Protease Inhibitor Cocktail and optionally 1 mM phenylmethylsulfonyl fluoride) and rotated vertically for 10 min at 4 °C. Samples were centrifuged for 5 min at 1,350 g at 4 °C, resuspended in 300 μl LB3 per sonicated sample, and incubated for 10 min on ice. Samples were sonicated in 1.5 ml Bioruptor Plus TPX microtubes (Diagenode, c30010010-50) on Bioruptor Plus for 10 cycles of 30 s on–30 s off. Every 5 cycles, samples were lightly vortex and briefly centrifuged. Samples were diluted in additional LB3 to 1 ml, pelleted at 16,000 RCF for 10 min, and the supernatant was removed. Triton X-100 was added to 1%.

To check DNA size distribution and quantity, a 10-μl aliquot of sonicated chromatin from each sample was diluted to 100 μl in elution buffer (50 mM Tris, 10 mM EDTA, 1% SDS) with 0.0125 M NaCl and 0.2 mg ml⁻¹ RNase A and incubated at 65 °C for 1 h, followed by addition of proteinase K to 0.2 mg ml⁻¹ and an additional 1 h of 65 °C incubation. DNA was purified using Zymo DNA Clean & Concentrator Kit with ChIP DNA Binding Buffer (Zymo, D5201-1-50) and size distribution and quantity was assessed by separation on a 1% agarose gel and Qubit HS DNA kit, respectively. Qubit measurements were used to normalize samples to the same DNA concentration.

Following normalization, the chromatin was divided for input (2%) and ChIP samples. A minimum of 25 μg DNA was used for histone ChIP analyses, and 50 μg for V5 ChIP analyses. A 5 μg quantity of anti-H3K27ac (Active Motif, 39133) antibody (1:200 dilution) or a 10 μg quantity of anti-V5 (Abcam, ab9116 or ab15828) or TWIST1 (Abcam, ab50887) antibody (1:100 dilution) was added per ChIP sample, and incubated overnight at 4 °C. Protein G Dynabeads (ThermoFisher) were first blocked with Block solution (0.5% BSA (w/v) in 1× PBS) and then added to cleared chromatin to bind antibody-bound chromatin for a 4–6 h incubation. Chromatin-bound Dynabeads were washed at least 6 times with chilled RIPA wash buffer (50 mM HEPES-KOH pH 7.5, 500 mM LiCl, 1 mM EDTA, 1% NP-40, 0.7% Na deoxycholate), followed by a wash with chilled TE + 50 mM NaCl. Chromatin was eluted for 30 min in elution buffer (50 mM Tris, 10 mM EDTA, 1% SDS) at 65 °C with frequent vortexing. The ChIP and input samples were then incubated at 65 °C overnight to reverse crosslinks (12–16 h). Samples were diluted and sequentially digested with RNase A (0.2 mg ml⁻¹) for 2 h at 37 °C followed by proteinase K (0.2 mg ml⁻¹) for 2 h at 55 °C for 2–4 h to digest protein. ChIP and input samples were purified by Zymo DNA Clean & Concentrator Kit with ChIP DNA binding buffer.

For library preparation, samples were quantified by Qubit dsDNA HS assay kit, and 10–50 ng of ChIP DNA was used for library preparation with end repair, A-tailing and adaptor ligation (NEB). Following USER enzyme treatment, libraries were cleaned up with one round of single-side AMPure XP bead clean-up, and then amplified to add indices using NEBNext Ultra II Q5 Master Mix and NEBNext Multiplex Oligos for Illumina kit (NEB, E7335S) with 4–10 cycles (as determined by input amounts from NEB protocol). ChIP libraries were purified by two rounds of double-sided AMPure XP bead clean-up (0.5× then 0.4× initial sample volume of beads added) to remove large fragments and deplete adaptors. Library concentration and quality within ChIP or input groups was assessed by Qubit dsDNA HS assay kit and separation on a PAGE gel, and used to pool within ChIP or input groups. KAPA quantitative PCR was used to pool across ChIP or input groups. Pooled libraries were sequenced using the Novaseq 6000 platform (2 × 150 bp).

### Sequencing data pre-processing

**ATAC-seq and ChIP–seq.** For both ATAC-seq and ChIP–seq, Nextera (ATAC) or Truseq (ChIP) adapter sequences and low-quality bases (-Q10) were trimmed from sequencing reads using skewer v0.2.2 and aligned to the human genome (hg38) using bowtie2 v2.4.1 with the following settings: --very-sensitive, --X 2000. Read mate pair information was corrected with samtools v1.10 fixmate, PCR duplicates were removed using samtools markdup, and mitochondrial reads and low-mapping-quality reads (-q 20) were removed using samtools v1.10 view. bigWig files for visualization were generated using deeptools v3.5.0 bamCoverage with the following settings: -bs 10 --normalizeUsing RPGC --samFlagInclude 64 --samFlagExclude 8 --extendReads.

For ATAC-seq, a custom approach was used to define regions that showed reproducible peaks of accessibility across samples. Shifted bed sites were obtained from mapped and filtered ATAC bam files, and bed files for each sample were used to call peak summits using MACS2 v2.2.7.1 callpeak with the following settings: --nomodel --keep-dup all --extsize 200 --shift 100 --SPMR. Then, within each differentiation/line replicate, summits within 75 bp were merged, taking the average location across summits as the location of the merged summit. Then, across each differentiation/line, summits within 150 bp were merged, again taking the average location. Only those merged summits with at least one constituent summit from three or more differentiation/line instances were carried forward. These summits were extended 250 bp in either direction (using bedtools v2.29.2 slop), and finally all such regions were merged (using bedtools v2.29.2 merge) such that there were no overlapping regions, resulting in 151,457 reproducible peak regions. For TWIST1 ChIP–seq, peaks were called using MACS2 v2.2.7.1 callpeak with default settings, and the fraction of reads that lay in peaks was calculated for each ChIP–seq experiment using samtools v1.10 view.

**RNA-seq.** TruSeq adapter sequences and low-quality bases were trimmed from sequencing reads using skewer v0.2.2, and transcript levels were quantified using salmon v1.4.0 quant with the following settings: --gcBias --seqBias -l A. Salmon abundance files were summarized to the gene level and imported into R with the tximport package v1.20.0 with countsFromAbundance = 'lengthScaledTPM'. The human reference genome hg38 and Ensembl transcriptome v99 were used.

**SLAM-seq.** Lexogen adapter sequences and low-quality bases were trimmed from sequencing reads (read 1 only) using skewer, followed by trimming of poly(A) sequences. Trimmed reads were used as input to slamdunk v0.4.3 (ref. 63), with the following individual step parameters modified from default: map, -n 100 -5 0; count, -l 150.

### Quantification and statistical analysis

**Sequence motif matching.** TF sequence motif position weight matrices for the indicated TFs were obtained from HOCOMOCO core motifs:

SOX9, SOX9_HUMAN.H11MO.0.B; TFAP2A, AP2A_HUMAN.H11MO.0.A; NR2F1, COT2_HUMAN.H11MO.0.A. The coordinator motif corresponding to TWIST1 was obtained from a previous publication[38]. The SOX9 palindrome motif was constructed by inverting the single HOCOMOCO position weight matrix at various spacings from 0 to 10 bp. All motifs were matched to the human genome (hg38) using fimo v5.1.1 with a $P$-value threshold of $1 \times 10^{-4}$.

**Differential expression/accessibility testing.** Differential expression or accessibility between pairs of SOX9 concentrations (ATAC/RNA) or time points of full SOX9 depletion (ATAC, SLAM, H3K27ac/V5/TWIST1 ChIP) was carried out using DESeq2 v1.32.0, with CNCC differentiation batch as a covariate and raw counts as input. For SLAM one additional surrogate variable, discovered using sva 3.4.0, was also used as a covariate. For TWIST1 ChIP, the fraction of reads in peaks was also used as a covariate to correct for overall ChIP enrichment (which is not expected to change as a function of SOX9 dosage). For ATAC and H3K27ac/V5/TWIST1 ChIP, counts over all 151,457 reproducible peak regions were used; for RNA, only protein-coding genes with at least 1 transcript per million in at least 6 samples were used; and for SLAM-seq, only protein-coding genes with at least 1 CPM in at least 3 samples were used. The independentFiltering option in DESeq2 was set to FALSE, except for H3K27ac/V5/TWIST1 ChIP differential analyses.

**Modeling of SOX9 dose–response curves (ATAC/RNA).** All RE/gene CPM values were first TMM-normalized using the edgeR package v3.34.0. For each SOX9-dependent RE/gene, defined by 5% FDR comparing depleted versus fully depleted SOX9, CPM values across all *SOX9*-tagged samples (that is, from all six SOX9 concentrations) were corrected for differentiation batch effect by linear regression using the lm() function. Differentiation-corrected CPM values were scaled by dividing by the maximum absolute value across samples. Sample outliers, defined as $z$-score greater than 3, were removed from the analysis of that RE/gene. The data were then fitted to either a linear model as a function of SOX9 dosage (defined by flow cytometry), or to the Hill equation using the drm() function in the drc R package v3.0-1. All comparisons of $ED_{50}$/Hill coefficients between sets of genes/REs were carried out using the Hill equation. For most genes/REs, a two-parameter Hill equation (that is, with minimum and maximum fixed as the mean CPM at full or no depletion, respectively) was sufficient. However, for a small subset of REs (8%) and genes (5%), a three-parameter Hill equation with fixed minimum but free maximum was a better fit (decrease in AIC > 2 relative to the two-parameter model); for these genes/REs, the three-parameter Hill was used. The type of Hill equation (two or three parameter) used for each gene/RE is indicated in Supplementary Tables 1 and 2. To calculate the 'buffering index' at a given SOX9 dosage such as 50% (see Extended Data Fig. 3), the change in the fitted Hill equation curve going from 100% to 50% SOX9 dosage was divided by the total SOX9-dependent change (that is, going from 100% to 0%), multiplied by 100, and then subtracted from 100. A value of 0 of this statistic indicates no buffering (that is, the entirety of SOX9-dependent change has occurred by 50% SOX9 dosage) while a value of 100 indicates complete buffering (that is, no change until <50% SOX9 dosage).

**Bootstrapping for $ED_{50}$/Hill exponent confidence interval estimation.** Point estimates for $ED_{50}$ and the Hill exponent from fitted Hill equations vary nonrandomly with both the relative quality of the fitted Hill equation (with fitted parameters for REs/genes fitted better by a linear model having more uncertainty) and the overall magnitude of $ED_{50}$/Hill exponents (higher magnitudes having greater uncertainty). We noticed instability in the $ED_{50}$/Hill standard errors obtained from parametric least-squares fitting in the drc package; we therefore implemented a bootstrap procedure to quantify uncertainty in $ED_{50}$/Hill estimates at either the individual RE/gene level or when

comparing groups of REs/genes in their $ED_{50}$ or Hill exponent values. For each RE/gene, a set of 200 bootstrapped datasets was generated by sampling the number of replicates (generally 7) with replacement from each of the six conditions. Note that while the number of potential bootstraps from a single condition is relatively small (7!), carrying out this sampling independently in each of the six conditions generates a very large number of unique datasets (7!⁶). Hill equations were fitted to each bootstrapped dataset and $ED_{50}$/Hill exponents were extracted.

For uncertainty estimates for individual genes, the 200 bootstrap replicates were summarized to determine 95% confidence intervals. When comparing groups of genes, rather than first summarizing bootstraps within genes, the relative group statistic (typically median) was computed across all genes for each of 200 bootstrap replicates separately; the resulting 200 group statistics were then used to construct 95% confidence intervals.

**Prediction of SOX9-dependent RNA changes from ATAC changes.** An extension of the ABC model[46] was used to predict gene expression fold changes at each SOX9 concentration (relative to undepleted) from ATAC-seq fold changes at nearby REs from the same comparisons. Briefly, the (ABC) model defines the contribution, or ABC score, of a given RE within 5 Mb of a gene transcription start site as:

$$ABC_{RE,G} = \frac{dist^{-0.7} \times \sqrt{ATAC \times K27ac}}{\sum \left( dist^{-0.7} \times \sqrt{ATAC \times K27ac} \right)}$$

ABC scores for all RE–gene pairs (within 5 Mb) were calculated using this formula. In this case, a linear distance–power law function was used as a proxy for 'contact,' as it has been shown to have a similar performance to Hi-C[46]. A gene's own promoter (defined as an RE within 1 kb of the consensus transcription start site) was excluded for the purposes of gene-level predictions, as promoter accessibility is often reflective of gene transcriptional changes. For 'activity' calculations, ATAC-seq and H3K27ac counts from unperturbed (*SOX9*-tagged, DMSO-treated) CNCCs were used.

A gene's predicted relative level at a certain SOX9 concentration was calculated as the sum of the ABC scores of all REs within 5 Mb. As H3K27ac ChIP–seq was available only from unperturbed or fully depleted *SOX9*-tagged CNCCs, RE ABC scores at lower SOX9 concentrations were calculated by multiplying the unperturbed ABC score by the DESeq-estimated fold change for that RE when comparing unperturbed CNCCs to the given SOX9 concentration. While this assumes an identical decrease in H3K27ac at every SOX9 concentration, fold changes in RE ATAC and H3K27ac signals were observed to be highly correlated following full SOX9 depletion. Effectively, this approach predicts the fold change in gene expression as a weighted sum of fold changes in all REs within 5 Mb, for which the weights are the RE ABC scores from the unperturbed setting:

$$\Delta G = \frac{\sum_{RE \, within \, 5 \, Mb} ABC_{RE,G} \times \Delta ATAC_{RE}}{\sum_{RE \, within \, 5 \, Mb} ABC_{RE,G}}$$

**Analysis of gene–craniofacial disorder associations.** The list of genes that cause PRS-like phenotypes when mutated in humans or mice was obtained from ref. 48. Genes with craniofacial disorder associations distinct from PRS were defined as the list of craniofacial disorder genes from ref. 24, removing all of the genes that cause PRS-like phenotypes. This non-PRS-like gene set was further stratified into causing dominant or recessive disorders on the basis of the corresponding annotation in Online Mendelian Inheritance in Man[64].

**PRS endophenotype definition and GWAS**
**Sample.** The control sample of healthy individuals comprised three-dimensional facial scans of 8,246 unrelated individuals of European ancestry (60.3% females; median age = 18.0 years, interquartile

range = 9.0 years) originating from the USA and the UK. The sample of PRS comprised 13 participants (9 females; median age = 12.01 years, interquartile range = 5.17 years). Images were excluded if participants were laughing, crying or otherwise emoting or judged to be of poor quality or if the non-rigid registration failed. Participants with missing covariate information (for example, age or sex) were also removed.

**Genotyping.** Imputed genotypes were available for all individuals of the European control sample. After quality control, 7,417,619 SNPs were used for analysis. SNPs on the X chromosome were coded 0/2 for hemizygous males, to match with the 0/1/2 coding for females.

## Phenotyping
**Correction for asymmetry and covariates.** Facial images were processed in MeshMonk to obtain a standard facial representation, characterized by 7,160 homologous quasi-landmarks including midline and bilaterally paired quasi-landmarks[65]. Each configuration was made symmetrical following the Klingenberg protocol[66]: for each configuration, a reflected copy was made by reversing the sign of the $x$ coordinate of each quasi-landmark. Bilaterally paired quasi-landmarks were relabeled left to right and right to left in the reflected copy. The reflected and relabeled copy was then aligned to the original by least-squares Procrustes superimposition. The average of the two copies was taken as the symmetrical version of the configuration.

The US and UK samples were adjusted for covariates sex, age and age-squared as follows. All symmetrized quasi-landmark configurations were aligned by generalized Procrustes analysis. The average configuration was recorded. A partial least-squares regression of the configurations onto the covariates was carried out. The average configuration was added to the residuals to produce the corrected configurations of the US and UK samples. The regression coefficients were retained to adjust the PRS sample for the same covariates using the same regression model. Specifically, each symmetrized landmark configuration of the PRS sample was aligned to the recorded average configuration. The predicted configuration for their sex, age and age-squared was calculated from the recorded regression coefficients and was subtracted from their symmetrized and aligned configuration. The coordinates of the average configuration were then added back on to produce the corrected version of the participant with PRS.

**PRS-driven phenotyping.** Facial shape was partitioned into 63 global-to-local segments by hierarchical spectral clustering[30]. For each subset of quasi-landmarks belonging to each of the 63 facial segments, a PRS-driven univariate trait was defined as follows. First the symmetrized and adjusted quasi-landmark configurations of the US and UK samples were co-aligned by generalized Procrustes analysis, and this was carried out separately for each segment. The dimensionality was reduced by PC analysis with the optimal number of PCs to retain determined by parallel analysis. Projections on each PC were normalized to have unit variance by dividing each projection by the standard deviation of all projections. These standard deviations were retained. The symmetrized and adjusted landmark configurations of the PRS sample were then aligned to the average and projected into the space of the PCs and normalized by the recorded standard deviations. Finally, per facial segment, a PRS-driven facial trait was defined as the vector or direction passing through the global average and average PRS facial shape.

Each participant in the US and UK samples was 'scored' on the PRS-driven facial traits by computing the cosine of the angle between: the vector from the average of the PC projections of the US and UK samples to the PC projections of the participant; and the vector from the average of the US and UK projections to the average of the PRS projections. These scores were computed by leave-one-out such that each participant was excluded from training the vectors on which they were scored.

**Significance testing.** To test the significance of the PRS-driven trait in each facial module, the PRS sample was compared to a matched control sample of equal size drawn from the US and UK samples. The matched control sample was selected randomly as follows, separately for each facial module. In random order, each participant in the PRS sample was matched to the participant from the combined US and UK samples of the same sex that was closest in age. This participant was then removed from the possible matches so that each US/UK participant could be matched to only one PRS participant. The covariate-adjusted and symmetrized quasi-landmarks were co-aligned by generalized Procrustes analysis and regressed onto group membership (0 = US/UK; 1 = PRS) using partial least-squares regression. A $P$ value was generated by a permutation test on $R$-squared with 10,000 permutations. In 30 out of 63 facial segments, a significant difference ($P < 0.05$) in facial shape was observed between the two groups (PRS versus healthy controls).

## GWAS
The scores on the 30 PRS-driven univariate traits, for which a significant difference was observed, were combined into a single phenotype matrix ($[N \times M]$ with $N = 8,246$ controls and $M = 30$ facial segments). This matrix was tested for genotype–phenotype associations in a multivariate meta-analysis framework using canonical correlation analysis (canoncorr in Matlab 2017b). However, instead of carrying out a separate GWAS per facial segment, information across multiple segments is now combined into a single multivariate GWAS. As canonical correlation analysis does not accommodate adjustments for covariates, we removed the effect of relevant covariates (sex, age, age-squared, height, weight, facial size, four genomic ancestry axes, camera system), on both the independent (SNP) and the dependent (facial shape) variables using partial least-squares regression (plsregress from Matlab 2017b) before GWAS.

The US and UK subsamples served both as identification and replication sets in a two-stage design, after which the $P$ values were meta-analyzed using Stouffer's method[67,68]. Per SNP, the lowest $P$ value was selected (meta$_{US}$ versus meta$_{UK}$) and compared against the genome-wide Bonferroni threshold ($5 \times 10^{-8}$). We observed 1,767 SNPs at the level of genome-wide significance, which were clumped into 22 independent loci as follows. Starting from the lead SNP (lowest $P$ value), SNPs within 10 kb or within 1 Mb but with $r^2 > 0.01$ were clumped into the same locus represented by the lead SNP. Next, considering only the lead SNPs, signals within 10 Mb and with an $r^2 > 0.01$ were merged. Third, any locus with a singleton lead SNP was removed.

## Post-GWAS analyses
To define facial shape, GWAS SNPs that affect facial shape in either a PRS-like or non-PRS-like manner, we obtained the combined list of SNPs affecting normal-range variation (and not orofacial clefting) in facial shape from ref. 24. We tested each of these SNPs directly for association with the PRS endophenotype from the above-described GWAS. SNPs with a Bonferroni-corrected $P$ value < 0.01 (corresponding to an uncorrected $P$-value cutoff of $7 \times 10^{-5}$) in either the UK or UK cohort from the PRS GWAS were considered significant. We then considered all SNPs in linkage disequilibrium ($r^2 > 0.5$, identified with SNiPA[69] 'Proxy Search' tools using 1000 Genomes Phase 3 v5 European reference panel) with either set of facial GWAS SNPs (no PRS endophenotype association or significantly associated). SOX9-dependent REs containing these SNPs were assigned as 'PRS-like' or affecting other aspects of facial shape according to the type of linked SNP they contained.

## Reporting summary
Further information on research design is available in the Nature Portfolio Reporting Summary linked to this article.

## Data availability

The raw sequencing files generated during this study are available on the Gene Expression Omnibus (accession number GSE205904); corresponding processed data are available on Zenodo[70]. TF-binding motifs were obtained from HOCOMOCO v11 (https://hocomoco11.autosome.org/). Gene Ontology assignments were obtained from AmiGO (http://amigo.geneontology.org/amigo). All analyses were carried out on human genome version hg38, except for PRS endophenotype GWAS (hg19). The raw source data for the facial phenotypes—the three-dimensional facial surface models in.obj format—are available through the FaceBase Consortium (https://www.facebase.org). Access to these three-dimensional facial surface models requires proper institutional ethics approval and approval from the FaceBase data access committee. Facial scans from patients with PRS (used to define the PRS endophenotype) are available through the FaceBase Consortium (https://www.facebase.org FB00000861) under controlled access. The participants making up the US dataset of healthy individuals used for PRS endophenotype GWAS were not collected with broad data sharing consent. Given the highly identifiable nature of both facial and genomic information and unresolved issues regarding risks to participants of inherent reidentification, participants were not consented for inclusion in public repositories or the posting of individual data. This restriction is not because of any personal or commercial interests. Further information about access to the raw three-dimensional facial images and/or genomic data can be obtained from the PSU IRB (IRB-ORP@psu.edu, and the IUPUI IRB (irb@iu.edu). The ALSPAC (UK) data will be made available to bona fide researchers on application to the ALSPAC Executive Committee (https://www.bristol.ac.uk/alspac/researchers/access/). Summary statistics from the PRS endophenotype GWAS are available on the GWAS Catalog (GCP000517). Plasmids generated in this study have been deposited in Addgene (plasmid no. 194971). All other reagents are available upon request to J.W. Source data are provided with this paper.

## Code availability

Custom code used for analysis of processed sequencing data is available on Zenodo[70]. KU Leuven provides the MeshMonk (v.0.0.6) spatially dense facial-mapping software, free to use for academic purposes (https://github.com/TheWebMonks/meshmonk)[71]. Matlab 2017b implementations of the hierarchical spectral clustering to obtain facial segmentations are available on Figshare (https://doi.org/10.6084/m9.figshare.7649024.v1)[72]. This custom code, combined with the publicly available Matlab 2017b functions described in the relevant Methods sections, can be used to reproduce the PRS endophenotype GWAS.

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

## Acknowledgements

We thank members of the laboratories of J.W. and J.K.P., as well as James Ferrell, for helpful comments. This research was supported by National Institutes of Health (NIH) grants R01HG008140 and R01HG011432 (J.K.P.), and R35GM131757-1 (J.W.), and financial support from the Howard Hughes Medical Institute and the Nomis Foundation. S.N. was supported by a Helen Hay Whitney Fellowship. S.K. was supported by a Damon Runyon Fellowship (DRG-2420-21). J.W. was supported by a Lorry Lokey endowed professorship and a Stinehart Reed award. The KU Leuven research team (P.C., H.H., H.S.M.) was supported by the NIH (1-R01-DE027023, 2-R01-DE027023), The Research Fund KU Leuven (BOF-C1, C14/15/081 and C14/20/081) and The Research Program of the Research Foundation - Flanders (Belgium) (FWO, G078518N). R.A.S., B.H. and O.D.K. were supported by NIH U01DE024440. The computational resources and services used in this work were provided by the VSC (Flemish Supercomputer Center), funded by the Research Foundation - Flanders (FWO) and the Flemish Government – department EWI. Some of the computing for this project was carried out on the Sherlock cluster. We thank Stanford University and the Stanford Research Computing Center for providing computational resources and support that contributed to these research results. The funders had no role in study design, data collection and analysis, decision to publish or preparation of the manuscript.

## Author contributions

Conceptualization: S.N., T.S., J.K.P. and J.W.; formal analysis: S.N., S.K., H.H. and H.S.M.; investigation: S.N. and S.K.; resources: R.A.S, B.H., O.D.K., P.C., J.K.P. and J.W.; writing - original draft, S.N.; writing - review and editing: S.N., J.K.P. and J.W.; visualization: S.N., H.H. and H.S.M.; supervision: J.K.P. and J.W.; funding acquisition: J.K.P. and J.W.

## Competing interests

J.W. serves on the scientific advisory board for Camp4 Therapeutics and Paratus Sciences. All other authors declare no competing interests.

## Additional information

**Extended data** is available for this paper at https://doi.org/10.1038/s41588-023-01366-2.

**Correspondence and requests for materials** should be addressed to Jonathan K. Pritchard or Joanna Wysocka.

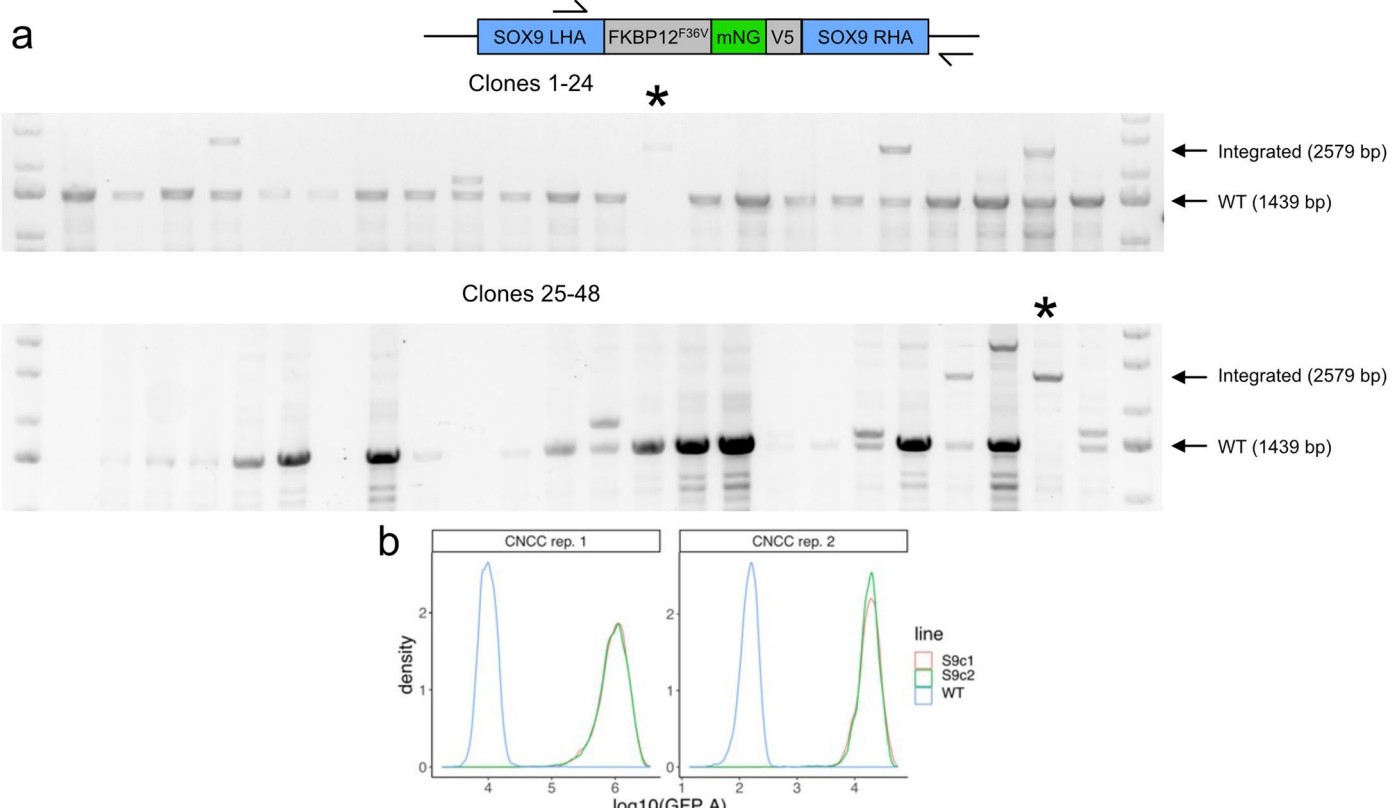

**Extended Data Fig. 1 | Endogenous C-terminal tagging of SOX9.** (**a**) Genome editing of WT hESCs to derive SOX9-tagged hESCs. Top, schematic depicting primer (arrow) locations for clonal genotyping of the SOX9 locus. SOX9LHA-FKBP12FV36-mNG-V5-SOX9RHA is the full homology-directed repair template provided by AAV6, so the right primer is located outside the homology arm. Bottom, agarose gel images of PCR using depicted primers on 48 analyzed hESC clones nucleofected with SOX9 sgRNA-Cas9 RNP and transduced with tag-containing AAV6. * clones with bi-allelic knock-in. (**b**) Single-cell distributions of mNeonGreen fluorescence (at least 7,000 cells per histogram) between two SOX9-tagged clones from two CNCC replicates. Representative of two independent experiments.

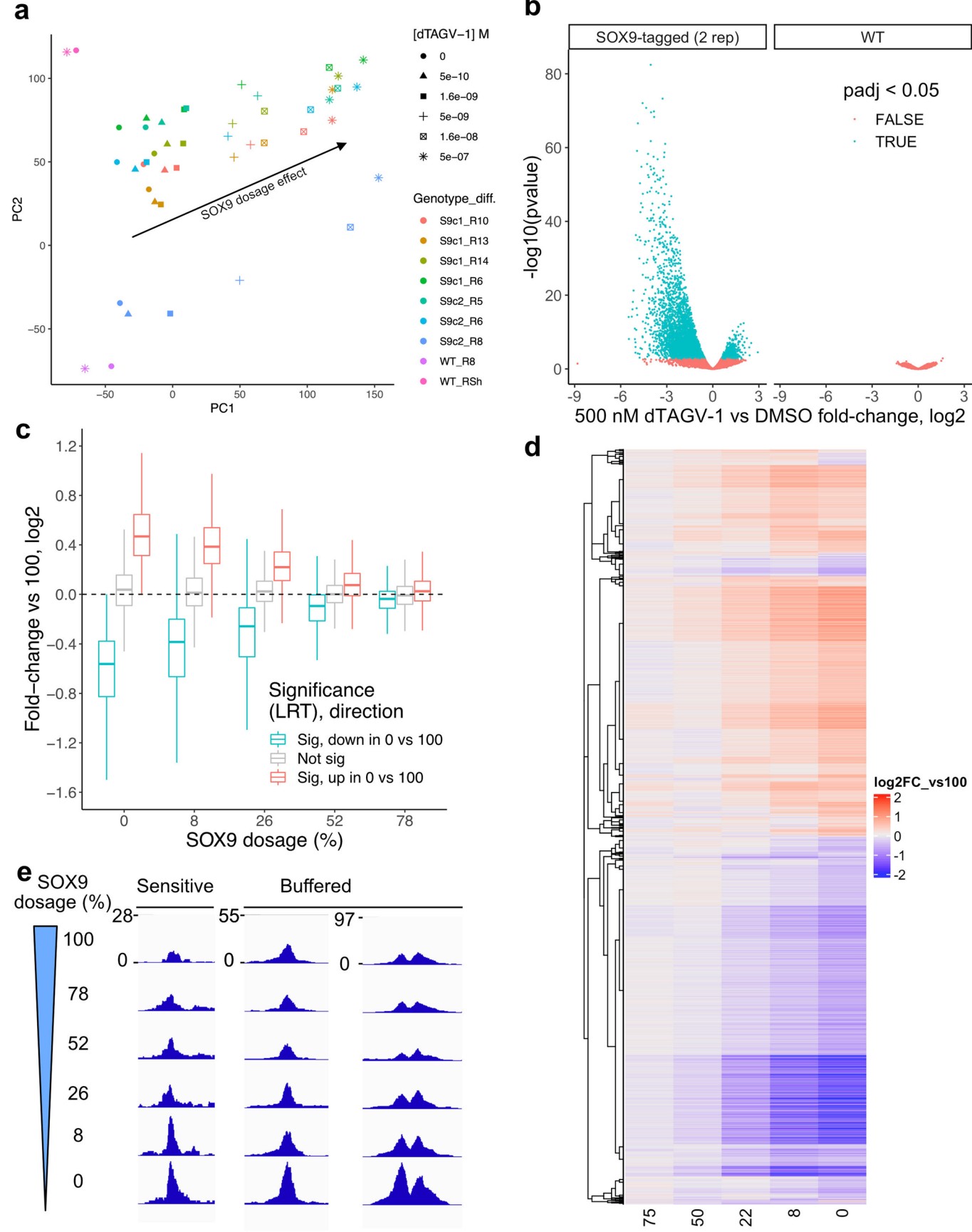

**Extended Data Fig. 2 | See next page for caption.**

**Extended Data Fig. 2 | Effects of SOX9 dosage changes on chromatin accessibility.** (**a**) Principal component analysis of ATAC-seq counts per million (CPM) of all 151,457 REs across all CNCC samples. Shapes indicate the dTAGV-1 concentration treated for 48 h. Colors indicate the combination of hESC line from which CNCCs were derived and differentiation batch (S9c1/2 = SOX9-tagged clone1/2). Arrow indicates the SOX9 dosage effect. (**b**) Volcano plot of 500 nM dTAGV-1 treatment on two SOX9-tagged (left) or two WT (right) CNCC differentiation replicates for all 151,457 REs. -log10(p-value) (y-axis) represents the unadjusted two-sided p-value from DESeq2, point color represents Benjamini-Hochberg adjusted p-value. (**c**) Distributions of fold-changes versus full (100) SOX9 dosage for all REs for which SOX9 dosage explains a significant (5% FDR; red, 16,538; blue, 27,334) or nonsignificant (grey, n = 107,585) amount of variance (likelihood ratio test, LRT), stratified by the direction of change in full SOX9 depletion. Boxplot center represents median, box bounds represent 25th and 75th percentiles, and whiskers represent 5th and 95th percnetiles. (**d**) Same fold-change values as in (c) for a random subset (n = 10,000) of significant REs, plotted as a heatmap and clustered by row based on Kendall distance. (**e**) Example ATAC-seq browser tracks of individual RE accessibility at different SOX9 dosages (y-axis, normalized coverage in 10 bp bins), averaged across all replicates at each dosage.

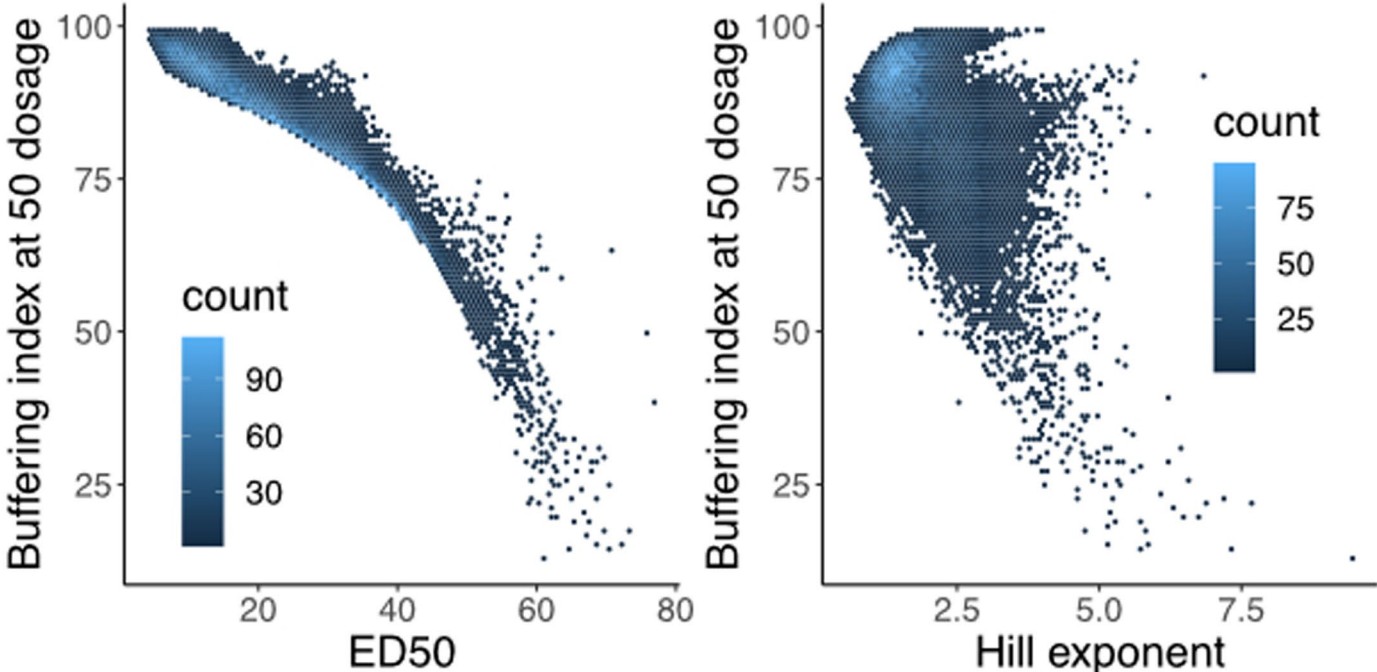

**Extended Data Fig. 3 | Fitted ED$_{50}$ values are well-correlated with orthogonally calculated measures of sensitivity.** For all SOX9-dependent REs with good Hill equation fits (p < 0.05 for both ED$_{50}$ and Hill exponent), correlation between either ED$_{50}$ (left, Spearman ρ −0.961) or Hill exponent (right, Spearman ρ −0.457) and buffering index calculated at 50% SOX9 dosage. See Methods for details on calculation of buffering index – 0 means no buffering (effect of 100 to 50% SOX9 dosage on RE accessibility is 50% of effect of 100 to 0% SOX9 dosage), 100 means full buffering (no effect of 100 to 50% SOX9 dosage on RE accessibility, but substantial effect of 100 to 0% SOX9 dosage).

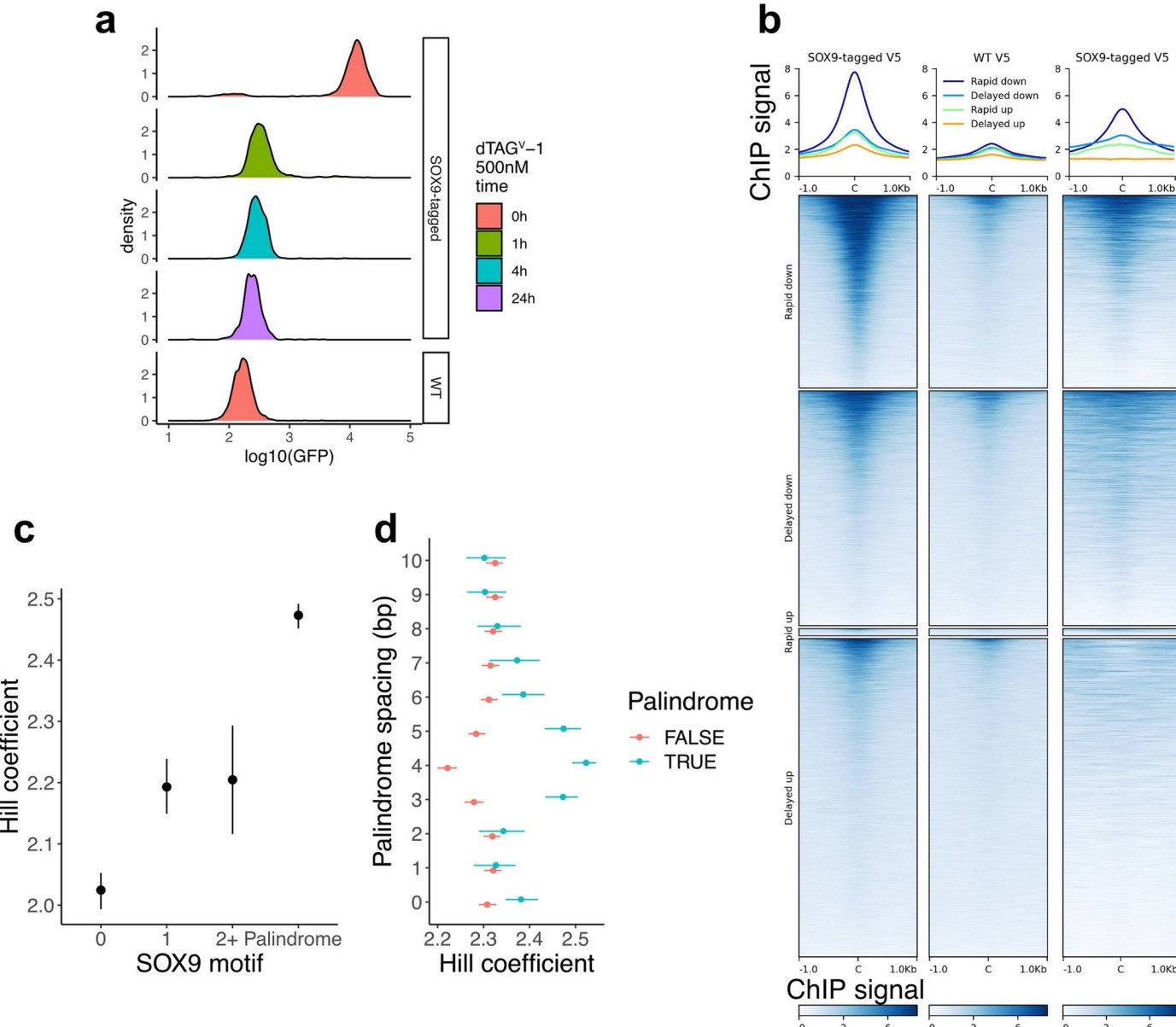

**Extended Data Fig. 4 | Rapidly responding REs are likely direct targets of SOX9.** (**a**) mNeonGreen fluorescence intensity (at least 5,000 cells per histogram) in SOX9-tagged or WT CNCCs with the times of treatment by dTAGV-1. (**b**) V5 ChIP-seq signal from CNCCs with V5-tagged SOX9 present ('SOX9-tagged V5') or absent ('WT') plotted over sets of SOX9-dependent REs as defined in Fig. 3a. (**c**) Hill exponent of rapid down REs stratified by SOX9 motif type, with motif position weight matrices as in Fig. 3e. N of groups from left to right: 2,263,

1,315, 360, 5,221. (**d**) For the SOX9 palindrome motif at with a 0–10 bp spacing between the inverted repeats (y-axis), rapid down REs were stratified on the basis of that motif match. N of blue color groups from bottom to top: 448, 1,249, 1,010, 1,740, 2,628, 1,593, 1,157, 931, 839, 1,249, 1,233. N of red color groups from bottom to top: 7,532, 7,933, 8,200, 7,410, 6,458, 7,568, 8,034, 8,289, 8,371,. Points and error bars represent median and 95% confidence intervals as computed by bootstrap (see Methods).

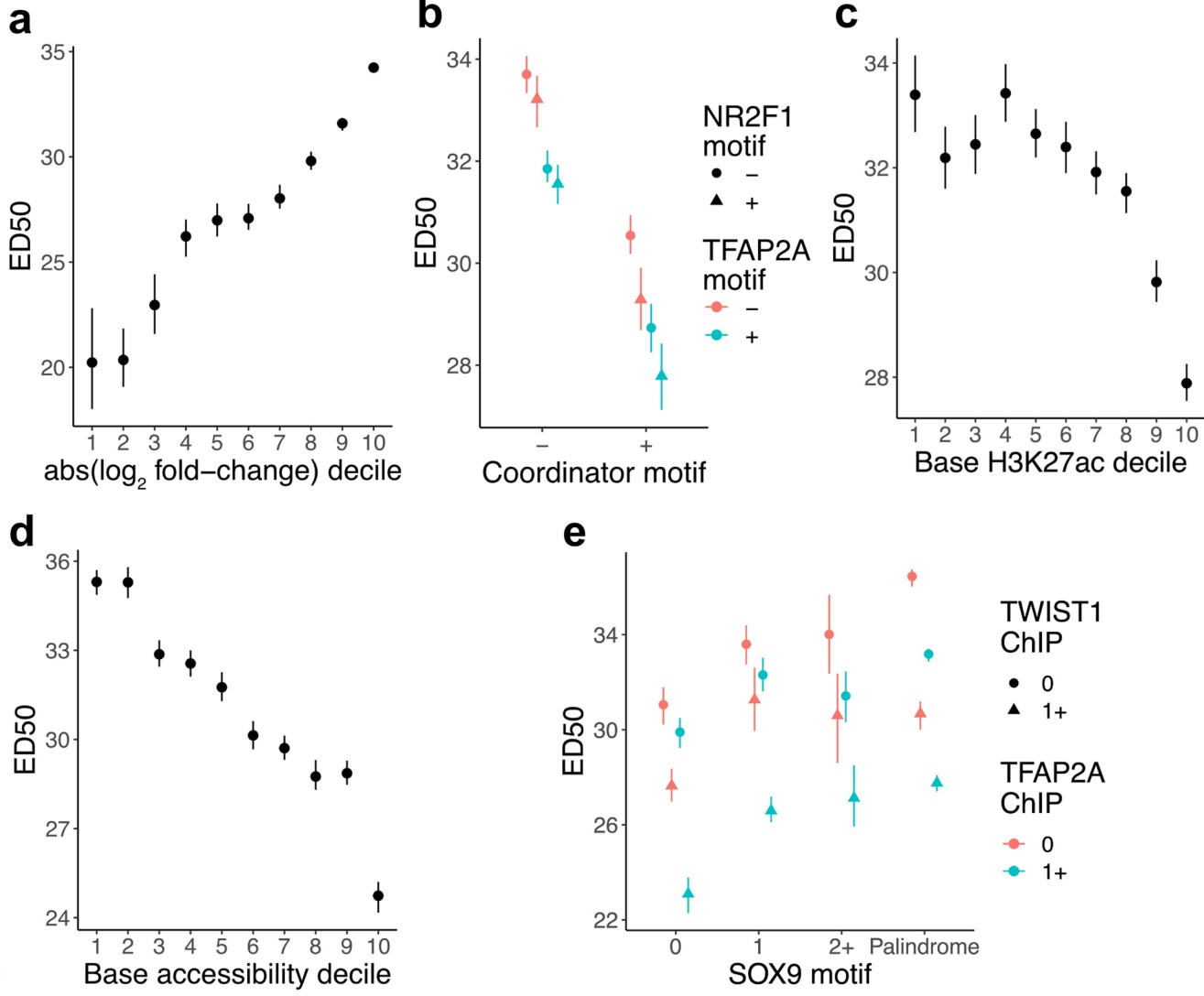

**Extended Data Fig. 5 | Additional features affecting sensitivity of the RE response to SOX9 dosage changes among direct SOX9 targets.** $ED_{50}$ of rapid down SOX9-dependent REs, stratified by (**a**) magnitude of change in response to full SOX9 depletion, (**b**) presence of Coordinator (TWIST1), NR2F1, or TFAP2A sequence motif matches, baseline levels of (**c**) H3K27ac or (**d**) chromatin accessibility, or (**e**) the combination of SOX9 motif type and TWIST1/TFAP2A binding by ChIP-seq. For (a), (c), and (d), higher deciles mean higher values, and N of each decile is 928 (except for deciles 1 and 6 for which N = 927). For (b), N of groups from left to right: 1,539, 879, 1,735, 1,200, 1,693, 718, 952, 563. For (e), N of red circle groups from left to right: 578, 353, 87, 1,603; N of blue circle groups from left to right: 660, 389, 110, 1,712; N of red triangle groups from left to right: 454, 208, 54, 650; N of blue triangle groups from left to right: 691, 375, 107, 1,248. Points and error bars represent median and 95% confidence intervals as computed by bootstrap (see Methods).

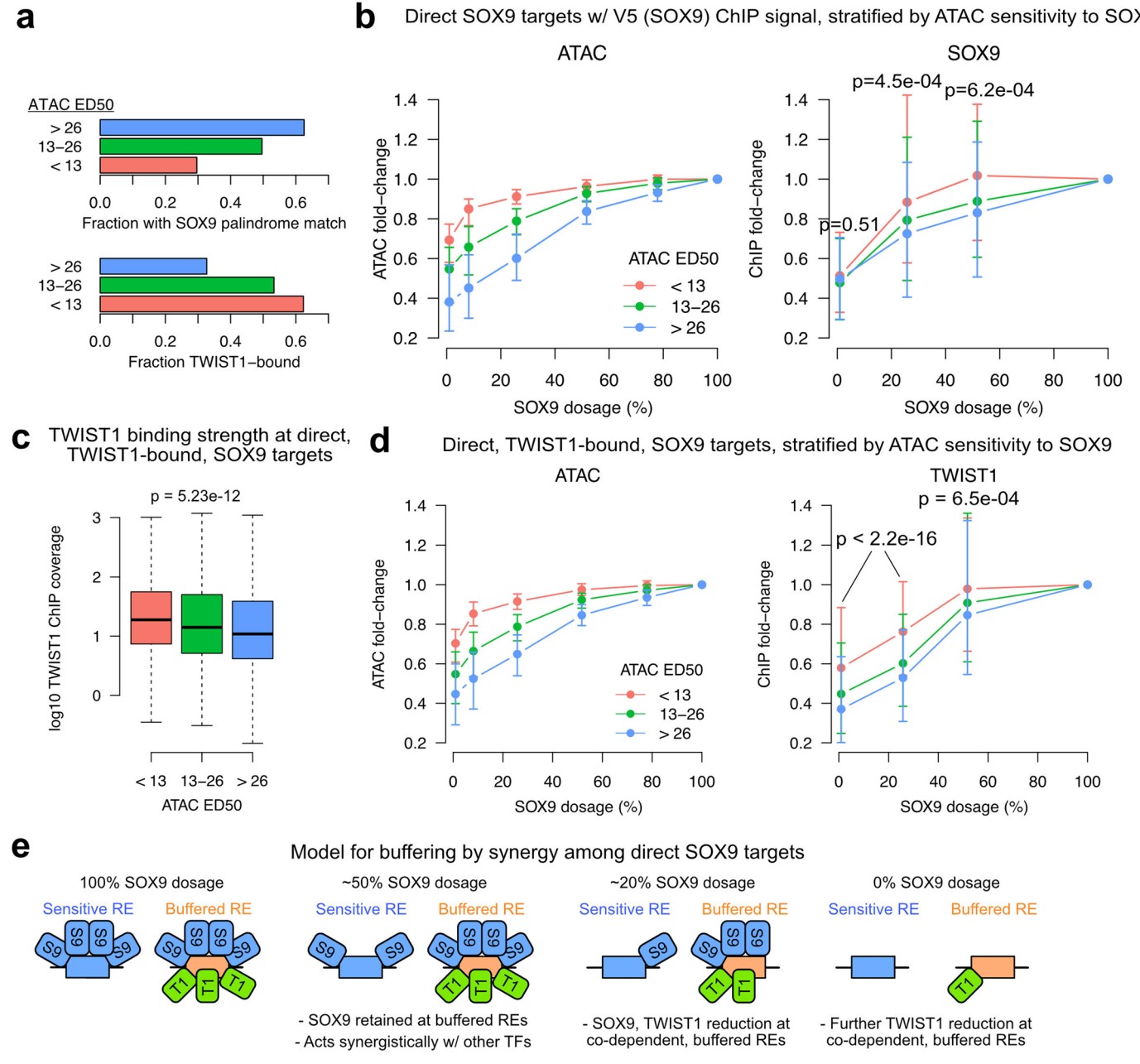

**Extended Data Fig. 6 | Binding of SOX9 and TWIST1 in response to SOX9 dosage changes.** (**a**) Fractions of rapidly downregulated REs with SOX9 palindrome match (top) or TWIST1 binding (bottom), stratified by ATAC-seq sensitivity ($ED_{50}$) to SOX9 dosage (colors). N of groups by color: red, 823; green, 2,473; blue, 5983. (**b**) Distributions of ATAC-seq (left) or SOX9 (V5) ChIP-seq (right) fold-changes vs full SOX9 dosage at each concentration for rapidly downregulated REs with SOX9-dependent ChIP-seq signal (DESeq2 log2FoldChange < 0 in 0 vs 100 SOX9 dosage), stratified based on ATAC-seq sensitivity to SOX9 dosage (colors). N of groups by color: red, 106; green, 391; blue, 666. P-values from two-sided Kruskal-Wallis tests comparing each instance

of three groups. (**c**) Unperturbed TWIST1 ChIP-seq signal at TWIST1-bound, rapidly downregulated REs stratified based on ATAC-seq sensitivity to SOX9 dosage (colors). N of groups by color: red, 513; green, 1318; blue, 1956. Boxplot center represents median, box bounds represent 25th and 75th percentiles, and whiskers represent 5th and 95th percentiles. (**d**) TWIST1 ChIP-seq fold-changes vs full SOX9 dosage at same groups of REs as in (c). In (b) and (d), points and error bars represent median and 25th and 75th percentiles of distribution. (**e**) Models for RE buffering by synergistic TF functions. P-values from two-sided Kruskal-Wallis tests comparing each instance of three groups.

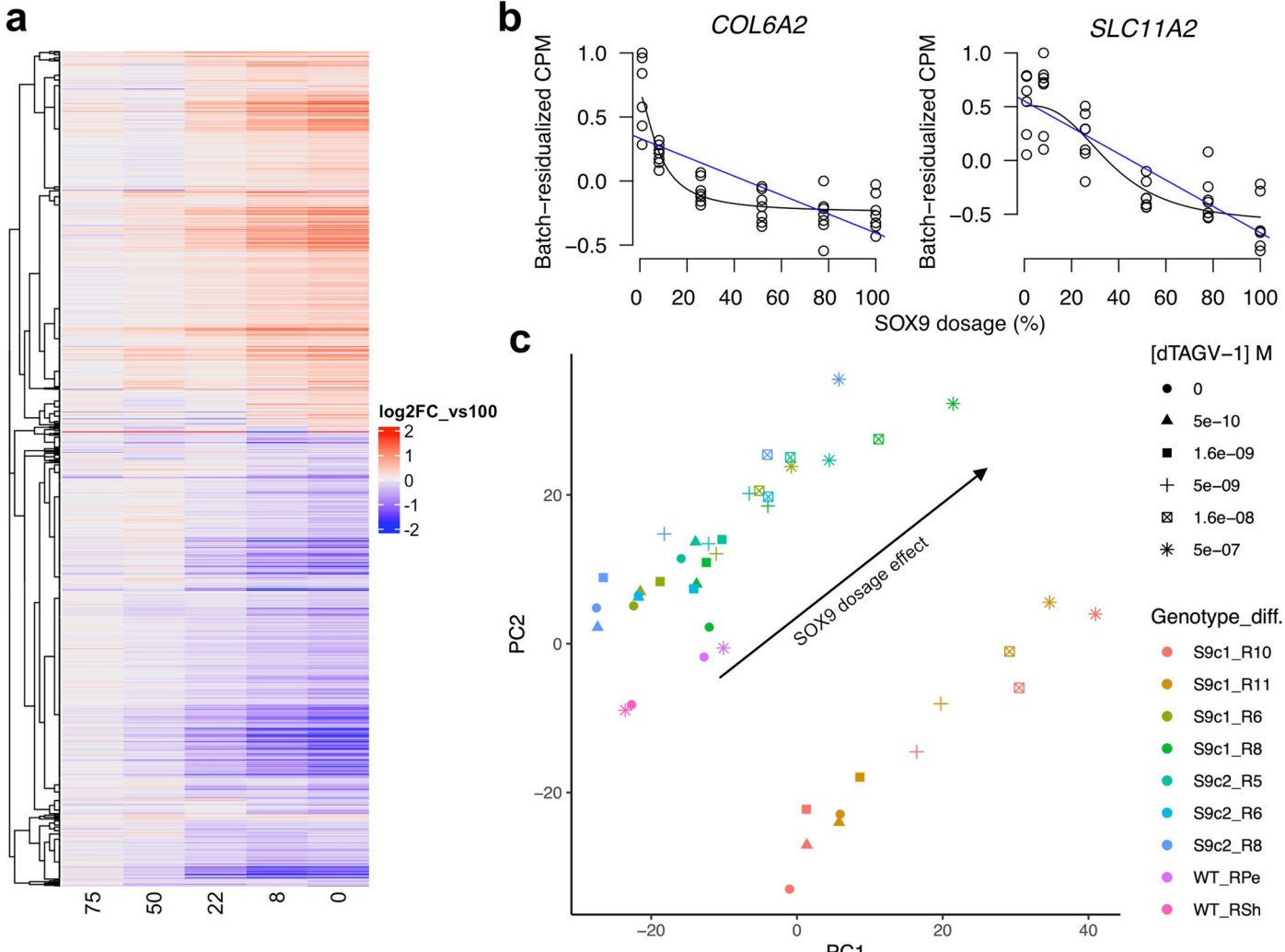

**Extended Data Fig. 7 | Effects of SOX9 dosage changes on gene expression. (a)** Heatmap of fold-changes versus full (100) SOX9 dosage for all all genes for which SOX9 dosage explains a significant (5% FDR) amount of variance (likelihood ratio test), clustered by row based on Kendall distance **(b)** Examples of genes upregulated in response to SOX9 depletion with buffered (left) or sensitive (right) responses. Black and blue lines represent Hill and linear fits, respectively

**(c)** Principal component analysis of RNA-seq counts per million (CPM) across all SOX9-tagged and WT CNCC samples. Shapes indicate the dTAGV-1 concentration treated for 48 h. Colors indicate the combination of hESC line from which CNCCs were derived and differentiation batch (S9c1/2 = SOX9-tagged clone1/2). Arrow indicates the SOX9 dosage effect.

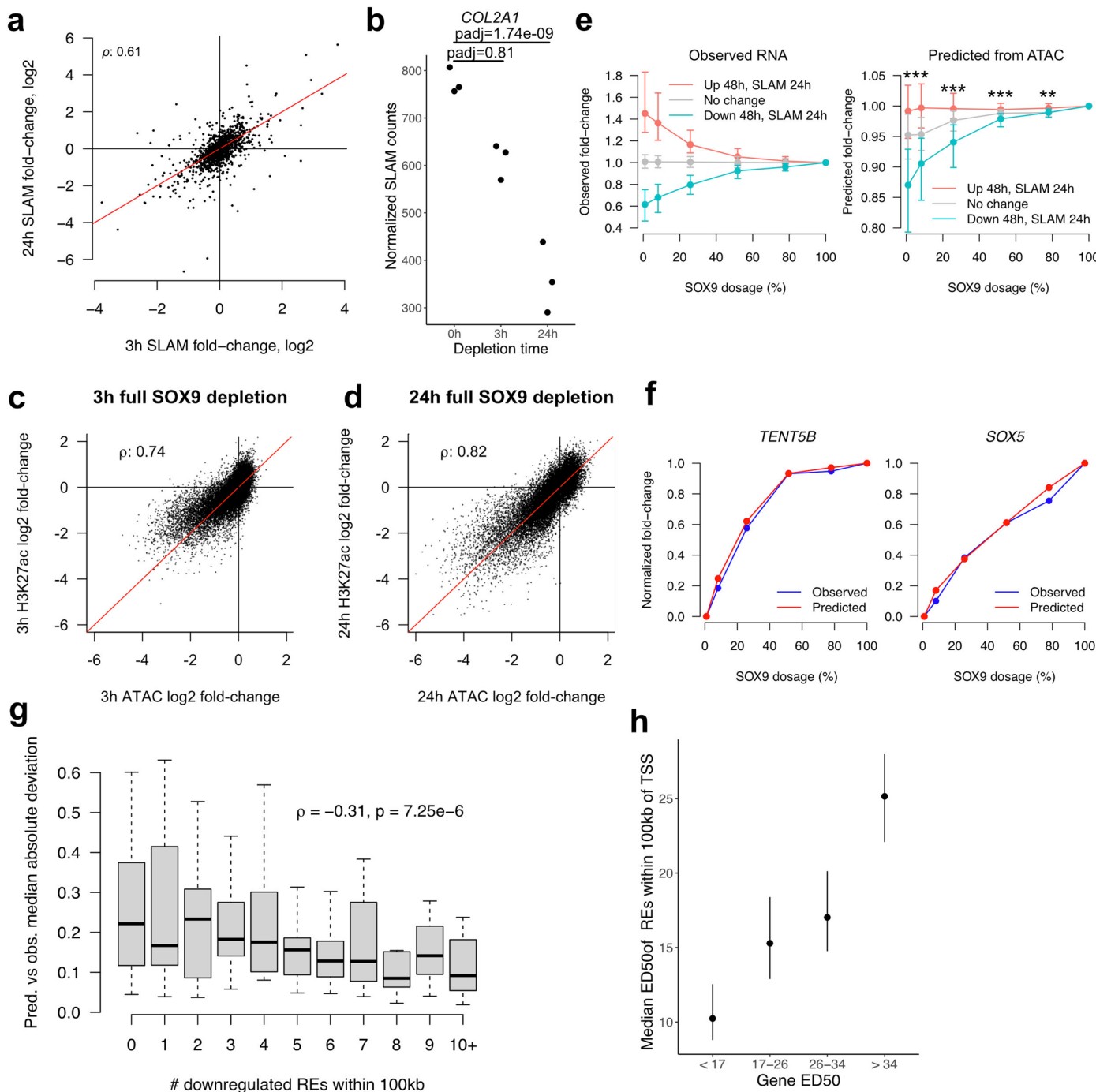

**Extended Data Fig. 8 | Measuring and predicting response of transcriptionally regulated SOX9 target genes.** (**a**) Scatterplot of effects of full SOX9 depletion for 3 h (x-axis) versus 24 h (y-axis) on nascent transcription, as assayed by SLAM-seq, for all SOX9-dependent genes (that is responding to full SOX9 depletion for 48 h in RNA-seq). Y = x line in red. (**b**) Effects of full SOX9 depletion on transcription of COL2A1, a known direct target of SOX9. Points represent SLAM-seq counts from biological replicates, adjusted p-value calculated by DESeq2. (**c, d**) For all REs responding to full depletion of SOX9 at 48 h, the effects of 3 h (c) or 24 h (d) full SOX9 depletion on chromatin accessibility (ATAC-seq, x-axis) or H3K27ac levels (ChIP-seq, y-axis) is plotted. (**e**) Distributions of observed (left) or predicted (right) fold-changes vs full SOX9 dosage at each concentration, stratified based on direction of transcriptional response to full SOX9 depletion (colors). N of groups by color: red, 184; grey, 10,339; blue, 197. Points and error bars represent median and 25th and 75th

percentiles of distribution. ** p = 2.1e-12, *** p < 2.2e-16, two-sided Kruskal-Wallis test comparing the three groups. (**f**) Examples of predictions for a buffered (left) or sensitive (right) gene. (**g**) Median absolute deviation between observed and predicted dosage response curves for transcriptionally downregulated genes, stratified by number of SOX9-downregulated REs within 100 kb of TSS. Spearman rho for correlation and associated two-sided p-value are shown. N of groups from left to right: 227, 238, 173, 162, 105, 77, 55, 35, 32, 26, 49. Boxplot center represents median, box bounds represent 25th and 75th percentiles, and whiskers represent 5th and 95th percentiles. (**h**) Median ED$_{50}$ of REs within 100 kb of the TSS of transcriptionally downregulated genes stratified by sensitivity to SOX9 dosage. N of groups from left to right: 27, 27, 27, 28. Points and error bars represent median and 95% confidence intervals as computed by bootstrap (see Methods). All reported correlation coefficients in this figure are Spearman rho values.

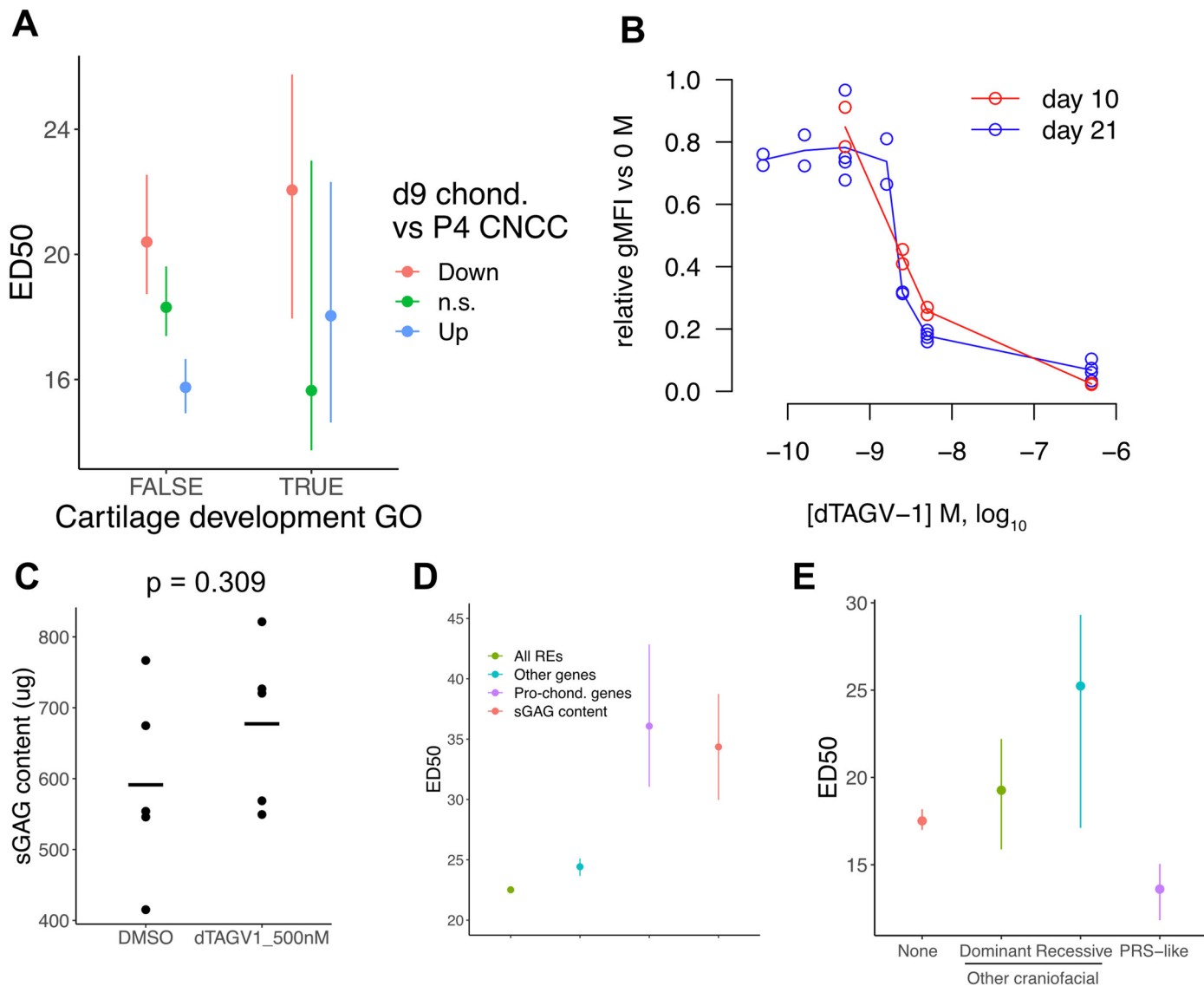

**Extended Data Fig. 9 | Effects of SOX9 dosage on chondrogenesis. (a)** $ED_{50}$ of SOX9-upregulated genes stratified by presence in the 'Cartilage development' Gene Ontology (GO) category (x-axis), and expression change in chondrocytes compared to CNCCs (color, data from Long et al.[29]). N of groups from left to right: 94, 217, 204, 3, 9, 17. Points and error bars in (a,e) represent median and 95% confidence intervals as computed by bootstrap (see Methods). **(b)** Fluorescence intensity at day 10 (red) or 21 (blue) of chondrogenesis in SOX9-tagged chondrocytes as a function of dTAGV-1 concentration. gMFI, geometric mean. **(c)** Sulfated glycosaminoglycan (sGAG, representative of mature cartilage) at day 21 of chondrogenesis in WT CNCCs treated with DMSO or 500 nM dTAGV-1 (N = 5 for each group). Bars represent mean, p-value from two-sided T-test.

**(d)** Fitted $ED_{50}$ values for all SOX9-downregulated REs (N = 20,346) and other genes (N = 688), pro-chondrogenic genes (N = 11), and sGAG content (experiment depicted in Fig. 5d with N = 6 replicates per SOX9 concentration). Points (median) and error bars (95% confidence) for genes and REs computed by bootstrap, $ED_{50}$ point estimate and 95% confidence interval for sGAG content estimated from Hill equation model fit. **(e)** $ED_{50}$ by craniofacial disorder association for genes upregulated upon SOX9 depletion. Gene-craniofacial disorder associations determined as in Fig. 6a. N of groups from left to right: 508, 20, 9, 5. Points and error bars represent median and 95% confidence intervals as computed by bootstrap (see Methods).

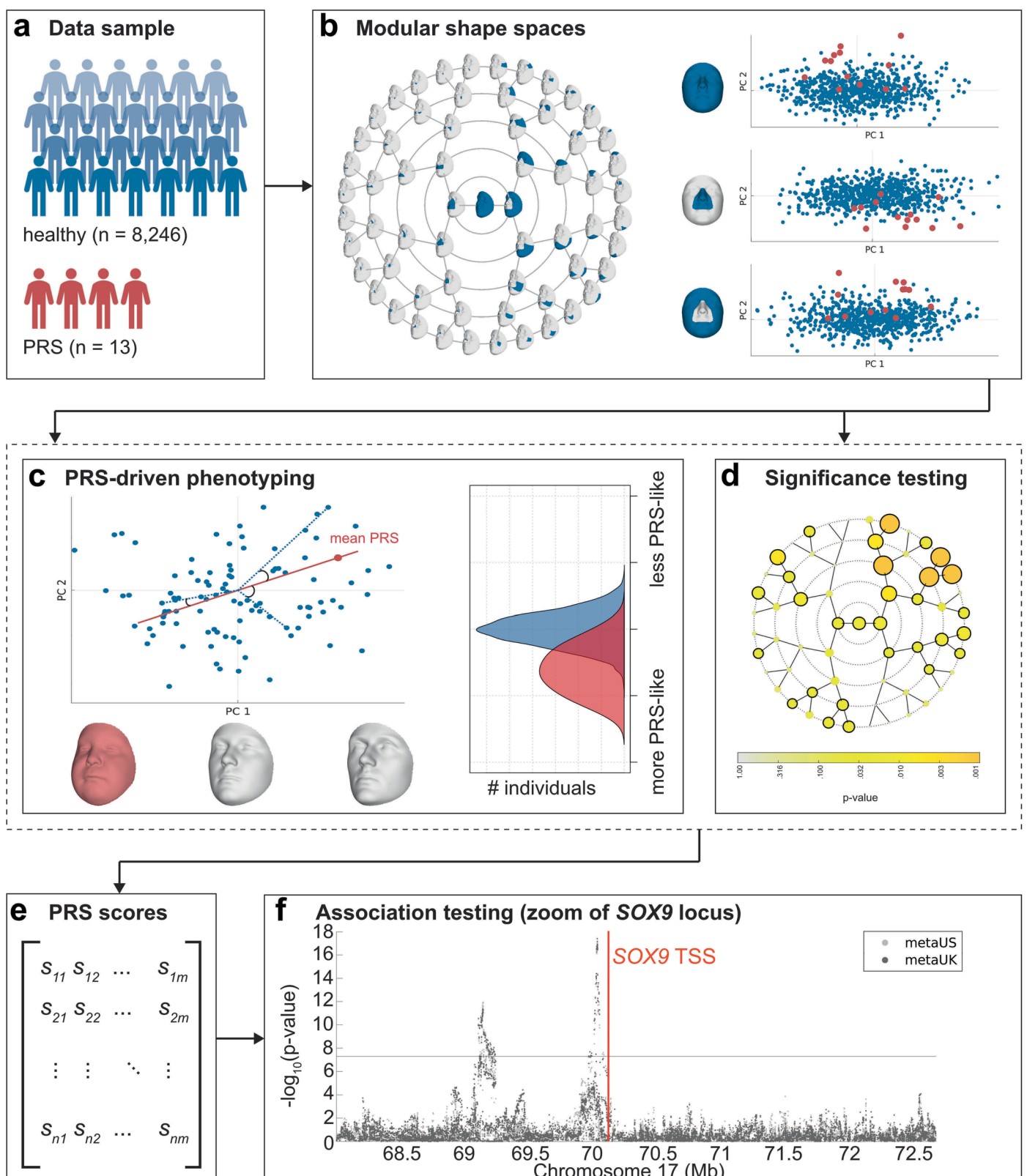

**Extended Data Fig. 10 | See next page for caption.**

**Extended Data Fig. 10 | Endophenotype definition approach and GWAS in healthy individuals.** (**a**) The study sample consisted of 8,246 healthy, unrelated European-ancestry individuals and 13 patients with Pierre Robin Sequence (PRS). (**b**) Global-to-local segmentation of 3D facial shape obtained using hierarchical spectral clustering of the European cohort. For each of the facial segments (n = 63) a shape space is established based on the larger European cohort (blue dots) using PCA, describing the main axes of variation in the data. The PRS facial shapes (red dots) are then aligned and projected onto each segment-derived PCA space. (**c**) Per facial segment, a PRS-driven univariate trait is defined as the vector passing through the global European average facial shape (center) and the PRS average (red dot). Each trait or direction (red line) represents a complex shape transformation that codes for PRS-characteristic facial features, as displayed by the three facial morphs (right = typical PRS face; middle = average face; left = opposite or anti-face). In a leave-one-out approach, each individual was scored on the PRS-driven facial traits by computing the cosine of the angle between the vector going from the global European average to each participant (blue dotted lines), and the vector from the global European average to the average PRS projection (red line). Scores range from 0 to 2, with scores close to 0 indicating the presence of facial features similar to those typically observed in PRS, whereas scores close to 2 correspond to features opposite to PRS. (**d**) To test the significance of the PRS-driven trait in each facial segment, the sample of PRS were compared to a matched control sample of equal size drawn from the larger European cohort using partial least squares regression and a p-value was generated by a 10,000-fold permutation test. In 30 out of 63 facial segments a significant difference (p < =0.05, black encircled segments) was observed between the PRS sample and healthy controls. (**e**) The scores on each of the 30 significant traits were combined into a single phenotype matrix ([8246 ×30]) (**f**) and subsequently tested for genotype-phenotype associations in a multivariate GWAS meta-analysis approach using canonical correlation analyses. Association statistics (y-axis) per SNP (x-axis) are displayed in the Manhattan plot zoomed into the SOX9 locus in two different cohorts (color). Location of the SOX9 transcription start site (TSS) is indicated in red. Horizontal line represent genome-wide significance.

# Reporting Summary

## Statistics

For all statistical analyses, confirm that the following items are present in the figure legend, table legend, main text, or Methods section.

| n/a | Confirmed | |
|---|---|---|
| ☐ | ☒ | The exact sample size (*n*) for each experimental group/condition, given as a discrete number and unit of measurement |
| ☐ | ☒ | A statement on whether measurements were taken from distinct samples or whether the same sample was measured repeatedly |
| ☐ | ☒ | The statistical test(s) used AND whether they are one- or two-sided *Only common tests should be described solely by name; describe more complex techniques in the Methods section.* |
| ☐ | ☒ | A description of all covariates tested |
| ☐ | ☒ | A description of any assumptions or corrections, such as tests of normality and adjustment for multiple comparisons |
| ☐ | ☒ | A full description of the statistical parameters including central tendency (e.g. means) or other basic estimates (e.g. regression coefficient) AND variation (e.g. standard deviation) or associated estimates of uncertainty (e.g. confidence intervals) |
| ☐ | ☒ | For null hypothesis testing, the test statistic (e.g. *F*, *t*, *r*) with confidence intervals, effect sizes, degrees of freedom and *P* value noted *Give P values as exact values whenever suitable.* |
| ☒ | ☐ | For Bayesian analysis, information on the choice of priors and Markov chain Monte Carlo settings |
| ☒ | ☐ | For hierarchical and complex designs, identification of the appropriate level for tests and full reporting of outcomes |
| ☒ | ☐ | Estimates of effect sizes (e.g. Cohen's *d*, Pearson's *r*), indicating how they were calculated |

*Our web collection on statistics for biologists contains articles on many of the points above.*

## Software and code

Policy information about availability of computer code

| Data collection | Flow Cytometry Data Collection: CytExpert Acquisition and Analysis Software v2.4. Flow cytometry analysis: FCSalyzer v0.9.18 |
|---|---|
| Data analysis | Custom code used for analysis of processed sequencing data is available on Zenodo9. KU Leuven provides the MeshMonk (v.0.0.6) spatially dense facial-mapping software, free to use for academic purposes (https://github.com/TheWebMonks/meshmonk)10. Matlab 2017b implementations of the hierarchical spectral clustering to obtain facial segmentations are available on Figshare 11.The following versions of software were used: skewer v0.2.2; bowtie2 v2.4.1; samtools v1.10; deeptools v3.5.0; MACS2 v2.2.7.1; fimo v5.1.1; slamdunk v0.4.3; salmon v1.4.0; |

For manuscripts utilizing custom algorithms or software that are central to the research but not yet described in published literature, software must be made available to editors and reviewers. We strongly encourage code deposition in a community repository (e.g. GitHub). See the Nature Portfolio guidelines for submitting code & software for further information.

# Data

Policy information about availability of data

All manuscripts must include a data availability statement. This statement should provide the following information, where applicable:

- Accession codes, unique identifiers, or web links for publicly available datasets
- A description of any restrictions on data availability
- For clinical datasets or third party data, please ensure that the statement adheres to our policy

The raw sequencing files generated during this study are available on GEO (accession number GSE205904); corresponding processed data is available on Zenodo9. Transcription factor binding motifs were obtained from HOCOMOCO v11 (https://hocomoco11.autosome.org/). Gene ontology assignments were obtained from AmiGO (http://amigo.geneontology.org/amigo). All analyses were done on human genome version hg38, except for PRS endophenotype GWAS (hg19). The raw source data for the facial phenotypes -the 3D facial surface models in.obj format- are available through the FaceBase Consortium (www.facebase.org). Access to these 3D facial surface models requires proper institutional ethics approval and approval from the FaceBase data access committee. Facial scans from PRS patients (used to define the PRS endophenotype) are available through the FaceBase Consortium (https://www.facebase.org FB00000861) under controlled access. The participants making up the US dataset of healthy individuals used for PRS endophenotype GWAS were not collected with broad data sharing consent. Given the highly identifiable nature of both facial and genomic information and unresolved issues regarding risks to participants of inherent reidentification, participants were not consented for inclusion in public repositories or the posting of individual data. This restriction is not because of any personal or commercial interests. Further information about access to the raw 3D facial images and/or genomic data can be obtained from the PSU IRB (IRB-ORP@psu.edu, and the IUPUI IRB (irb@iu.edu). The ALSPAC (UK) data will be made available to bona fide researchers on application to the ALSPAC Executive Committee (http://www.bris.ac.uk/alspac/researchers/dataaccess). Summary statistics from the PRS endophenotype GWAS are available on GWAS Catalog (GCP000517).

# Human research participants

Policy information about studies involving human research participants and Sex and Gender in Research.

| | |
|---|---|
| Reporting on sex and gender | Both males and females were included in both the control sample of 8,246 healthy individuals (60.3% female, remainder male) and the sample of Pierre Robin Sequence (9 female, 4 male). Sex was based on self-reporting |
| Population characteristics | The control sample of healthy individuals comprised three-dimensional facial scans of 8,246 unrelated individuals of European ancestry (60.3% female; median age = 18.0 years, IQR = 9.0 years) originating from the US and the UK. The sample of Pierre Robin Sequence comprised 13 participants (9 female; median age = 12.01 years, IQR = 5.17 years). |
| Recruitment | See White et al, Nature Genetics 2021 for details on recruitment of the control sample of 8,246 healthy individuals. Pierre Robin Sequence individuals were enrolled as part of a larger study following syndromic clinical and/or genetic diagnosis (see https://www.facebase.org/chaise/record/#1/isa:dataset/RID=TJ0) |
| Ethics oversight | Collection of data from PRS patients was carried out with overall approval and oversight of the Colorado Multiple Institutional Review Board (IRB #09–0731), was additionally approved by the institutional review boards of the University of Calgary, Florida State University, the University of California San Francisco, and the Catholic University of Health and Allied Sciences (Mwanza, Tanzania), and was carried out with the approval of the National Institute for Medical Research (Tanzania). Written informed consent was obtained from all study subjects or their parents, as appropriate. No subjects received compensation.

The PRS endophenotype GWAS in this study was conducted on individuals of European ancestry. The conclusions of this GWAS therefore may not be applicable to individuals of other, diverse ancestries. For the PRS endophenotype GWAS conducted in healthy individuals, ethical approval was obtained at each recruitment site and all participants gave their written informed consent prior to participation. For individuals under 18 years of age, written consent was obtained from a parent or legal guardian. For the US sample, the following local ethics approvals were obtained: Pittsburgh, PA (PITT IRB #PRO09060553 and #RB0405013); Seattle, WA (Seattle Children's IRB #12107); Houston, TX (UT Health Committee for the Protection of Human Subjects #HSC-DB-09-0508); Iowa City, IA (University of Iowa Human Subjects Office IRB #200912764 and #200710721); Urbana-Champaign, IL (PSU IRB #13103); New York, NY (PSU IRB #45727); Cincinnati, OH (UC IRB #2015–3073); Twinsburg, OH (PSU IRB #2503); State College, PA (PSU IRB #44929 and #4320); Austin, TX (PSU IRB #44929); San Antonio, TX (PSU IRB #1278); Indianapolis, IN and Twinsburg, OH (IUPUI IRB #1409306349). For the UK sample, ethical approval for the study (Project B2261: "Exploring distinctive facial features and their association with known candidate variants") was obtained from the ALSPAC Ethics and Law Committee and the Local Research Ethics Committees. Informed consent for the use of data collected via questionnaires and clinics was obtained from participants following the recommendations of the ALSPAC Ethics and Law Committee at the time. Consent for biological samples has been collected in accordance with the Human Tissue Act (2004). |

Note that full information on the approval of the study protocol must also be provided in the manuscript.

# Field-specific reporting

Please select the one below that is the best fit for your research. If you are not sure, read the appropriate sections before making your selection.

☒ Life sciences ☐ Behavioural & social sciences ☐ Ecological, evolutionary & environmental sciences

For a reference copy of the document with all sections, see nature.com/documents/nr-reporting-summary-flat.pdf

# Life sciences study design

All studies must disclose on these points even when the disclosure is negative.

| | |
|---|---|
| Sample size | Seven independent biological replicates were used for ATAC-seq and RNA-seq analysis in differing SOX9 concentrations, as a higher degree of accuracy was desired than typical in the field for modeling quantitative changes at individual regulatory elements/genes. Six chondrocyte differentiation replicates were used for sGAG assays, as similar accuracy for detecting quantitative changes was desired. Two independent biological replicates were used for 3h or 24h ATAC and H3K27ac depletion experiments, similar to other studies in the field, where large effects can be easily detected at this sample size. V5 and TWIST1 ChIP-seq were performed with two biological replicates, as the goal was to assess trends in signal across many regulatory elements as a group, rather than individually with high accuracy. Three independent biological replicates were used for 3h or 24h SLAM-seq full depletion experiments due to the lower sequencing depth (and thus likely lower statistical power) expected with nascent RNA sequencing. 2-4 independent biological replicates were used for flow cytometry-based analysis of SOX9 protein levels as a function of dTAG concentration. |
| Data exclusions | Two RNA-seq samples (WT_R8_5e-7M and S9CC47_R6_5e-7M) were identified as extreme outliers in initial principle component analysis. This was confirmed to be due to a library quality issue and so these samples were excluded prior to any further downstream analyses.  When fitting Hill equation or linear models to each RE/gene, individual sample outliers with a z-score greater than 3 for that RE/gene only were removed. <br><br>For PRS endophenotype GWAS, images were excluded if participants were laughing, crying or otherwise emoting or judged to be of poor quality or if the non-rigid registration failed. Participants with missing covariate information (e.g. age, sex) were additionally removed |
| Replication | Western blots, DNA gels, and representative flow cytometry analyses were run twice, independently, with similar results. <br><br>To ensure reproducibility of the fitted ED50 values for each RE/gene, which form the basis of many analyses in the manuscript, a bootstrapping procedure was used which randomly resampled data points at each SOX9 concentration. This was used to construct 95% confidence intervals when comparing ED50 values between groups of genes. The strong enrichment for known SOX9 motifs in both SOX9-dependent and SOX9-bound REs also served as biological validation for those experiments. The presence of TWIST binding motifs from TWIST1 ChIP-seq data served as biological validation. The high correlation between effects of full SOX9 depletion (at 3h or 24h) with full SOX9 depletion at 28h on ATAC-seq signal served as independent validation. The fact that all SNPs specifically affecting the PRS endophenotype had been previously identified and replicated in other studies for different facial phenotypes served as independent validation. The high correlation between the effects of 24h SOX9 depletion of nascent transcription (SLAM-seq) and 48h depletion on mRNA (RNA-seq) served as independent validation. |
| Randomization | Samples were assigned to groups based on treatment conditions and cell line genotype. RNA-seq ATAC-seq and H3K27ac ChIP-seq counts were adjusted for differentiation batch effects using the DESeq2 framework. |
| Blinding | No blinding was done, but all samples were processed and analyzed equally |

# Reporting for specific materials, systems and methods

We require information from authors about some types of materials, experimental systems and methods used in many studies. Here, indicate whether each material, system or method listed is relevant to your study. If you are not sure if a list item applies to your research, read the appropriate section before selecting a response.

## Materials & experimental systems

| n/a | Involved in the study |
|---|---|
| ☐ | ☒ Antibodies |
| ☐ | ☒ Eukaryotic cell lines |
| ☒ | ☐ Palaeontology and archaeology |
| ☒ | ☐ Animals and other organisms |
| ☒ | ☐ Clinical data |
| ☒ | ☐ Dual use research of concern |

## Methods

| n/a | Involved in the study |
|---|---|
| ☐ | ☒ ChIP-seq |
| ☐ | ☒ Flow cytometry |
| ☒ | ☐ MRI-based neuroimaging |

## Antibodies

| | |
|---|---|
| Antibodies used | Abcam rabbit anti-V5 (ab9116, ab15828), ActiveMotif anti-H3K27ac (39133), Abcam mouse anti-TWIST1 (ab50887), Sigma-Aldrich rabbit anti-SOX9 (AB5535); Abcam HRP anti-beta Actin (ab49900) |
| Validation | For rabbit anti-V5, ChIP-seq was performed in cells with present or depleted SOX9, and decreased signal was observed in the SOX9-depleted cells (see Extended Data Figure 4b). For rabbit anti-SOX9, Western blotting of protein lysates was performed in cells with present or depleted SOX9, and no signal was observed in the SOX9-depleted cells (see Figure 1c). See ActiveMotif website (https://www.activemotif.com/catalog/details/39133/histone-h3-acetyl-lys27-antibody-pab) for anti-H3K27ac ChIP-seq validation. See Abcam website (https://www.abcam.com/twist-antibody-twist2c1a-ab50887.html) for anti-TWIST1 ChIP-seq validation. See Abcam website (https://www.abcam.com/hrp-beta-actin-antibody-ac-15-ab49900.html) for anti-beta Actin Western blot validation. |

# Eukaryotic cell lines

Policy information about cell lines and Sex and Gender in Research

| | |
|---|---|
| Cell line source(s) | WA09 (H9) hESCs: Wicell, female.  HEK293FT (R70007): Invitrogen, female. |
| Authentication | H9 hESCs and HEK293FT cells were obtained commercially and validated by their commercial source (WiCell for H9, Invitrogen for HEK293FT) by karyotyping, STR profiling, and marker expression |
| Mycoplasma contamination | All lines tested negative for mycoplasma contamination |
| Commonly misidentified lines (See ICLAC register) | None |

# ChIP-seq

## Data deposition

☒ Confirm that both raw and final processed data have been deposited in a public database such as GEO.

☒ Confirm that you have deposited or provided access to graph files (e.g. BED files) for the called peaks.

| | |
|---|---|
| Data access links *May remain private before publication.* | GEO accession GSE205904: https://www.ncbi.nlm.nih.gov/geo/query/acc.cgi?acc=GSE205904 |
| Files in database submission | S9CC13_R4_dTAG0h_K27ac_1.fastq.gz<br>S9CC13_R4_dTAG0h_K27ac_2.fastq.gz<br>S9CC13_R4_dTAG0h_input_1.fastq.gz<br>S9CC13_R4_dTAG0h_input_2.fastq.gz<br>S9CC13_R4_dTAG24h_K27ac_1.fastq.gz<br>S9CC13_R4_dTAG24h_K27ac_2.fastq.gz<br>S9CC13_R4_dTAG24h_input_1.fastq.gz<br>S9CC13_R4_dTAG24h_input_2.fastq.gz<br>S9CC13_R4_dTAG3h_K27ac_1.fastq.gz<br>S9CC13_R4_dTAG3h_K27ac_2.fastq.gz<br>S9CC13_R4_dTAG3h_input_1.fastq.gz<br>S9CC13_R4_dTAG3h_input_2.fastq.gz<br>S9CC47_R5_dTAG0h_K27ac_1.fastq.gz<br>S9CC47_R5_dTAG0h_K27ac_2.fastq.gz<br>S9CC47_R5_dTAG0h_input_1.fastq.gz<br>S9CC47_R5_dTAG0h_input_2.fastq.gz<br>S9CC47_R5_dTAG24h_K27ac_1.fastq.gz<br>S9CC47_R5_dTAG24h_K27ac_2.fastq.gz<br>S9CC47_R5_dTAG24h_input_1.fastq.gz<br>S9CC47_R5_dTAG24h_input_2.fastq.gz<br>S9CC47_R5_dTAG3h_K27ac_1.fastq.gz<br>S9CC47_R5_dTAG3h_K27ac_2.fastq.gz<br>S9CC47_R5_dTAG3h_input_1.fastq.gz<br>S9CC47_R5_dTAG3h_input_2.fastq.gz<br>S9CC47_R5_untr_V5.bin10.bw.gz<br>S9CC47_R5_untr_V5_1.fastq.gz<br>S9CC47_R5_untr_V5_2.fastq.gz<br>S9CC47_R5_untr_input_1.fastq.gz<br>S9CC47_R5_untr_input_2.fastq.gz<br>WT_R4_dTAG0h_K27ac_1.fastq.gz<br>WT_R4_dTAG0h_K27ac_2.fastq.gz<br>WT_R4_dTAG0h_input_1.fastq.gz<br>WT_R4_dTAG0h_input_2.fastq.gz<br>WT_R4_dTAG24h_K27ac_1.fastq.gz<br>WT_R4_dTAG24h_K27ac_2.fastq.gz<br>WT_R4_dTAG24h_input_1.fastq.gz<br>WT_R4_dTAG24h_input_2.fastq.gz<br>WT_R4_dTAG3h_K27ac_1.fastq.gz<br>WT_R4_dTAG3h_K27ac_2.fastq.gz<br>WT_R4_dTAG3h_input_1.fastq.gz<br>WT_R4_dTAG3h_input_2.fastq.gz<br>WT_R5_dTAG0h_K27ac_1.fastq.gz<br>WT_R5_dTAG0h_K27ac_2.fastq.gz<br>WT_R5_dTAG0h_input_1.fastq.gz<br>WT_R5_dTAG0h_input_2.fastq.gz<br>WT_R5_dTAG24h_K27ac_1.fastq.gz<br>WT_R5_dTAG24h_K27ac_2.fastq.gz<br>WT_R5_dTAG24h_input_1.fastq.gz |

```
WT_R5_dTAG24h_input_2.fastq.gz
WT_R5_dTAG3h_K27ac_1.fastq.gz
WT_R5_dTAG3h_K27ac_2.fastq.gz
WT_R5_dTAG3h_input_1.fastq.gz
WT_R5_dTAG3h_input_2.fastq.gz
WT_R5_untr_V5.bin10.bw.gz
WT_R5_untr_V5_1.fastq.gz
WT_R5_untr_V5_2.fastq.gz
WT_R5_untr_input_1.fastq.gz
WT_R5_untr_input_2.fastq.gz
all.sub.150bpclust.greater2.500bp.merge.k27ac.txt.gz
C13R18_0nM_V5_1.fq.gz
C13R18_0nM_V5_2.fq.gz
C13R18_0nM_in_1.fq.gz
C13R18_0nM_in_2.fq.gz
C13R18_1p6nM_V5_1.fq.gz
C13R18_1p6nM_V5_2.fq.gz
C13R18_1p6nM_in_1.fq.gz
C13R18_1p6nM_in_2.fq.gz
C13R18_500nM_V5_1.fq.gz
C13R18_500nM_V5_2.fq.gz
C13R18_500nM_in_1.fq.gz
C13R18_500nM_in_2.fq.gz
C13R18_5nM_V5_1.fq.gz
C13R18_5nM_V5_2.fq.gz
C13R18_5nM_in_1.fq.gz
C13R18_5nM_in_2.fq.gz
C13R21_0nM_V5_1.fq.gz
C13R21_0nM_V5_2.fq.gz
C13R21_0nM_in_1.fq.gz
C13R21_0nM_in_2.fq.gz
C13R21_1p6nM_V5_1.fq.gz
C13R21_1p6nM_V5_2.fq.gz
C13R21_1p6nM_in_1.fq.gz
C13R21_1p6nM_in_2.fq.gz
C13R21_500nM_V5_1.fq.gz
C13R21_500nM_V5_2.fq.gz
C13R21_500nM_in_1.fq.gz
C13R21_500nM_in_2.fq.gz
C13R21_5nM_V5_1.fq.gz
C13R21_5nM_V5_2.fq.gz
C13R21_5nM_in_1.fq.gz
C13R21_5nM_in_2.fq.gz
S9C13_R18_0nM_AP_1.fq.gz
S9C13_R18_0nM_AP_2.fq.gz
S9C13_R18_0nM_T1_1.fq.gz
S9C13_R18_0nM_T1_2.fq.gz
S9C13_R18_0nM_in_1.fq.gz
S9C13_R18_0nM_in_2.fq.gz
S9C13_R18_1p6nM_AP_1.fq.gz
S9C13_R18_1p6nM_AP_2.fq.gz
S9C13_R18_1p6nM_T1_1.fq.gz
S9C13_R18_1p6nM_T1_2.fq.gz
S9C13_R18_1p6nM_in_1.fq.gz
S9C13_R18_1p6nM_in_2.fq.gz
S9C13_R18_500nM_AP_1.fq.gz
S9C13_R18_500nM_AP_2.fq.gz
S9C13_R18_500nM_T1_1.fq.gz
S9C13_R18_500nM_T1_2.fq.gz
S9C13_R18_500nM_in_1.fq.gz
S9C13_R18_500nM_in_2.fq.gz
S9C13_R18_5nM_AP_1.fq.gz
S9C13_R18_5nM_AP_2.fq.gz
S9C13_R18_5nM_T1_1.fq.gz
S9C13_R18_5nM_T1_2.fq.gz
S9C13_R18_5nM_in_1.fq.gz
S9C13_R18_5nM_in_2.fq.gz
S9C13_R21_0nM_AP_1.fq.gz
S9C13_R21_0nM_AP_2.fq.gz
S9C13_R21_0nM_T1_1.fq.gz
S9C13_R21_0nM_T1_2.fq.gz
S9C13_R21_0nM_in_1.fq.gz
S9C13_R21_0nM_in_2.fq.gz
S9C13_R21_1p6nM_AP_1.fq.gz
S9C13_R21_1p6nM_AP_2.fq.gz
S9C13_R21_1p6nM_T1_1.fq.gz
```

```
S9C13_R21_1p6nM_T1_2.fq.gz
S9C13_R21_1p6nM_in_1.fq.gz
S9C13_R21_1p6nM_in_2.fq.gz
S9C13_R21_500nM_AP_1.fq.gz
S9C13_R21_500nM_AP_2.fq.gz
S9C13_R21_500nM_T1_1.fq.gz
S9C13_R21_500nM_T1_2.fq.gz
S9C13_R21_500nM_in_1.fq.gz
S9C13_R21_500nM_in_2.fq.gz
S9C13_R21_5nM_AP_1.fq.gz
S9C13_R21_5nM_AP_2.fq.gz
S9C13_R21_5nM_T1_1.fq.gz
S9C13_R21_5nM_T1_2.fq.gz
S9C13_R21_5nM_in_1.fq.gz
S9C13_R21_5nM_in_2.fq.gz
all.sub.150bpclust.greater2.500bp.merge.SOX9titr.TWIST1.TFAP2A.in.counts.txt.gz
all.sub.150bpclust.greater2.500bp.merge.SOX9titr.V5.counts.txt.gz
S9CC47_R18_0h_V5_1.fq.gz
S9CC47_R18_0h_V5_2.fq.gz
S9CC47_R18_0h_in_1.fq.gz
S9CC47_R18_0h_in_2.fq.gz
S9CC47_R18_3h_V5_1.fq.gz
S9CC47_R18_3h_V5_2.fq.gz
S9CC47_R18_3h_in_1.fq.gz
S9CC47_R18_3h_in_2.fq.gz
S9CC47_R18_0h_V5.bin10.bw
S9CC47_R18_3h_V5.bin10.bw
```

Genome browser session
(e.g. UCSC)

*Provide a link to an anonymized genome browser session for "Initial submission" and "Revised version" documents only, to enable peer review. Write "no longer applicable" for "Final submission" documents.*

## Methodology

Replicates

Two CNCC differentiation replicates per hESC line

Sequencing depth

```
Sample #total reads, #uniquely mapped reads, paired-end
S9CC13_R4_dTAG0h_K27ac 19702909 17573660 Paired-end
S9CC13_R4_dTAG0h_input 17171513 14804257 Paired-end
S9CC13_R4_dTAG24h_K27ac 22811174 20455584 Paired-end
S9CC13_R4_dTAG24h_input 4398526 3803373 Paired-end
S9CC13_R4_dTAG3h_K27ac 14035485 12632483 Paired-end
S9CC13_R4_dTAG3h_input 37150935 32120133 Paired-end
S9CC47_R5_dTAG0h_K27ac 16115252 14376813 Paired-end
S9CC47_R5_dTAG0h_input 17131811 14687275 Paired-end
S9CC47_R5_dTAG24h_K27ac 16826112 15029996 Paired-end
S9CC47_R5_dTAG24h_input 16575684 14201610 Paired-end
S9CC47_R5_dTAG3h_K27ac 20208835 17909269 Paired-end
S9CC47_R5_dTAG3h_input 16650520 14398709 Paired-end
WT_R4_dTAG0h_K27ac 23949200 21492733 Paired-end
WT_R4_dTAG0h_input 20173375 17391911 Paired-end
WT_R4_dTAG24h_K27ac 15887196 14220467 Paired-end
WT_R4_dTAG24h_input 16342587 14070696 Paired-end
WT_R4_dTAG3h_K27ac 19413998 17457160 Paired-end
WT_R4_dTAG3h_input 17294971 14924313 Paired-end
WT_R5_dTAG0h_K27ac 16775538 14991098 Paired-end
WT_R5_dTAG0h_input 16300130 14025931 Paired-end
WT_R5_dTAG24h_K27ac 18299418 16349153 Paired-end
WT_R5_dTAG24h_input 18136091 15475742 Paired-end
WT_R5_dTAG3h_K27ac 18473353 16354827 Paired-end
WT_R5_dTAG3h_input 19514872 16736835 Paired-end
S9C13_R18_0nM_AP 5092567 3371626 Paired-end
S9C13_R18_0nM_T1 13220990 10497414 Paired-end
S9C13_R18_0nM_in 10095191 4074233 Paired-end
S9C13_R18_1p6nM_AP 10689895 7529710 Paired-end
S9C13_R18_1p6nM_T1 19014598 15056178 Paired-end
S9C13_R18_1p6nM_in 8996062 3907893 Paired-end
S9C13_R18_500nM_AP 8297763 5490568 Paired-end
S9C13_R18_500nM_T1 12576772 9757678 Paired-end
S9C13_R18_500nM_in 12139794 4744689 Paired-end
S9C13_R18_5nM_AP 3898204 2683774 Paired-end
S9C13_R18_5nM_T1 9432451 7075177 Paired-end
S9C13_R18_5nM_in 11954385 4772522 Paired-end
S9C13_R21_0nM_AP 5621370 3323295 Paired-end
S9C13_R21_0nM_T1 6117521 4051422 Paired-end
S9C13_R21_0nM_in 13494864 5921549 Paired-end
```

S9C13_R21_1p6nM_AP 18776354 14814544 Paired-end
S9C13_R21_1p6nM_T1 3332709 2023755 Paired-end
S9C13_R21_1p6nM_in 10455700 4827845 Paired-end
S9C13_R21_500nM_AP 11196045 8037365 Paired-end
S9C13_R21_500nM_T1 6325174 4234446 Paired-end
S9C13_R21_500nM_in 9677671 4044655 Paired-end
S9C13_R21_5nM_AP 12200307 9322397 Paired-end
S9C13_R21_5nM_T1 8421010 6139030 Paired-end
S9C13_R21_5nM_in 10498564 5010970 Paired-end
C13R18_0nM_V5 8596070 4080234 Paired-end
C13R18_0nM_in 5190485 1500706 Paired-end
C13R18_1p6nM_V5 8302768 4014248 Paired-end
C13R18_1p6nM_in 5668285 1664329 Paired-end
C13R18_500nM_V5 7220039 3521834 Paired-end
C13R18_500nM_in 5122178 1379264 Paired-end
C13R18_5nM_V5 7585309 3988019 Paired-end
C13R18_5nM_in 4447450 1246852 Paired-end
C13R21_0nM_V5 6675128 3692716 Paired-end
C13R21_0nM_in 7359899 2048934 Paired-end
C13R21_1p6nM_V5 7023127 2870076 Paired-end
C13R21_1p6nM_in 5679400 1588508 Paired-end
C13R21_500nM_V5 8901195 4456518 Paired-end
C13R21_500nM_in 6418679 1712338 Paired-end
C13R21_5nM_V5 6984077 3624310 Paired-end
C13R21_5nM_in 4773704 1344702 Paired-end

| | |
|---|---|
| Antibodies | Abcam rabbit anti-V5 (ab9116, ab15828), ActiveMotif anti-H3K27ac (39133), Abcam mouse anti-TWIST1 (ab50887) |
| Peak calling parameters | For TWIST1 ChIP peaks were called using MACS2 with default parameters; In other cases ChIP counts were calculated over reproducible ATAC-seq peaks as described in methods |
| Data quality | Effects of SOX9 depletion on H3K27ac ChIP-seq signal were observed to be highly correlated with effects on ATAC-seq signal at the same regulatory elements (Spearman rho 0.74 for 3h depletion, 0.82 for 24h depletion). |
| Software | skewer v0.2.2; bowtie2 v2.4.1; samtools v1.10; deeptools v3.5.0; MACS2 v2.2.7.1; processed data and code is available on Zenodo (10.5281/zenodo.6596465) |

# Flow Cytometry

## Plots

Confirm that:

☒ The axis labels state the marker and fluorochrome used (e.g. CD4-FITC).

☒ The axis scales are clearly visible. Include numbers along axes only for bottom left plot of group (a 'group' is an analysis of identical markers).

☒ All plots are contour plots with outliers or pseudocolor plots.

☒ A numerical value for number of cells or percentage (with statistics) is provided.

## Methodology

| | |
|---|---|
| Sample preparation | CNCCs were harvested for flow cytometry using accutase and quenching with FACS buffer (5% FBS in PBS). Chondrocytes were harvested as described previously with the following modifications. Chondrocytes were incubated in digestion medium for ~1hr with gentle agitation every 15 min. Digestion medium: DMEM-KO, 1mg/mL Pronase (Roche, 11459643001), 1mg/mL Collagenase B (Roche, 11088815001), 4U/mL Hyalauronidase (Sigma, H3506-500mg). Digested cells were then washed twice in PBS |
| Instrument | Beckman Coulter Cytoflex V2-B3-R2 |
| Software | Collection: CytExpert Acquisition and Analysis Software v2.4 Analysis: FCSalyzer v0.9.18 |
| Cell population abundance | Between 80% (CNCCs) and 50% (chondrocytes) passed the FSC/SSC gating used for analyzing live cells. Cells were not sorted, only analyzed for FITC signal (representing mNeonGreen fluorescence) |
| Gating strategy | Cells were gated based on FSC and SSC to select viable cells with known size and scatter properties as previously observed in CNCCs and chondrocytes, which was also shown to select viable cells in preliminary studies with 7-AAD viability staining. The FSC/SSC-gated cells were then analyzed for FITC signal (representing mNeonGreen fluorescence) |

☒ Tick this box to confirm that a figure exemplifying the gating strategy is provided in the Supplementary Information.

