## [Peer Review File · Nature Genetics]

Peer Review Information

Manuscript Title: Precise modulation of transcription factor levels identifies features underlying dosage sensitivity

Corresponding author name(s): Professor Joanna Wysocka, Professor Jonathan Pritchard

Reviewer Comments & Decisions:

Decision Letter, initial version:
--

1st Aug 2022

Dear Joanna,

Your Article, "Precise modulation of transcription factor levels reveals drivers of dosage sensitivity" has now been seen by 3 referees. You will see from their comments below that while they find your work of interest, some important points are raised. We are interested in the possibility of publishing your study in Nature Genetics, but would like to consider your response to these concerns in the form of a revised manuscript before we make a final decision on publication.

The three reviewers have submitted engaged reports that appreciate this work's novelty.

Reviewer #1 is the most supportive, and does not provide much specific guidance for improvement. Their major request is that you use the system to further explore the activity of other TFs under varying SOX9 dosage.

Reviewer #2 is similarly strongly positive. They provide a list of specific requests, of which the two most important were for further SOX9 ChIPseq data to bolster your modelling; and (see also Reviewer #1) further mechanistic understanding of buffering.

Reviewer #3, conversely, sounds more skeptical, but also seems open to considering a revision. They have some important criticisms about the methodological bases of your work, but they seem largely addressable.

In our overall reading of these reviews, we think there is a clear path to a potential publication. We think Reviewer #3's requests would strengthen the methodological bedrock of your study, and encourage you to comprehensively address them. The suggestion from Reviewers #1 and #2 to use

this system to identify further buffering mechanisms is also a very thoughtful one that could, potentially, substantially improve the novelty of your work (which we note Reviewer #3 sounds least enthusiastic for, at this stage).

To guide the scope of the revisions, the editors discuss the referee reports in detail within the team, including with the chief editor, with a view to identifying key priorities that should be addressed in revision and sometimes overruling referee requests that are deemed beyond the scope of the current study. We hope that you will find the prioritized set of referee points to be useful when revising your study. Please do not hesitate to get in touch if you would like to discuss these issues further.

We therefore invite you to revise your manuscript taking into account all reviewer and editor comments. Please highlight all changes in the manuscript text file. At this stage we will need you to upload a copy of the manuscript in MS Word .docx or similar editable format.

*2) If you have not done so already please begin to revise your manuscript so that it conforms to our Article format instructions, available [here](http://www.nature.com/ng/authors/article_types/index.html). Refer also to any guidelines provided in this letter.

[redacted]

We hope to receive your revised manuscript within four to eight weeks. If you cannot send it within this time, please let us know.

Sincerely,

Michael Fletcher, PhD
Senior Editor, Nature Genetics

ORCID: 0000-0003-1589-7087

Referee expertise:

Referee #1: development, neural crest

Referee #2: transcription factors, genomics

Referee #3: genomics, gene expression regulation

Reviewers' Comments:

Reviewer #1:

Remarks to the Author:

In this study, Naqvi et al. investigated the effects of dosing the levels of critical developmental and neural crest regulator, Sox9. They analysed the impact of such dosage on chromatin, gene expression and cellular and developmental phenotype level using an in vitro model of human cranial neural crest development. The study is carefully executed, appropriately controlled and brings new insight not only into the effects of the necessary dosage of Sox9 factor but also reveals critical effects that vary from the buffered response on chromatin and expression levels to profound consequences on a specific subset of heightened sensitivity regulatory elements genes, associated with Pierre Robin Sequence-like phenotypes and pro-chondrogenic programme. The authors propose a model that integrates genes/elements sensitive to Sox9 dosage, resulting in specific cellular and morphological effects. Although findings are not completely surprising, this study, for the first time, quantifies and interprets the impact of fine-tuning a critical developmental transcription factor at the mechanistic level, and as

such, this study is of high value.

The authors also assess the effect of other critical TFs on regulatory elements in conditions of varying SOX9 dosage. This is another crucial aspect of the study, as it investigates the competitive model of activity of other TFs substantially that would bind to regulatory elements and further decrease sensitivity/dependence on Sox9. This system and data offer an excellent model to investigate the dynamics of neural; crest regulatory cores, both by looking and cooperative as well as competitive/compensating action. Unfortunately, this latter aspect was insufficiently explored in the study, which is a pity, as such analysis could contribute novel mechanistic insights into the nature of neural crest/craniofacial gene regulatory cores.

Minor points:

1. GWAS identified for multiple essential genes – discussed only for Sox9 locus – can those be detailed for *Sfrp2*, *Dlx5/6*, *Alx1* etc.?
2. It is not evident if the authors have evidence of synergistic binding in the condition of low Sox or if the only modus operandi is competition. Could this please be detailed?

Reviewer #2:

Remarks to the Author:

The study by Naqvi et al examined the quantitative impact of TF dosage on chromatin accessibility and gene expression using mathematical modelling of CNCC ATAC-seq, RNA-seq and new and publicly available TF ChIP-seq datasets. They showed that genes relating to chondrogenesis were particularly sensitive to SOX9 dosage and validated this in an in vitro model of chondrocyte differentiation. They further correlated the impact of these buffered or sensitive genes to normal craniofacial variability and disease manifestations, showing that at both ends of the buffered-sensitive gene spectrum there were high impact phenotypes.

The experimental design from basic transcription factor biology, to development, and correlation with disease phenotypes is excellent. The conclusions drawn are supported by the data shown and interpretations are mostly well explained. While the study's insights are novel for SOX9 and the graded dosage of SOX9 allows for fine separation of buffered v, sensitive genes, the mechanisms for why some genes are buffered or sensitive to TF dosage could be expanded upon. Fortunately, the authors have the tools in hand to be able to do this.

The novelty of the study would be strengthened by additional SOX9 ChIP data The inclusion of V5-ChIP-seq in cells with reduced SOX9 dosage would validate much of the mathematical modelling in Figure 2 and 3. It would also be a significant piece of data for the TF biology field, as the impact of partially reduced TF dosage, ie heterozygous loss, on DNA binding/occupancy has not been experimentally shown. Are the REs with palindrome motifs the first to lose SOX9 occupancy? This data should be integrated with the chromatin accessibility and RNA-seq results to bolster why some REs/genes are sensitive and others are buffered.

Other comments:

Is there heterogeneity within the CNCC populations used in this study?

It's not exactly clear how Hill exponent gives requirement for SOX9 binding as currently written in the text and relating to Extended data 4C,D "The SOX9 palindrome with 3-5 bp spacing was also specifically associated with a modest increase in the Hill exponent, consistent with previous reports showing its requirement for cooperative SOX9 binding (page 6)" – could the biological relevance of this model be clarified in the text accompanying Figure 2 and the initial descriptions of the Hill and ED50 models.

Could the proportion of sensitive v buffered REs and whether they increase or decrease in accessibility be stated more clearly in a figure or text. Might be 74% buffered, but text description doesn't make that clear (page 5) – the details of the mathematical modelling in the text distract from the important conclusions of Figure 2.

Could the authors speculate on the mechanisms by which other TF binding would change the sensitivity to SOX9 dosage?

Are the changes in gene expression monotonic like chromatin accessibility? A heatmap for gene expression changes, and supplementary table of the list of genes and REs that are buffered v. sensitive would be beneficial to readers. In addition to cartilage development (Fig 5A), what other pathways show sensitivity to SOX9 dosage?

What is the overlap between buffered v. sensitive genes from RNA-seq and RE closest genes from ATAC-seq?

Reviewer #3:

Remarks to the Author:

The manuscript submitted by Naqvi et al. presents an elegant method to study the dosage sensitivity of cells to transcription factor levels. The authors engineer cranial neural crest cells (CNCCs) so that SOX9 levels can be titrated down by the addition of a drug. They then measure the effects of decreasing SOX9 levels on the DNA accessibility of regulatory elements (REs) by ATAC-seq, on mRNA expression levels by RNA-seq, and on the in vitro differentiation of CNCCs. The main claims of the manuscript are 1) that the sequence properties of REs make them either sensitive or buffered to decreases in Sox9, 2) that the sensitivity of REs causes sensitivity in the expression of downstream mRNAs, and 3) that the craniofacial defects that result from decreases in Sox9 are caused by a subset of sensitive pro-chondrogenic genes. As written, the data do not strongly support these claims.

The claims rely heavily on the definition of sensitivity, which is defined as the empirical dose 50 (ED50) 'representing the SOX9 dosage at which the RE reaches half of its maximal levels' (page 5). However, the Hill equation in Figure 2C seems to contradict the subsequent figures. If the Hill equation in Figure 2C is correct, then as x (SOX9 concentration) increases y (ATAC CPM values) should decrease, which is counterintuitive and opposite of what is shown in Figure 2E and similar figures throughout the manuscript. The solid and dashed lines in Figure 2D should be swapped if the Hill equation in Figure 2C is correct. The authors should clarify the Hill equation in Figure 2C. If the equation in Figure 2C is not a typo, then the interpretation of the subsequent results would be compromised.

Some clarification should also be given as to how the Hill equation was fit to data. Specifically, were all REs fit using the Hill, or were some fit with the linear model? If both the linear model and the Hill were used, is it justified to compare ED50s fit from different models?. It is also unclear how the Hill equation with 3 parameters (mentioned in the Methods) was fitted, and for which REs it was used. It is again unclear whether the REs fitted with different numbers of parameters should be directly compared with each other.

Figure 4E is used to support the claim that the dose-response curves of REs predicts the dose-response of their target genes. However, the model in Figure 4E does not distinguish strongly between high and low ED50 genes (right panel of Figure 4E) and fails to capture the qualitative difference in curve shape between genes with high vs low ED50s. Subsequent analyses of mRNA ED50s (and the final model) assume a strong relationship between RE and gene ED50s despite Figure 4E. The final model presented in the paper explicitly predicts a correlation between the ED50s of REs and their target genes ("Genes with nearby sensitive REs will themselves show more sensitive responses to SOX9 dosage" (Page 11)) but this correlation is not shown.

The final claim of the paper is that pro-chondrogenic genes are particularly sensitive to SOX9 depletion, and that their sensitivity explains the craniofacial-related phenotypic sensitivity to SOX9. The result in Figure 5E is intriguing and shows that the dose-response of a small set of sensitive genes closely follows the response of the differentiation phenotype. However, a caveat of this analysis is that the depletion of Sox9 also affects a much larger number of buffered genes, albeit to a lesser extent. The decreases in these other genes may also impact the phenotype, even if they are not as downregulated. It is also unclear what the y-axis (Predicted levels) means in Figure 5E. In Figure 5B, both curves are meant to be examples of sensitive pro-chondrogenic genes, but the curves look qualitatively different from each other, suggesting that comparisons of qualitative curve shapes might not be the most reliable measure. The claim about sensitive genes underlying the phenotype should be de-emphasized if correlative comparisons are the only line of evidence.

Author Rebuttal to Initial comments

Reviewer #1:

Remarks to the Author:

In this study, Naqvi et al. investigated the effects of dosing the levels of critical developmental and neural crest regulator, Sox9. They analysed the impact of such dosage on chromatin, gene expression and cellular and developmental phenotype level using an in vitro model of human cranial neural crest development. The study is carefully executed, appropriately controlled and brings new insight not only into the effects of the necessary dosage of Sox9 factor but also reveals critical effects that wary from the buffered response on chromatin and expression levels to profound consequences on a specific subset of heightened sensitivity regulatory elements genes, associated with Pierre Robin Sequence-like phenotypes and pro-chondrogenic programme. The authors propose a model that integrates genes/elements sensitive to Sox9 dosage, resulting in specific cellular and morphological effects. Although findings are not completely surprising, this study, for the first time, quantifies and interprets the impact of fine-tuning a critical developmental transcription factor at the mechanistic level, and as such, this

study is of high value.

We thank Reviewer 1 for their positive comments and for the succinct and accurate summary of our study.

The authors also assess the effect of other critical TFs on regulatory elements in conditions of varying SOX9 dosage. This is another crucial aspect of the study, as it investigates the competitive model of activity of other TFs substantially that would bind to regulatory elements and further decrease sensitivity/dependence on Sox9. This system and data offer an excellent model to investigate the dynamics of neural crest regulatory cores, both by looking and cooperative as well as competitive/compensating action. Unfortunately, this latter aspect was insufficiently explored in the study, which is a pity, as such analysis could contribute novel mechanistic insights into the nature of neural crest/craniofacial gene regulatory cores.

In a series of new experiments, we have performed ChIP-seq of both SOX9 (using the V5 tag) and another key CNCC TF (TWIST1) at 4 different SOX9 concentrations (100%, ~50%, ~20%, 0%). We focused on TWIST1 because of our previous observation that, among several examined TFs, TWIST1 binding showed the strongest effect on sensitivity of RE responses to SOX9 dosage (Fig. 3F). We used the newly generated ChIP-seq data to distinguish between two types of buffering mechanisms: 1) active compensation, where the binding of TWIST1 increases with partial SOX9 loss at buffered REs, and 2) synergy, where the combined actions of SOX9 and TWIST1 are sufficient to retain accessibility under partial loss of SOX9 dosage.

The data are overall more consistent with synergistic function of the TFs, as detailed in lines 273-288 and Extended Data Figure 6. REs most sensitive to SOX9 dosage in their chromatin accessibility were also most sensitive with respect to loss of SOX9 binding upon perturbation, whereas the accessibility-buffered REs were also buffered with respect to SOX9 binding (Extended Data Fig. 6B). In particular, we observed that at 50% SOX9 dosage, buffered sites are able to largely retain SOX9 binding (Extended Data Figure 6B). Inconsistent with a compensatory mechanism of buffering by other TFs, TWIST1 binding does not increase at buffered REs at 50% SOX9 dosage. However, both the fraction of TWIST1-bound REs and the overall level of TWIST1 binding (at full SOX9 dosage) are generally higher at the buffered as compared to the sensitive REs (Extended Data Figure 6A,C). Interestingly, stronger SOX9 perturbation (~20% of unperturbed levels or complete loss) results in diminished TWIST1 binding at both buffered and sensitive REs (Extended Data Figure 6D), consistent with a model in which SOX9 and TWIST1 binding is co-dependent at a subset of sites, but their synergistic function buffers such co-regulated REs against small changes in TF dosage (Extended Data Figure 6E).

We thank Reviewer 1 for the thoughtful suggestion.

Minor points:

1. GWAS identified for multiple essential genes – discussed only for Sox9 locus – can those be detailed for Sfrp2, Dlx5/6, Alx1 etc.?

We have highlighted some of the additional loci identified in the PRS endophenotype GWAS (lines 459-461). Notably, Sfrp2 has been shown to cause defects in mouse chondrogenesis, in line with the notion that defects in chondrogenesis drive much of the phenotype in PRS. Furthermore, both Dlx5 and Dlx6 are required for lower jaw identity in mice, consistent with the major mandibular defects in PRS that culminate in micrognathia.

2. It is not evident if the authors have evidence of synergistic binding in the condition of low Sox or if the only modus operandi is competition. Could this please be detailed?

The results of the newly-generated SOX9 (V5) and TWIST1 ChIP-seq in distinct SOX9 concentrations, detailed in lines 273-288 and Extended Data Figure 6, are consistent with synergistic binding of SOX9 and TWIST1 at buffered REs, as such REs retained nearly unperturbed levels of both TFs at ~50% SOX9 dosage. Furthermore, at lower SOX9 dosages (~20% and complete loss), TWIST1 binding is diminished at both SOX9-buffered and -sensitive REs, suggesting co-dependence of SOX9 and TWIST1 binding at a subset of sites.

Reviewer #2:

The study by Naqvi et al examined the quantitative impact of TF dosage on chromatin accessibility and gene expression using mathematical modelling of CNCC ATAC-seq, RNA-seq and new and publicly available TF ChIP-seq datasets. They showed that genes relating to chondrogenesis were particularly sensitive to SOX9 dosage and validated this in an in vitro model of chondrocyte differentiation. They further correlated the impact of these buffered or sensitive genes to normal craniofacial variability and disease manifestations, showing that at both ends of the buffered-sensitive gene spectrum there were high impact phenotypes.

The experimental design from basic transcription factor biology, to development, and correlation with disease phenotypes is excellent. The conclusions drawn are supported by the data shown and interpretations are mostly well explained. While the study's insights are novel for SOX9 and the graded dosage of SOX9 allows for fine separation of buffered v, sensitive genes, the

mechanisms for why some genes are buffered or sensitive to TF dosage could be expanded upon. Fortunately, the authors have the tools in hand to be able to do this.

We thank Reviewer 2 for their appreciation for the quality for our work and the succinct and accurate summary of our study.

The novelty of the study would be strengthened by additional SOX9 ChIP data. The inclusion of V5-ChIP-seq in cells with reduced SOX9 dosage would validate much of the mathematical modelling in Figure 2 and 3. It would also be a significant piece of data for the TF biology field, as the impact of partially reduced TF dosage, ie heterozygous loss, on DNA binding/occupancy has not been experimentally shown. Are the REs with palindrome motifs the first to lose SOX9 occupancy? This data should be integrated with the chromatin accessibility and RNA-seq results to bolster why some REs/genes are sensitive and others are buffered.

In a series of new experiments, we have performed V5 ChIP-seq in CNCCs with 4 different SOX9 concentrations (100%, ~50%, ~20%, 0%), the results of which are detailed in Extended Data Figure 6 and lines 273-282. We note that the V5 ChIP-seq signal was overall somewhat weak (and unfortunately, tested endogenous SOX9 antibodies did not perform well). Therefore, we were only able to examine the effects of SOX9 dosage on SOX9 binding at a subset of SOX9-dependent REs with sufficient SOX9-dependent V5 ChIP-seq signal (n=1,163). However, even with this subset of REs (and despite increased noise due to overall smaller numbers) we were able to draw valuable conclusions. We found that under partial (~50%) SOX9 dosage, SOX9 binding is not reduced uniformly across REs, but rather is retained at REs which are buffered in their accessibility and diminished at accessibility-sensitive REs (which are generally enriched for the SOX9 palindrome motif and depleted of binding by other TFs) (Extended Data Figure 6 A,B).

We also performed analogous ChIP-seq analyses of another TF (TWIST1) whose binding we previously found to be correlated with RE buffering against SOX9 dosage changes. Combined with the V5 ChIP-seq, these data suggest a model in which the synergistic function of TWIST1 with SOX9 at a subset of sites retains TF binding and accessibility at buffered REs under partial (~50%) SOX9 dosage; conversely, the lack of such other TFs at sensitive REs leads to greater loss of SOX9 binding under partial SOX9 dosage. These observations regarding TWIST1 binding are shown in Extended Data Figure 6A,C,D and E and we also discuss them in more detail in response to one of the points below.

We thank Reviewer 2 for the thoughtful suggestion.

Other comments:

Is there heterogeneity within the CNCC populations used in this study?

Regarding heterogeneity with respect to SOX9 levels, there is some heterogeneity (see the spread of mNeonGreen levels representing SOX9 levels in Figure 1D), but the response to dTAGV-1 treatment is uniform, indicated by the entire, unimodal distribution shifting upon addition of different dTAGV-1 concentrations. We have made this clear in the text (line 152).

Regarding general heterogeneity in the CNCC population, this CNCC differentiation protocol has been extensively characterized previously (Prescott et al, Cell 2015), using both flow cytometry of CNCC-specific multiple cell surface markers, as well as immunofluorescence of key TF markers. Both approaches point to a relatively homogenous population of CNCCs with consistently high expression of cell surface and TF markers in over 90% of cells within the population (Prescott et al, Cell 2015). We have noted this in the text (lines 139-140)

It's not exactly clear how Hill exponent gives requirement for SOX9 binding as currently written in the text and relating to Extended data 4C,D "The SOX9 palindrome with 3-5 bp spacing was also specifically associated with a modest increase in the Hill exponent, consistent with previous reports showing its requirement for cooperative SOX9 binding (page 6)" – could the biological relevance of this model be clarified in the text accompanying Figure 2 and the initial descriptions of the Hill and ED50 models.

We apologize for the unclear wording in this sentence. "its" refers to the SOX9 palindrome with 3-5 bp spacing rather than the Hill exponent. We have modified the text to make this distinction clear (lines 237-240). Our intention with this statement was simply to note that the presence of the SOX9 palindrome is associated with an increase in the Hill exponent, which is often interpreted as higher cooperativity. This increased cooperativity is consistent with the earlier studies demonstrating cooperative binding of SOX9 to the palindrome.

Could the proportion of sensitive v buffered REs and whether they increase or decrease in accessibility be stated more clearly in a figure or text. Might be 74% buffered, but text description doesn't make that clear (page 5) – the details of the mathematical modelling in the text distract from the important conclusions of Figure 2.

We thank Reviewer 2 for raising this point, and apologize for the lack of clarity in the text. The 74% statistic refers to the fraction of REs that are better fit by the Hill equation than a linear equation (as determined by AIC), but for a comparable measure of sensitivity we fit all REs with the Hill equation and compared the ED50 among them. Because the ED50 is a quantitative measure, the binary assignment of REs as buffered or sensitive is somewhat arbitrary. Nevertheless, we can provide approximate numbers based on ED50 ranges: Approximately 73% of REs have ED50 < 30 (which we consider mostly buffered), ~14% of REs have ED50 between 30 and 40 (which we consider somewhat sensitive), and ~13% of REs have ED50 > 40 (which we consider highly sensitive). The proportion of REs downregulated or upregulated in response to SOX9 depletion in each of these groups is relatively consistent, ~68% downregulated vs ~32% upregulated. We have updated the text to include these numbers (lines 209-212), as well as analogous numbers for SOX9-dependent genes (lines 299-301).

Could the authors speculate on the mechanisms by which other TF binding would change the sensitivity to SOX9 dosage?

We have performed ChIP-seq of one such other TF (TWIST1) at 4 different SOX9 concentrations (100%, ~50%, ~20%, 0%), the results of which are detailed in Extended Data Figure 6 and lines 283-288. Similar to the results for SOX9, TWIST1 binding at buffered REs is retained at nearly unperturbed levels under partial (~50%) SOX9 dosage. Interestingly, stronger SOX9 perturbation (~20% of unperturbed levels or complete loss) results in diminished TWIST1 binding at both buffered and sensitive sites (Extended Data Figure 6D), consistent with a model in which SOX9 and TWIST1 binding is co-dependent at a subset of sites, but their synergistic function buffers such co-regulated REs against small changes in TF dosage (Extended Data Figure 6E).

Are the changes in gene expression monotonic like chromatin accessibility? A heatmap for gene expression changes, and supplementary table of the list of genes and REs that are buffered v. sensitive would be beneficial to readers.

We have added a heatmap of gene expression changes at each SOX9 concentration (Extended Data Figure 7A). Like chromatin accessibility, changes in gene expression in response to SOX9 dosage are largely monotonic. We have provided lists of all SOX9-dependent REs and genes, along with their fitted ED50 values and sensitivity bins (three rough groupings) and 'buffering index' (an alternative measure of sensitivity), in Supplementary Tables 1 and 2, respectively.

In addition to cartilage development (Fig 5A), what other pathways show sensitivity to SOX9 dosage?

We have performed gene ontology enrichment to identify pathways specifically enriched in sensitive (ED50 > 30) genes. This analysis identified “cartilage condensation” as enriched, consistent with our previous analyses, as well as additional potentially important pathways known to be involved in craniofacial development such as TGF-beta and BMP. There were also a number of enriched pathways related to neuronal development and function, whose relevance is less straightforward to interpret, given that the CNCCs we studied are already committed to a mesenchymal lineage. We described these results in the text (lines 373-377) and have included lists of GO terms specifically enriched in sensitive genes in Supplementary Table 3.

What is the overlap between buffered v. sensitive genes from RNA-seq and RE closest genes from ATAC-seq?

We directly analyzed the relationship between SOX9-dependent gene sensitivity (as measured by ED50) and sensitivity of REs within 100kb of the corresponding TSS. This analysis indicates that more sensitive genes have more sensitive nearby REs. We detail these results in Extended Figure 8H and lines 355-356.

Reviewer #3

The manuscript submitted by Naqvi et al. presents an elegant method to study the dosage sensitivity of cells to transcription factor levels. The authors engineer cranial neural crest cells (CNCCs) so that SOX9 levels can be titrated down by the addition of a drug. They then measure the effects of decreasing SOX9 levels on the DNA accessibility of regulatory elements (REs) by ATAC-seq, on mRNA expression levels by RNA-seq, and on the in vitro differentiation of CNCCs. The main claims of the manuscript are 1) that the sequence properties of REs make them either sensitive or buffered to decreases in Sox9, 2) that the sensitivity of REs causes sensitivity in the expression of downstream mRNAs, and 3) that the craniofacial defects that result from decreases in Sox9 are caused by a subset of sensitive pro-chondrogenic genes. As written, the data do not strongly support these claims.

We thank Reviewer 3 for this summary of our paper. We have clarified and addressed their critiques of our claims below, and we believe that the revised manuscript only makes claims strongly supported by the data.

The claims rely heavily on the definition of sensitivity, which is defined as the empirical dose 50

(ED50) 'representing the SOX9 dosage at which the RE reaches half of its maximal levels' (page 5). However, the Hill equation in Figure 2C seems to contradict the subsequent figures. If the Hill equation in Figure 2C is correct, then as x (SOX9 concentration) increases y (ATAC CPM values) should decrease, which is counterintuitive and opposite of what is shown in Figure 2E and similar figures throughout the manuscript. The solid and dashed lines in Figure 2D should be swapped if the Hill equation in Figure 2C is correct. The authors should clarify the Hill equation in Figure 2C. If the equation in Figure 2C is not a typo, then the interpretation of the subsequent results would be compromised.

There was indeed a typo in Figure 2C – the quotient x/a should be flipped to a/x (now corrected in the revised manuscript). Nonetheless, all the plots of theoretical and observed fitted curves were calculated with the correct equation, and the typo was only present in the figure. We apologize for this error and are grateful to the Reviewer 3 for noticing the typo.

Some clarification should also be given as to how the Hill equation was fit to data. Specifically, were all REs fit using the Hill, or were some fit with the linear model? If both the linear model and the Hill were used, is it justified to compare ED50s fit from different models?. It is also unclear how the Hill equation with 3 parameters (mentioned in the Methods) was fitted, and for which REs it was used. It is again unclear whether the REs fitted with different numbers of parameters should be directly compared with each other.

Initially (prior to any comparison of ED50 values), we fit both linear and Hill equations to each RE/gene, finding that the Hill was a better fit in the majority of cases. Importantly, to allow for direct comparisons among REs, we only used the Hill equation for all subsequent analyses of sensitivity (ED50). We generally fit a 2 parameter Hill equation, except for a small fraction of REs (8%) and genes (5%) where the 3 parameter Hill (which allows for the maximum response to vary as a free, third parameter rather than constraining it as the response at 100% SOX9 dosage) was a significantly better fit, as determined by a decrease in AIC of 2 or more relative to the 2 parameter model. The type of Hill equation used for each gene/RE is indicated in Supplementary Tables 1 and 2. For REs (92%) and genes (95%) fit with the 2 parameter Hill, the 3 parameter Hill yields very similar ED50 point estimates ($\rho = 0.84$ and 0.80 , respectively), but with higher uncertainty (due to the inclusion of a third free parameter in the model). Therefore, we are confident that the use of a 3 parameter Hill for a small fraction of REs and genes does not lead to any type of bias in our results. We have added these additional details regarding modeling specifics to the Methods section 'Modeling of SOX9 dose-response curves (ATAC/RNA)'

Figure 4E is used to support the claim that the dose-response curves of REs predicts the dose-response of their target genes. However, the model in Figure 4E does not distinguish strongly

between high and low ED50 genes (right panel of Figure 4E) and fails to capture the qualitative difference in curve shape between genes with high vs low ED50s.

Subsequent analyses of mRNA ED50s (and the final model) assume a strong relationship between RE and gene ED50s despite Figure 4E. The final model presented in the paper explicitly predicts a correlation between the ED50s of REs and their target genes (“Genes with nearby sensitive REs will themselves show more sensitive responses to SOX9 dosage” (Page 11)) but this correlation is not shown.

We acknowledge that the predicted responses do not perfectly capture the qualitative shape of the dosage response for all genes – indeed, prediction of gene expression from chromatin and/or sequence features in the context of complex regulatory landscapes has been a huge challenge for the field for well over a decade. We have revised our claim to say that RE dose-response curves ‘partially predict’ gene dose-response curves (i.e. lines 38, 291, 330), as our predictions do in fact capture the shapes of some genes remarkably well, see the examples in Extended Data Figure 8F.

The reviewer’s comment has inspired us to conduct additional analyses to understand why the predictions are highly accurate for some genes, but not for others. As might have been expected, we found that genes with accurate predictions tend to have more SOX9-dependent REs in their vicinity (i.e. within 100 kb of their TSS) (Extended Data Figure 8G). However, it is important to note that even genes that are sensitive to SOX9 loss typically have many REs in the vicinity that are not SOX9-dependent. Given the complexity of mammalian cis-regulatory landscapes, current RE-gene models (such as the Activity-by-Contact model) do not accurately capture all regulatory inputs into gene activity, confounding the predictions for genes with many SOX9-independent REs. Nevertheless, when the overall regulatory effects near a gene are substantial enough such that a large fraction of REs change in response to SOX9, accurate predictions are possible.

In Extended Data Figure 8H we have now directly analyzed the relationship between SOX9-dependent gene sensitivity (as measured by ED50) and median sensitivity of SOX9-dependent REs within 100kb of the corresponding TSS. This analysis supports our claim that more sensitive genes have nearby REs that are on average more sensitive to SOX9 dosage. We discuss these results in lines 355-356.

The final claim of the paper is that pro-chondrogenic genes are particularly sensitive to SOX9 depletion, and that their sensitivity explains the craniofacial-related phenotypic sensitivity to

SOX9. The result in Figure 5E is intriguing and shows that the dose-response of a small set of sensitive genes closely follows the response of the differentiation phenotype. However, a caveat of this analysis is that the depletion of Sox9 also affects a much larger number of buffered genes, albeit to a lesser extent. The decreases in these other genes may also impact the phenotype, even if they are not as downregulated.

We agree with this point, and in fact we have already discussed it in the original manuscript in the context of common trait variation, where we have stated: “These minimal effects can contribute substantially to variation when summed up genome-wide, but most effects with individually appreciable contributions to variation result from a subset of REs and genes sensitive to SOX9 dosage changes.” Our main point is about the relative impact of sensitive vs buffered genes on phenotypes. We have modified the text in various places to state that sensitive genes/REs ‘primarily’ contribute to trait variation/disease phenotypes, to emphasize the point that they may not be the sole contributors (i.e. lines 428, 549).

It is also unclear what the y-axis (Predicted levels) means in Figure 5E.

The y-axis in Figure 5E is an arbitrary measure, indicating relative fitted levels of the dependent variables (colors), since the different curves are based solely on the Hill equation for different groups of REs or genes (or sGAG content). We have revised the axis label to indicate this (‘Relative fitted levels’).

In Figure 5B, both curves are meant to be examples of sensitive pro-chondrogenic genes, but the curves look qualitatively different from each other, suggesting that comparisons of qualitative curve shapes might not be the most reliable measure.

We agree that solely qualitative comparison of dosage response curves is challenging – throughout the manuscript, we accompany plots of dosage response curves with quantitative comparisons of the relevant metrics, typically ED50. We have added a plot that formally compares the ED50 of all REs, pro-chondrogenic genes, and other genes with the ED50 of sGAG content (the differentiation phenotype). This analysis (Extended Data Figure 9D) indicates that the estimated ED50 for the sGAG content is outside the range of all other genes, but within the range of pro-chondrogenic genes.

The claim about sensitive genes underlying the phenotype should be de-emphasized if correlative comparisons are the only line of evidence.

We have modified our language in various places throughout the manuscript to state that it is 'likely' that SOX9-sensitive genes underlie craniofacial phenotypes (i.e. lines 428, 504, 507, 551). However, we would like to note that we provide additional evidence beyond the chondrocyte differentiation phenotype that also supports this claim. In Figure 6A, we show that genes with associated PRS-like disorders (which are partially, but not completely, overlapping with pro-chondrogenic genes) are sensitive to SOX9 dosage. In Figure 6E, we show that REs linked to SNPs associated with a PRS endophenotype in healthy individuals are more sensitive to SOX9 dosage than REs associated with other facial shape effects.

Decision Letter, first revision:

12th Dec 2022

Dear Joanna,

Your Article, "Precise modulation of transcription factor levels reveals drivers of dosage sensitivity" has now been seen by the original 3 referees. You will see from their comments below that while they continue to find your work of interest, there are still some important points raised. We remain interested in the possibility of publishing your study in Nature Genetics, but would like to consider your response to these concerns in the form of a revised manuscript before we make a final decision on publication.

Briefly, two reviewers (#1 and #2) are satisfied and now support publication.

Conversely, Reviewer #3 - previously critical - appreciates the revision but does not think that your data supports the proposed model of buffering. They suggest that a simpler and as-consistent interpretation of the evidence presented is that the differences in RE activity and gene expression downstream of Sox9 activity is due to a distinction between direct and indirect Sox9 targets, rather than buffering.

While we think that this interpretation is not unreasonable, we editorially consider these two mechanisms (in/direct effects vs buffering) as not necessarily mutually exclusive and we would like to consider your responses to this concern. We note that Reviewer #3 has some misconceptions of the data presented (e.g. they say that Fig 2b shows two examples of buffered genes, when one is sensitive). We also note that there is substantial data in e.g. the ED Figures that might also be of use to the reader in clarifying the mechanism. Overall, thus, we think that there is a path to publication and would encourage you to respond as fully as possible to this important point.

To guide the scope of the revisions, the editors discuss the referee reports in detail within the team, including with the chief editor, with a view to identifying key priorities that should be addressed in revision and sometimes overruling referee requests that are deemed beyond the scope of the current study. We hope that you will find the prioritized set of referee points to be useful when revising your study. Please do not hesitate to get in touch if you would like to discuss these issues further.

We therefore invite you to revise your manuscript taking into account all reviewer and editor comments. Please highlight all changes in the manuscript text file. At this stage we will need you to upload a copy of the manuscript in MS Word .docx or similar editable format.

*2) If you have not done so already please begin to revise your manuscript so that it conforms to our Article format instructions, available

[here](http://www.nature.com/ng/authors/article_types/index.html).

*3) Include a revised version of any required Reporting Summary:

[redacted]

We hope to receive your revised manuscript within four to eight weeks. If you cannot send it within this time, please let us know.

Sincerely,

Michael Fletcher, PhD
Senior Editor, Nature Genetics

ORCID: 0000-0003-1589-7087

Reviewers' Comments:

Reviewer #1:

Remarks to the Author:

In this revision, the authors have addressed my main concerns satisfactorily. They have performed new experiments involving Sox9 ChIP at varying concentrations in the presence of other NC factors. Careful interpretation of the data and consideration of two models of interaction between the factors answers the question about the synergistic versus competitive activity in the tested case. I am happy with the revision and would recommend the manuscript for publication.

Reviewer #2:

Remarks to the Author:

The authors have addressed my main and minor queries.

Reviewer #3:

Remarks to the Author:

This is a review of the revised manuscript from Naqvi et al, which reports the effects of Sox9 perturbations on chromatin accessibility, mRNA expression, and chondrocyte differentiation. The strength of the manuscript is the attempt to address the quantitative effects of Sox9 levels on relevant endophenotypes. The primary claim of the manuscript is that regulatory elements are either 'buffered' or 'sensitive' and that these behaviors predict the effects of perturbations on gene expression and ultimately chondrocyte differentiation and variation in facial structure. The data do not support these

conclusions. Instead, the data support the much simpler conclusions that the direct targets of Sox9 have larger effects than the indirect targets of Sox9. The concept of buffering is an unnecessary addition that complicates the analysis of a simple set of perturbations.

As in the original submission, Figure 4E still does not support the claim that RE dose-response curves 'partially' predicts mRNA responses. The ABC model does not appear to capture even qualitative differences in mRNA. In a related point, the two examples of sensitive genes in Figure 2B have very different qualitative responses, with COL2A1 appearing to be buffered even though it is discussed as being sensitive. These results are not just examples of models not quite capturing a quantitative effect, as the authors imply in their response. These results cast doubt on the central claim that there are significant mechanistic differences between the effects of buffered and sensitive elements.

The primary weakness of the manuscript is that the concept of sensitive/buffered regulatory elements is not justified from the data, nor is it helpful in explaining the trends in the data. A much simpler interpretation of the data is that sensitive elements are the direct targets of Sox9 while buffered elements are indirect targets of Sox9. This interpretation is consistent with all the data in the manuscript, and the authors imply this throughout the manuscript. Thus, there is no need to introduce the extra concept of sensitive/buffered elements.

Minor:

Figure 4 legend, (F) should be (E), 'grey' should be green.

Type in Extended Data Figure 5a axis label.

Author Rebuttal, first revision:

Reviewer #3: Remarks to the Author: This is a review of the revised manuscript from Naqvi et al, which reports the effects of Sox9 perturbations on chromatin accessibility, mRNA expression, and chondrocyte differentiation. The strength of the manuscript is the attempt to address the quantitative effects of Sox9 levels on relevant endophenotypes. The primary claim of the manuscript is that regulatory elements are either 'buffered' or 'sensitive' and that these behaviors predict the effects of perturbations on gene expression and ultimately chondrocyte differentiation and variation in facial structure. The data do not support these conclusions. Instead, the data support the much simpler conclusions that the direct targets of Sox9 have larger effects than the indirect targets of Sox9. The concept of buffering is an unnecessary addition that complicates the analysis of a simple set of perturbations. The primary weakness of the manuscript is that the concept of sensitive/buffered regulatory elements is not justified from the data, nor is it helpful in explaining the trends in the data. A much simpler interpretation of the data is that sensitive elements are the direct targets of Sox9 while buffered elements are indirect targets of Sox9. This interpretation is consistent with all the data in the manuscript, and the authors imply this throughout the manuscript. Thus, there is no need to introduce the extra concept of sensitive/buffered elements. There seem to be two claims that Reviewer 3 is making in these two paragraphs. With all due respect, we believe that both claims are incorrect and may result from a misunderstanding of the results shown in Figs. 2 and 3: Their first claim is that "the concept of sensitive/buffered regulatory elements is

not justified from the data.” We believe that this claim stems from a misunderstanding of the use of the terms “sensitive” and “buffered.” In our manuscript, we use the terms “sensitive” and “buffered” to refer to the shape of the SOX9 dosage response curve, as reflected in the ED50 parameter that provides a measure of the TF dosage at which target RE accessibility or gene expression reaches half-maximal levels. Figure 2E show REs with clearly distinct shapes of SOX9 dosage response, which we term “sensitive” and “buffered.” We note that similar distinctions between the shape of dose-response curves have been previously made in the literature, using slightly different but analogous terminology: Giorgetti et al (2010, Molecular Cell) use the term “graded” to refer to what we term “sensitive” responses, while Janes et al (2008, Cell) use the term “saturated” to refer to what we term “buffered” responses. We now cite these previous studies (lines 196-198). The nomenclature we use is independent of the underlying mechanism that results in such differing responses. Indeed, Reviewer 3 seems to acknowledge as much, as they use the terms “sensitive” and “buffered” to distinguish SOX9-dependent REs in their own comments. Reviewer 3’s second claim is that variation among REs in such sensitivity versus buffering can be fully explained by the idea that direct SOX9 targets are sensitive while indirect targets are buffered. Assuming that Reviewer 3 agrees with our functional definition of direct versus indirect targets (directly regulated REs being rapidly downregulated upon loss of SOX9, while indirect RE show delayed effects after full depletion), we believe that this claim also stems from a misunderstanding of our analyses. While Figures 3A-D and the associated text do indeed show that such direct targets are more sensitive than indirect targets, Figure panels 3E-G, as well as Extended Data Figures 5 and 6, all show features correlated with sensitivity to SOX9 dosage among all (> 9,000) direct SOX9 targets. Furthermore, in the original manuscript, we stated “We sought to identify additional features associated with sensitivity among direct SOX9 targets.” The entire remainder of the section entitled “Feature affecting sensitivity of the RE response to SOX9 dosage” discusses these additional features. Reviewer 3’s proposed simpler interpretation of the data that “sensitive elements are the direct targets of SOX9 while buffered elements are indirect targets of SOX9” cannot explain these differences. We have modified Figures 3, Extended Data Figure 6, and the text (in the revised manuscript, lines 247-251, 284, 294-297, 299, 313, 524-525) to make this clear and avoid future misunderstanding. As in the original submission, Figure 4E still does not support the claim that RE dose-response curves ‘partially’ predicts mRNA responses. The ABC model does not appear to capture even qualitative differences in mRNA. At this point it is unclear what Reviewer 3 considers as adequate predictions of dose-response curves. There are clear statistical and visual differences between the predicted curves for the different sensitivity bins. We further point Reviewer 3 to Extended Data Figure 8F for two clear examples where the ABC model captures qualitative differences in the SOX9 dosage response for sensitive versus buffered genes. In our view, while the predictions may not be perfect, this is ample evidence that the simple model we developed ‘partially’ predicts dose-response curves. In a related point, the two examples of sensitive genes in Figure 2B have very different qualitative responses, with COL2A1 appearing to be buffered even though it is discussed as being sensitive. These results are not just examples of models not quite capturing a quantitative effect, as the authors imply in their response. These results cast doubt on the central claim that there are significant mechanistic differences

between the effects of buffered and sensitive elements. There appears to be a typo in this comment, as Figure 2B shows examples of RE responses to SOX9 dosage, not genes. We assume Reviewer 3 is referring to Figure 5B. The ED50 of the COL2A1 dose-response curve is 31. This puts it in the “moderately sensitive” category of genes that we define in the previous section (ED50 between 30 and 40), clearly more sensitive than buffered genes such as TGFBI (Figure 4A). COL11A1 is more sensitive still. We have modified Figure 5B and the text (line 409) to indicate COL2A1 as “moderately sensitive” and COL11A1 as “highly sensitive.” Minor: Figure 4 legend, (F) should be (E), ‘grey’ should be green. Type in Extended Data Figure 5a axis label. We have corrected these typos

Decision Letter, second revision:

11th Jan 2023

Dear Joanna,

Thank you for submitting your revised manuscript "Precise modulation of transcription factor levels reveals drivers of dosage sensitivity" (NG-A60334R1). We've made an editorial check of your responses to the last remaining comments from Reviewer #3 and we think they are satisfactory, such that no further review will be required. Therefore we'll be happy in principle to publish it in Nature Genetics, pending minor revisions to satisfy the referees' final requests and to comply with our editorial and formatting guidelines.

Sincerely,

Michael Fletcher, PhD
Senior Editor, Nature Genetics

Final Decision Letter:

7th Mar 2023

Dear Joanna,

I am delighted to say that your manuscript "Precise modulation of transcription factor levels identifies features underlying dosage sensitivity" has been accepted for publication in an upcoming issue of

Nature Genetics.

Your paper will be published online after we receive your corrections and will appear in print in the next available issue. You can find out your date of online publication by contacting the Nature Press Office (press@nature.com) after sending your e-proof corrections. Now is the time to inform your Public Relations or Press Office about your paper, as they might be interested in promoting its publication. This will allow them time to prepare an accurate and satisfactory press release. Include your manuscript tracking number (NG-A60334R2) and the name of the journal, which they will need when they contact our Press Office.

Please note that *Nature Genetics* is a Transformative Journal (TJ). Authors may publish their research with us through the traditional subscription access route or make their paper immediately open access through payment of an article-processing charge (APC). Authors will not be required to make a final decision about access to their article until it has been accepted. [Find out more about Transformative Journals](https://www.springernature.com/gp/open-research/transformative-journals)

Authors may need to take specific actions to achieve [compliance with funder and institutional open access mandates](https://www.springernature.com/gp/open-research/funding/policy-compliance-faqs). If your research is supported by a funder that requires immediate open access (e.g. according to a

[Plan S principles](https://www.springernature.com/gp/open-research/plan-s-compliance)) then you should select the gold OA route, and we will direct you to the compliant route where possible. For authors selecting the subscription publication route, the journal's standard licensing terms will need to be accepted, including <https://www.nature.com/nature-portfolio/editorial-policies/self-archiving-and-license-to-publish>. Those licensing terms will supersede any other terms that the author or any third party may assert apply to any version of the manuscript.

Please note that Nature Portfolio offers an immediate open access option only for papers that were first submitted after 1 January, 2021.

If you have not already done so, we invite you to upload the step-by-step protocols used in this manuscript to the Protocols Exchange, part of our on-line web resource, natureprotocols.com. If you complete the upload by the time you receive your manuscript proofs, we can insert links in your article that lead directly to the protocol details. Your protocol will be made freely available upon publication of your paper. By participating in natureprotocols.com, you are enabling researchers to more readily reproduce or adapt the methodology you use. [Natureprotocols.com](https://natureprotocols.com) is fully searchable, providing your protocols and paper with increased utility and visibility. Please submit your protocol to <https://protocolexchange.researchsquare.com/>. After entering your [nature.com](https://www.nature.com) username and password you will need to enter your manuscript number (NG-A60334R2). Further information can be found at <https://www.nature.com/nature-portfolio/editorial-policies/reporting-standards#protocols>

Sincerely,

Michael Fletcher, PhD
Senior Editor, Nature Genetics

ORCID: 0000-0003-1589-7087